**Letter** https://doi.org/10.1038/s41556-023-01178-w

# The Troyer syndrome protein spartin mediates selective autophagy of lipid droplets

Jeeyun Chung [1,2,7], Joongkyu Park [3], Zon Weng Lai[1,2], Talley J. Lambert [1,4], Ruth C. Richards[1,2], Jiuchun Zhang[1], Tobias C. Walther [1,2,5,6,8,9] ✉ & Robert V. Farese Jr. [1,2,5,8,9] ✉

Lipid droplets (LDs) are crucial organelles for energy storage and lipid homeostasis. Autophagy of LDs is an important pathway for their catabolism, but the molecular mechanisms mediating LD degradation by selective autophagy (lipophagy) are unknown. Here we identify spartin as a receptor localizing to LDs and interacting with core autophagy machinery, and we show that spartin is required to deliver LDs to lysosomes for triglyceride mobilization. Mutations in *SPART* (encoding spartin) lead to Troyer syndrome, a form of complex hereditary spastic paraplegia[1]. Interfering with spartin function in cultured human neurons or murine brain neurons leads to LD and triglyceride accumulation. Our identification of spartin as a lipophagy receptor, thus, suggests that impaired LD turnover contributes to Troyer syndrome development.

LDs are ubiquitous cytoplasmic organelles that store neutral lipids, such as triglycerides (TGs) or sterol esters, as reservoirs of metabolic fuel and membrane lipid precursors[2]. As such, they play key roles in metabolism and physiology, and abnormalities in LD biology are increasingly recognized as causes of human disease[3,4].

Key to LD physiology is the ability of cells to mobilize lipids from LDs. This occurs by one of two pathways[5]. First, lipids can be mobilized from LDs by a series of hydrolytic reactions, known as lipolysis, that is initiated by TG hydrolases, such as adipose TG lipase (ATGL), hormone-sensitive lipase or brain TG lipase (DDHD2) (refs. 6–8). Alternatively, selective autophagy delivers LDs to lysosomes, where their lipids are degraded by lysosomal acid lipase in a process known as lipophagy[9]. While lipophagy has been recognized for more than a decade and appears to operate in many cell types[5,9–12], the molecular mechanisms and physiological functions of lipophagy remain unclear.

One factor limiting our understanding of the contributions of lipophagy to lipid homeostasis is that the mechanisms and protein machinery linking LDs to autophagy are largely unknown[13]. In this Letter, we sought to identify such machinery. Selective targeting of LDs to lysosomes predicts the existence of a receptor protein that interacts with both LDs and autophagic machinery, thereby localizing LDs to lysosomal compartments. One candidate for such a receptor is spartin (Spg20). Spartin was previously found to localize to both LDs[14] and endosomal compartments[15,16]. It contains a ubiquitin-binding region (UBR)[17,18], and ubiquitin binding is a feature of other receptors in selective autophagy, such as p62/SQSTM1 (ref. 19). These findings suggested the hypothesis that spartin may act as a lipophagy receptor.

To determine whether endogenous spartin localizes to LDs, as observed by immunofluorescence[15], we genome-engineered human SUM159 cells to express spartin tagged with mScarlet-I from its endogenous genomic locus (Extended Data Fig. 1a). For comparison with

[1]Department of Cell Biology, Harvard Medical School, Boston, MA, USA. [2]Department of Molecular Metabolism, Harvard T. H. Chan School of Public Health, Boston, MA, USA. [3]Department of Pharmacology, Department of Neurology, Wayne State University School of Medicine, Detroit, MI, USA. [4]Department of Systems Biology, Harvard Medical School, Boston, MA, USA. [5]Broad Institute of Harvard and MIT, Cambridge, MA, USA. [6]Howard Hughes Medical Institute, Boston, MA, USA. [7]Present address: Department of Molecular and Cellular Biology, Harvard University, Cambridge, MA, USA. [8]Present address: Sloan Kettering Institute, Memorial Sloan Kettering Cancer Center, New York, NY, USA. [9]These authors contributed equally: Tobias C. Walther, Robert V. Farese Jr. ✉e-mail: twalther@mskcc.org; rfarese@mskcc.org

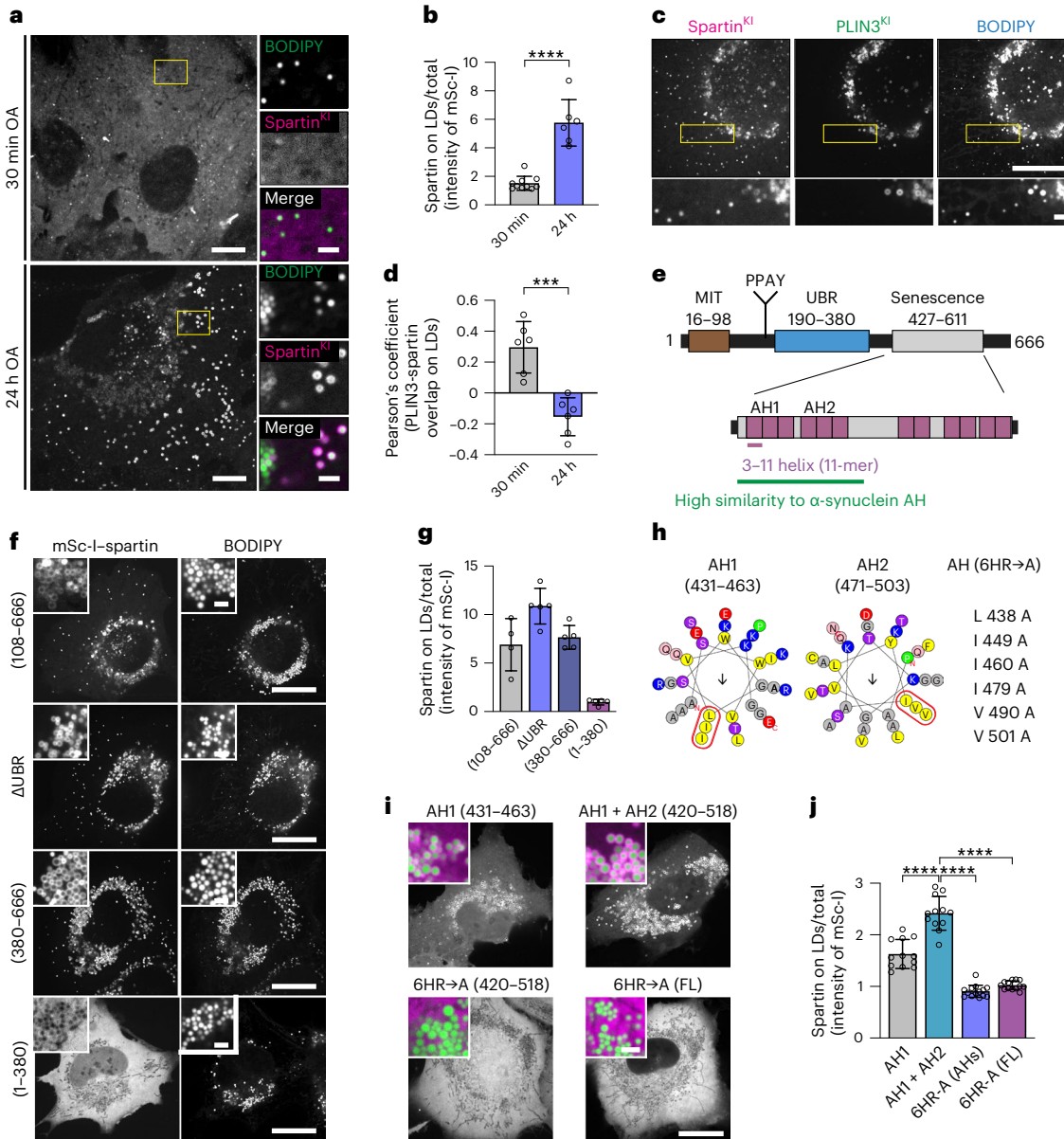

**Fig. 1 | Spartin targets to mature LDs via AH repeats in the senescence domain. a**, Cells expressing endogenously tagged spartin (with mScarlet-I) reveal preferential targeting to mature LDs. SUM159 cells treated with 0.5 mM OA for 30 min (top) or 24 h (bottom) and stained with BODIPY493/503. Scale: full-size, 20 μm; insets, 2 μm. **b**, Quantification of **a**. Mean ± s.d., *n* = 6 fields of view (24 h), 10 fields of view (30 min), three independent experiments, ****P = 0.0003, two-tailed unpaired *t*-test. **c**, Spartin and PLIN3 localize to different LD subpopulations. Cells expressing endogenously tagged spartin (with HaloTag) and PLIN3 (with mScarlet-I) treated with 0.5 mM OA (24 h) and stained with BODIPY493/503. HaloTag pre-labelled with 100 nM JF646. Scale bars as in **a**. **d**, Overlap of PLIN3 and spartin on LDs after 0.5 mM OA treatment, Pearson's coefficient analysis. Mean ± s.d., *n* = fields of view, three independent experiments, ***P < 0.001, two-tailed unpaired *t*-test. **e**, Schematic representation of spartin (top) with long AH regions (purple) in the senescence domain (bottom). **f**, Localization of expressed spartin truncation mutants

(with mScarlet-I tag) reveals spartin senescence domain is required for LD localization. Cells were treated with 0.5 mM OA (24 h) and stained with BODIPY493/503. Scale bars: full-size, 20 μm; insets, 2 μm. **g**, Quantification of **f**. Mean ± s.d., *n* = 4 cells (108–666), 5 cells (ΔUBR), 5 cells (380–666) and 5 cells (1–380). **h**, Helical wheel plot of spartin 33-mer repeats of AH1 (amino acids 431–463) and AH2 (amino acids 471–503), plotted as a 3–11 helix (*36*). AH(6HR→A), six mutations introduced into AH1 + AH2 (amino acids 431–503) and full length (FL). **i**, Cells expressing mScarlet-I-tagged spartin as in **h** show that AH1 and AH2 are sufficient for LD binding. Cells were treated with 0.5 mM OA (24 h) and stained with BODIPY493/503. Insets, overlay of spartin (magenta) and LDs (green). Scale bars as in **a**. **j**, Quantification of **h**. Mean ± s.d., *n* = 12 fields of view (AH1), 12 fields of view (AH1 + AH2), 14 fields of view (6HR-A[AHs]) and 13 fields of view (6HR-A[FL]), three independent experiments, ****P < 0.0001, one-way analysis of variance, Tukey's multiple comparisons test. Source numerical data are available in source data.

an LD marker protein, we also genome-engineered a cell line with both mScarlet-I–PLIN3 and HaloTag–spartin tagged at their endogenous loci (Extended Data Fig. 1a). We investigated the localization of spartin by fluorescence microscopy in cells grown with medium containing oleic acid (OA) to induce LD accumulation. Spartin was highly enriched at many LDs stained with BODIPY493/503 or labelled with *LiveDrop*,

a fluorescent LD biomarker generated from the hairpin motif of *Drosophila* GPAT4 (amino acids 152–308) (ref. 20) (Fig. 1a,b and Extended Data Fig. 1b,c). Spartin localization to LDs was much more apparent 24 h after oleate treatment than in the basal condition or 30 min after oleate addition (Fig. 1a,b and Extended Data Fig. 1d). In comparison with PLIN3, which targeted both nascent and mature

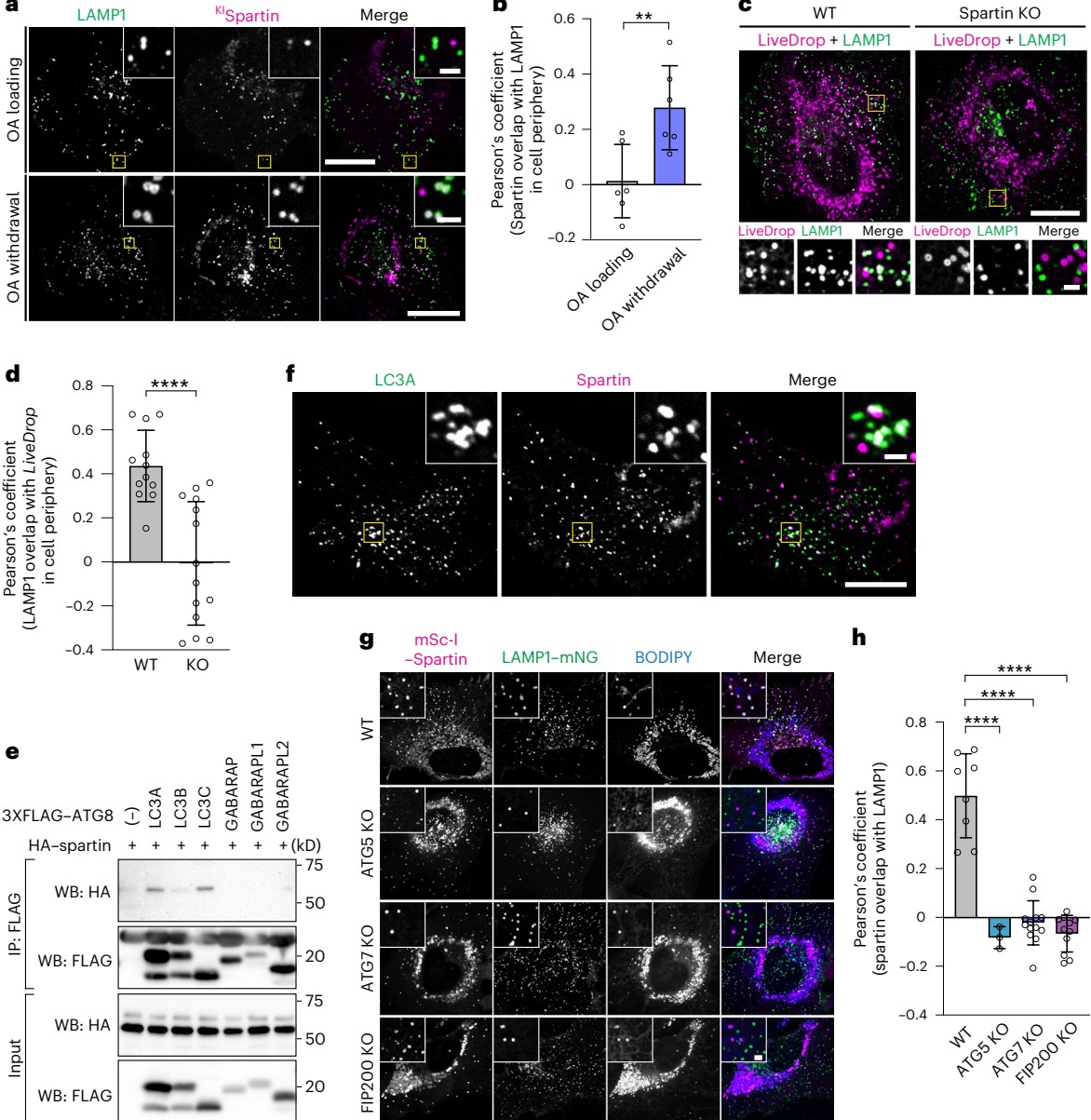

**Fig. 2 | Spartin mediates autophagy-dependent LD delivery to lysosomes.**
**a**, Cells expressing endogenously tagged spartin (with mScarlet-I) and transiently expressing LAMP1–mNG reveal spartin and LAMP1 co-localization after oleate withdrawal. SUM159 cells treated with 0.5 mM OA for 24 h (top) and chased for 3 h after OA withdrawal (bottom). Scale bars: full-size, 20 μm; insets, 2 μm. **b**, Overlap of spartin and LAMP1 in cell periphery with or without OA withdrawal for 2 h after 18 h OA addition as shown in **a** (Pearson's coefficient analysis). Mean ± s.d., *n* = 6 fields of view, three independent experiments, **P = 0.0091, two-tailed unpaired *t*-test. **c**, WT or spartin KO cells transiently co-expressing HaloTag–*LiveDrop* (pre-labelled with 100 nM JF549) and LAMP1–mNG reveal spartin deficiency impairs LD targeting to lysosomes. Cells treated with 0.5 mM OA (24 h) and chased for 3 h after OA withdrawal. Scale bars as in **a**. **d**, Overlap of *LiveDrop* and LAMP1 in cell periphery shown in **c** quantified by Pearson's coefficient analysis. Mean ± s.d., *n* = 12 cells (WT) and 14 cells (KO) from three independent experiments, ****P < 0.001, two-tailed unpaired *t*-test. **e**, Spartin interacts with LC3A and LC3C.

HEK293T cells transiently co-expressing HA-spartin together with 3xFLAG–ATG8 (LC3A, LC3B, LC3C, GABARAP, GABARAPL1 or GABARAPL2) were subjected to immunoprecipitation with anti-FLAG antibody and analysed by immunoblot with anti-HA and anti-FLAG antibodies. **f**, Spartin co-localizes with LC3A. Cells expressing mScarlet-I–spartin full-length and HaloTag–LC3A (pre-labelled with 100 nM JF646) were treated with 0.5 mM OA for 24 h and chased for 3 h after OA withdrawal. Scale bars: full-size, 20 μm; insets, 2 μm. **g**, Delivery of spartin-coated LDs to lysosomes is autophagy-dependent. Co-localization analyses between transiently overexpressed mScarlet-I–spartin and LAMP1–mNG in SUM159 cells lacking ATG5, ATG7 or FIP200. Scale bars: full-size, 20 μm; insets, 2 μm. **h**, Overlap of spartin and LAMP1 in cell periphery shown in **g** (Pearson's coefficient analysis). Mean ± s.d., *n* = 6 cells (WT), 11 cells (FIP200 KO), 13 cells (ATG7 KO) and 3 cells (ATG5 KO) cells from three independent experiments, ****P < 0.001, one-way analysis of variance, Dunnett's multiple comparisons test. Source numerical data and unprocessed blots are available in source data.

LDs (Extended Data Fig. 1e,f), spartin localized primarily to a subset of mature LDs that contained less PLIN3 (Fig. 1c,d). PLIN3 knockdown did not impair spartin recruitment to LDs, but slightly enhanced it (Extended Data Fig. 1g–i), suggesting that PLIN3 and spartin may compete in targeting LDs. Moreover, the LD population that

contained spartin was localized preferentially in the cell periphery (Fig. 1c,d). In contrast to a previous report that studied overexpressed spartin[21], we found no evidence for endogenously tagged spartin co-localizing with the ESCRT-III protein IST1 at the midbody (Extended Data Fig. 1j).

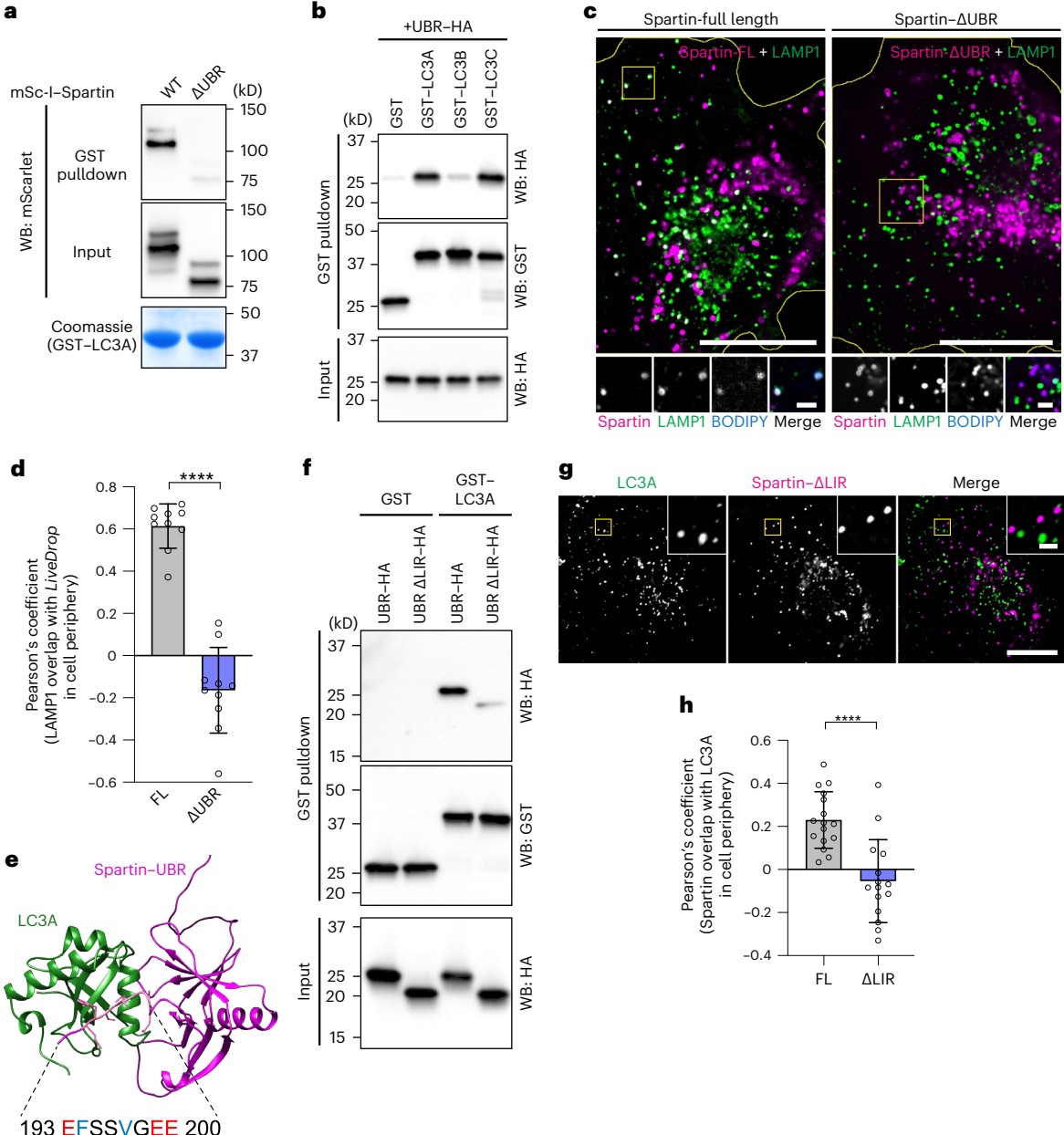

**Fig. 3 | UBR domain contains an LIR motif responsible for the interaction between spartin and LC3. a**, UBR domain is required for the interaction between spartin and LC3A. HEK293T WT cell lysates transiently expressing mScarlet-I–spartin WT or ΔUBR were incubated with recombinant GST–LC3A, followed by GST pulldown and detected with anti-mScarlet/mCherry antibody. Cells were treated will 0.5 mM OA for 24 h before lysate preparation. **b**, Recombinant UBR domain directly interacts with recombinant LC3A and LC3C. Recombinant spartin-UBR–HA was incubated with recombinant GST, GST–LC3A, GST–LC3B or GST–LC3C, followed by GST pulldown and detected with anti-GST and anti-HA antibodies. **c**, SUM159 cells lacking spartin, transiently expressing LAMP1–mNG and mScarlet-I–spartin full-length (FL) or mScarlet-I–spartin–ΔUBR. Cells were treated with 0.5 mM OA for 24 h, then replaced to the complete medium 3 h before image acquisition. LDs were stained with BODIPY493/504. Scale bars: full-size, 20 μm; insets, 5 μm. **d**, Overlap of spartin FL or ΔUBR and LAMP1 in cell periphery, quantified by Pearson's coefficient analysis. Mean ± s.d.,

$n = 10$ cells from three independent experiments, ****$P < 0.0001$, two-tailed unpaired $t$-test. **e**, AlphaFold2-based ColabFold structural prediction of the interaction between the UBR domain of spartin and LC3A. **f**, Recombinant spartin–UBR–HA or spartin-UBR ΔLIR (deletion of residues 193–200)–HA was incubated with recombinant GST or GST–LC3A, followed by GST pulldown and detected with anti-GST and anti-HA antibodies. **g**, Confocal imaging of live SUM159 cells lacking spartin, transiently expressing EGFP–LC3A and mScarlet-I–spartin FL or mScarlet-I–spartin–ΔLIR. Cells were treated with 0.5 mM OA for 24 h, then replaced to the complete medium 3 h before image acquisition. Scale bars: full-size, 20 μm; insets, 2 μm. **h**, Overlap between spartin FL or ΔLIR and LC3A in cell periphery shown in **g** was quantified by Pearson's coefficient analysis. Mean ± s.d., $n = 16$ cells (Spartin FL) and 15 cells (ΔLIR) from three independent experiments, ****$P < 0.0001$, two-tailed unpaired $t$-test. Source numerical data and unprocessed blots are available in source data.

We next investigated how spartin localizes to LDs. Spartin contains three evolutionarily conserved domains—a 'microtubule-interacting and trafficking' (MIT) domain[22], a central domain that has been reported to bind ubiquitin (known as spartin UBR[17]) and a 'plant-related

senescence domain (senescence domain)'[23,24] (Fig. 1e). The C-terminal senescence domain was implicated in binding to LDs[14,24]. Our domain truncation analyses of expressed spartin protein revealed that the senescence domain is indeed necessary and sufficient for spartin

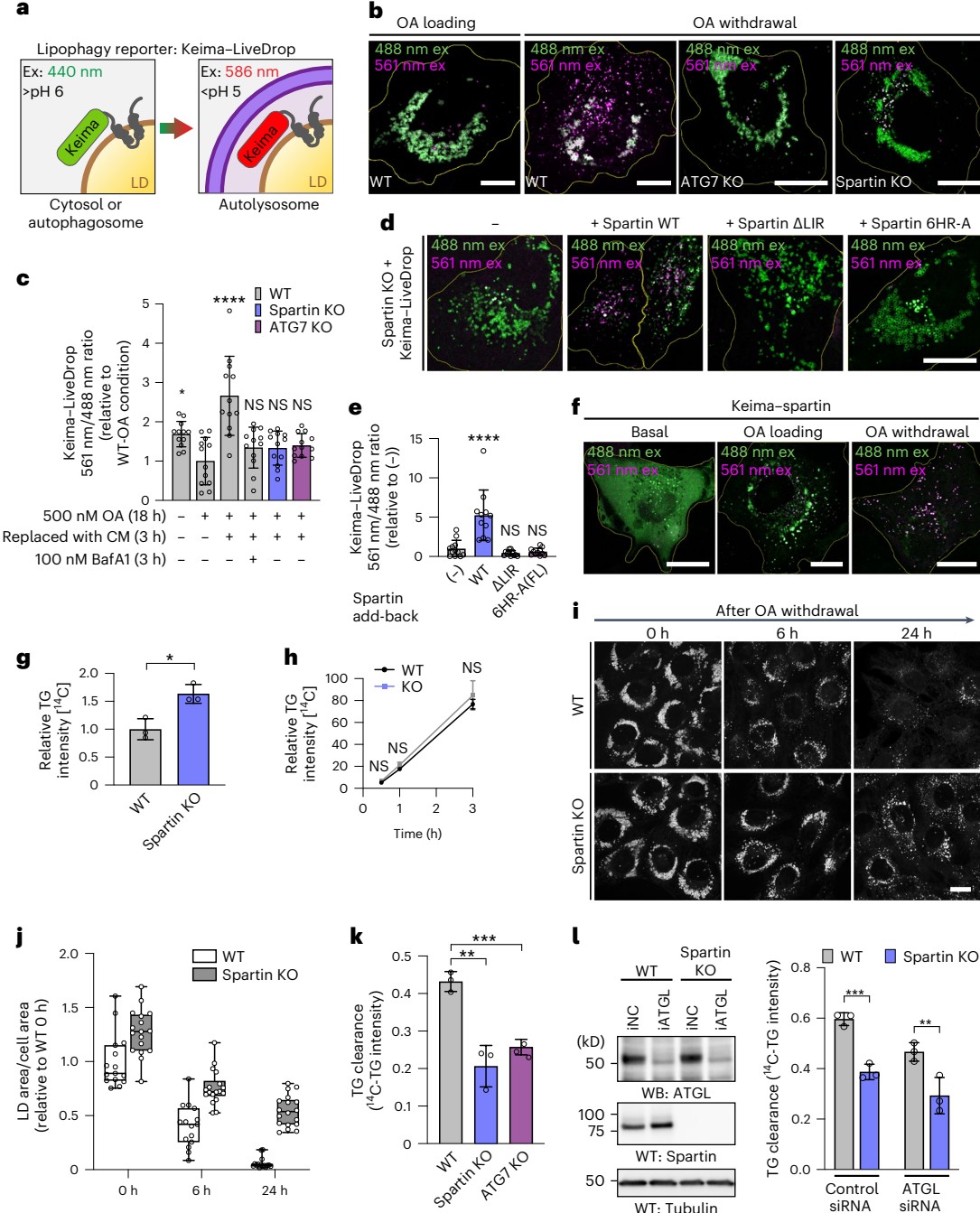

**Fig. 4 | Spartin deficiency causes defects in TG and LD turnover in cells.**
**a**, Schematic illustration for Keima–*LiveDrop* showing excitation spectrum conversion of Keima in various conditions. **b**, Overlay images of Keima–*LiveDrop*, expressed in SUM159 WT, ATG7 KO and spartin KO cells. Scale bars, 10 μm. **c**, Ratiometric fluorescence measurements of Keima–*LiveDrop* in SUM159 WT, ATG7 KO and spartin KO in various conditions. Mean ± s.d., $n$ = 12 cells from three independent experiments, *$P$ = 0.0223; ****$P$ < 0.0001, one-way analysis of variance (ANOVA), Dunnett's multiple comparisons test. **d**, Overlay images of Keima–*LiveDrop* expressed in spartin KO, transiently transfected with Halo–spartin constructs as described. Scale bars, 20 μm. **e**, Ratiometric fluorescence measurements of Keima–*LiveDrop* in spartin KO with transient expression of Halo–spartin constructs. ****$P$ < 0.0001. one-way ANOVA, Dunnett's multiple comparisons test. Mean ± s.d., $n$ = (left to right) 12, 11, 12 and 11 cells, three independent experiments. **f**, Overlay images of Keima–spartin expressed in WT cells. **g,h**, WT and spartin KO SUM159 cells pulse-labelled with [14C]-OA; incorporation into TG was measured after 0.5 mM OA treatment for 24 h (**g**), 0.5–3 h (**h**). Values calculated relative to WT (**d**) and spartin KO cells' highest

value at 3 h. Mean ± s.d., $n$ = 3 independent experiments, *$P$ = 0.0121; NS, not significant; two-tailed unpaired $t$-test (**d**) and two-way ANOVA with repeated measurements (**e**). **i**, Confocal imaging showing LDs (BODIPY493/503) in WT or spartin KO cells reveals impaired LD turnover. Cells were treated with 0.5 mM OA for 24 h and chased for 6 or 24 h after OA withdrawal. Scale bar, 20 μm. **j**, Area of LDs stained by BODIPY493/503 quantified from images shown in **g**. Median ± the 25th to 75th percentiles, the whiskers extended to the minima and the maxima, $n$ = 15 cells (WT; 0 h, 6 h and 24 h), 17 cells (spartin KO; 0 h), 18 cells (spartin KO; 6 h, 24 h), three independent experiments. **k**, Reduced TG degradation in spartin KO cells. WT, spartin KO or ATG7 KO SUM159 cells pulse-labelled with [14C]-OA, and incorporation into TG measured after treatment with 0.5 mM OA for 24 h and subsequent 3 h OA withdrawal. **l**, Reduced TG clearance is independent of ATGL. ATGL knockdown in WT and spartin KO cells for 48 h before [14C]-OA labelling is shown in the left panel. Mean ± SD, $n$ = 3 independent experiments, **$P$ = 0.0022 and ***$P$ = 0.0006, one-way ANOVA, Dunnett's multiple comparisons test. Source numerical data and unprocessed blots are available in source data. Excitation (ex); emission (em); complete medium (CM).

association with LDs (Fig. 1f,g). Analysing the sequence of the senescence domain for potential regions that may mediate its association with LDs, we detected a series of amphipathic helices (AHs) (Fig. 1e, bottom). AHs are often found in cytosolic proteins that interact with the LD surface, where they integrate into packing defects in the phospholipid monolayer of LDs[25–27]. Spartin contains up to 12 sequence stretches with predicted propensity to form 3–11 AHs (11 amino acids per three turns) (Fig. 1e,h), similar to those found in other LD-binding proteins, such as perilipins and α-synuclein[25,28,29]. In particular, the Phyre2 structure prediction revealed that spartin sequence of amino acids 427–517 shows high sequence and structural similarities to LD targeting AH repeats in α-synuclein (amino acids 4–93) (refs. 28,30). Analysing these sequences in expression studies, we found that the first extended AH (AH1, containing three 3–11 repeats, amino acids 431–463) was sufficient to mediate LD binding, and its localization to LDs was enhanced if the second extended AH2 (amino acids 464–503) was included (Fig. 1h–j). In agreement with the importance of AH1 and AH2 in mediating LD binding, mutating six hydrophobic residues across AH1 and AH2 to alanines (in a construct containing only the AH repeats or in the full-length protein) abolished association of the expressed protein with LDs (Fig. 1h–j). Consistent with this finding, a spartin mutation found in patients with Troyer syndrome that results in a frameshift at Lys370 (1110delA mutant; no protein detection in cell lysates from the patients[31])[1] and deletes the AH regions, abolished spartin recruitment to LDs (Extended Data Fig. 1k and ref. 14). Thus, the endogenous spartin protein localizes to LDs probably via interactions of its AH repeats with the LD surface.

To test the hypothesis that LDs decorated by spartin are intermediates of autophagic degradation, we analysed the localization of these LDs with respect to lysosomes. We found that spartin-decorated LDs co-localized with the lysosomal marker LAMP1 and the association between them was enhanced during OA withdrawal (Fig. 2a,b), consistent with a function for spartin in autophagic degradation of LDs. In addition, deletion of the spartin gene reduced the co-localization of lysosomes and LDs (labelled by *LiveDrop*), consistent with spartin's requirement for localizing a subset of LDs to lysosomes (Fig. 2c,d and Extended Data Fig. 2a,b).

If spartin acts as a LD receptor for lipophagy, it needs to interact not only with LDs but also specifically with the autophagy machinery. To test for this, we co-expressed HA-tagged spartin with 3xFLAG-tagged ATG8-type proteins (LC3A, LC3B, LC3C, GABARAP, GABARAPL1 or GABARAPL2) and analysed potential interactions by assaying for co-immunoprecipitation of the proteins. In these assays, we detected an interaction specifically between spartin and LC3A or LC3C (Fig. 2e). An interaction with LC3 was also found when endogenous spartin was immunoprecipitated (Extended Data Fig. 2c). Consistent with this finding, spartin-coated LDs co-localized in cells with fluorescently tagged LC3A (Fig. 2f). Moreover, the co-localization of LDs and lysosomes requires core autophagy machinery, as ATG5, ATG7 or FIP200 were needed for the association of LDs and lysosomes (Fig. 2g,h and Extended Data Fig. 2d).

To map the domain of spartin that is required for the interaction with LC3A, we performed GST-pulldown analyses of recombinant LC3A with HEK293T cell lysates expressing full-length spartin or various spartin truncation mutants (Extended Data Fig. 3a,b). Spartin lacking the UBR domain greatly reduced the interaction between spartin and recombinant LC3A (Fig. 3a and Extended Data Fig. 3a,b). To corroborate these results, we purified recombinantly produced LC3s and the UBR domain of spartin to assay for direct interactions between the proteins. By performing a pulldown assay with LC3s, we found LC3A and LC3C directly interacted with the UBR domain (Fig. 3b). Microscopy data corroborated these findings. The co-localization of spartin-marked LDs with lysosomes depended on spartin's UBR domain (Fig. 3c,d). In contrast, neither the deletion of spartin's MIT domain nor the PPAY motif affected the co-localization of spartin and LAMP1 (Extended Data Fig. 3c,d). These findings suggest that neither the MIT-domain-mediated ESCRT-III interaction of spartin[21] nor ubiquitination[14,18] are required for spartin's function in lipophagy.

Lipophagy receptors interact with LC3 proteins via LC3-interacting region (LIR) consensus motifs[32,33]. To determine how spartin interacts with LC3A, we utilized AlphaFold[34,35]-based algorithms to predict the interaction interface between spartin's UBR domain and LC3 (Fig. 3e). A putative LIR motif was identified (aa 194–197 with stretches of acidic adjacent residues), and deletion of residues 193–200 markedly diminished the interaction of recombinant LC3A with the spartin UBR domain (Fig. 3f and Extended Data Fig. 4e). Supporting this result, deletion of the LIR motif abolished co-localization of LC3A and spartin in cells (Fig. 3g,h).

Collectively, our results suggest that spartin interacts through AHs (in its senescence domain) with LDs and recruits core autophagy machinery via its LIR motif (within the UBR domain) to deliver the organelle to lysosomes. To test this model, we developed a lipophagy

**Fig. 5 | Interfering with spartin function leads to TG and LD accumulation in cultured human neurons or murine brain neurons. a**, Generation of spartin KO–iPS cell lines. **b**, Confocal imaging of fixed iMNs showing LD accumulation in spartin KO–iMN compared with parental cell line (day 12 post-differentiation). Cells were incubated with 100 nM OA for 24 h. Scale bar, 10 μm. **c**, Quantification of **b**. Mean ± s.d., $n = 10$ cells (Basal; WT, Spartin KO1, Spartin KO2), 12 cells (100 nM; WT and Spartin KO2) and 11 cells (100 nM Spartin KO1), three independent experiments, ***$P = 0.006$ (for comparison of WT and Spartin KO1); ***$P = 0.007$ (for comparison of WT and Spartin KO1), one-way analysis of variance, Dunnett's multiple comparisons test. **d**, AAV constructs to express mScarlet-I or mScarlet-I–spartin FL as controls or a dominant-negative form of spartin (spartin-DN) under a pan-neuronal *synapsin* promoter (*Syn*). **e**, Expression of mScarlet-I and mScarlet-I-fused spartin-DN in the mouse motor cortex. AAV–mSc-I or AAV–mSc-I–spartin-DN was stereotaxically injected into different hemispheres of the M1 motor cortex in 7–8-week-old WT mice. Scale bar, 100 μm. **f,g**, Accumulation of BODIPY493/503 in spartin-DN-expressing neurons of the mouse M1 motor cortex. Representative images of the AAV-infected M1 cortex slices stained with BODIPY493/503 10–11 days after AAV injection (**f**). Scale bar, 10 μm. Quantification of LD numbers of the cell bodies of M1 cortex slices (**g**). Mean ± s.d., $n = 9$ fields of view from $n = 3$ mice, ****$P < 0.0001$, two-tailed unpaired *t*-test. **h,i**, Lipidomic profiles of the AAV-infected M1 cortices show increased amounts of TG and DAG in neurons where spartin function was disrupted. Lipids were extracted from tissues and analysed by liquid chromatography–mass spectrometry (LC–MS) as described in Methods. Relative fold-changes are shown in (**h**). Triglyceride (TG); diacylglycerol (DAG); cholesterol ester (CE); ceramide (Cer); hexosylceramide (HexCer); sphingomyelin (SM); phosphatidylcholine (PC); ether-linked phosphatidylcholine (PC-O); phosphatidylethanolamine (PE); phosphatidylethanolamine plasmalogens (PE-P); ether-linked phosphatidylethanolamine (PE-O); phosphatidylinositol (PI); phosphatidylserine (PS); phosphatidic acid (PA); phosphatidylglycerol (PG); lysophosphatidylcholine (LPC); lysophosphatidylethanolamine (LPE); lysophosphatidylinositol (LPI); fatty acid (FA). LC–MS analysis verified widespread elevations in TGs from spartin-DN-expressing neurons of the M1 cortex (Spartin-DN), compared with a control (mSc-I) (**i**). Box-and-whisker plot, median ± the 25th to 75th percentiles, the whiskers extended to the minima and the maxima, $n = 6$ mice, **$P < 0.01$, ***$P < 0.001$ (**h**, TAG = 0.000506, DAG = 0.000081; **i**, 50:0 = 0.000782, 50:1 = 0.000688, 52:1 = 0.000452, 52:2 = 0.000516, 52:6 = 0.000860, 54:4 = 0.000849, 54:6 = 0.000463, 56:5 = 0.000738, 56:6 = 0.000444, 56:7 = 0.000393, 58:7 = 0.000754, 50:2 = 0.002913, 52:0 = 0.000684, 52:3 = 0.002428, 52:4 = 0.000621, 52:5 = 0.000911, 53:0 = 0.002413, 53:1 = 0.001690, 54:3 = 0.000166, 54:7 = 0.001090, 56:1 = 0.001576, 65:4 = 0.000350, 56:8 = 0.000765, 58:5 = 0.001099, 58:6 = 0.000570, 58:8 = 0.000339, 60:7 = 0.001566, 60:8 = 0.001432); two-tailed unpaired *t*-test in each row; multiple comparisons test using the two-stage step-up method of Benjamini, Krieger and Yekutieli. Source numerical data and unprocessed blots are available in source data.

reporter system based on the Keima-fluorophore. This fluorescent protein is sensitive to pH and shifts its excitation spectrum upon trafficking to the lysosome (pH of ~4.5) (ref. 36). We fused Keima to the N-terminus of *LiveDrop* to generate a fluorescence reporter that selectively shifts fluorescence depending on whether LDs are in the cytoplasm or the acidic lysosome (Fig. 4a). To assess the possibility that spartin affected catabolism of LDs, we developed an assay for lipophagy in which we first incubated SUM159 cells in medium containing fatty acids, resulting in the formation of abundant LDs, and then withdrew lipids to stimulate LD turnover. This withdrawal of OA from the cell-culture medium led to a shift in Keima fluorescence consistent with LDs being engulfed

by lysosomes (Fig. 4b, first left two panels, Fig. 4c). The transition of *LiveDrop* Keima fluorescence due to lysosomal localization depended on an intact autophagy pathway as it was abolished in ATG7 knockout (KO) cells (Fig. 4b, third left panel, Fig. 4c). Similarly, deletion of spartin also prevented this change in Keima emission spectrum, indicating spartin is required for lipophagy of this reporter (Fig. 4b, right-most panel, Fig. 4c). Moreover, expressing spartin mutants disrupting its LD binding (6HR-A: mutating six hydrophobic residues across AH1 and AH2 to alanines) or its binding to LC3 (ΔLIR) in spartin KO cells did not restore defects in lipophagic activity, as measured by the Keima–*Live-Drop* assay (Fig. 4d,e) and by LD accumulation (Extended Data Fig. 3f,g).

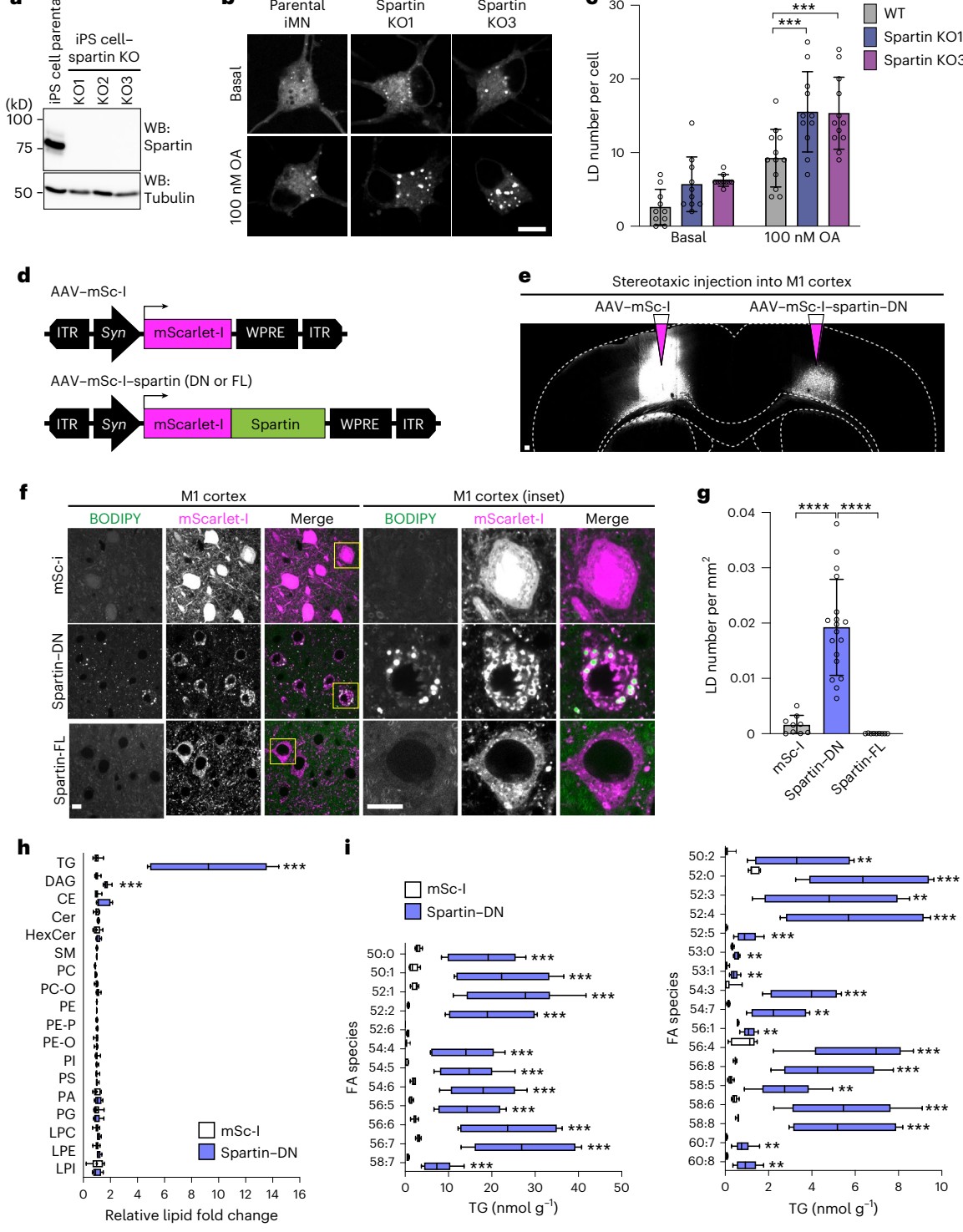

As a further test of the model that spartin mediates deliver of LDs to the lysosome for autophagy, we also developed an assay to monitor spartin trafficking. We expressed a Keima–spartin fusion protein in cells and found that it localized to an acidic compartment during LD mobilization after withdrawal of OA (Fig. 4f). Consistent with Keima–spartin result, we found that spartin is degraded by autophagy (Extended Data Fig. 4a).

Previous reports suggested that spartin is functionally connected to the ubiquitin pathway[14,17,18]. Spartin is ubiquitinated by WW domain-containing E3 ubiquitin ligase 1 (WWP1) at its PPAY motif and interacts with ubiquitin via its UBR domain[14]. To exclude that spartin recruits the autophagy machinery through ubiquitin-mediated protein interactions, we examined lipophagy after blocking ubiquitination by using TAK-243, an inhibitor of the primary mammalian E1 enzyme mediating the ubiquitin cascade[37] (Extended Data Fig. 4b). Treatment with TAK-243 efficiently inhibited ubiquitination, yet the association of LDs and LAMP1 was not affected (Extended Data Fig. 4c,d). Furthermore, depletion of known selective autophagy receptors (OPTN1, NBR1 and SQSTM1/p62) that require cargo-ubiquitin binding did not affect lipophagy flux as measured by the Keima–*LiveDrop* assay (Extended Data Fig. 4e,f). We also confirmed that spartin is not part of the general autophagy machinery. We found no differences between wild-type (WT) and spartin KO cells in assays monitoring LC3 lipidation (conversion of LC3-I to LC3-II) or autophagic flux (Keima–LC3B reporter) during nutrient starvation (Extended Data Fig. 5a,b).

Spartin-mediated degradation of LDs by selective autophagy also predicts an increase of stored lipids in cells lacking spartin. To test this, we measured TG synthesis, accumulation and turnover in WT and spartin KO cultured cells. After 24 h of oleate addition, spartin KO cells accumulated more TG than WT cells (Fig. 4g). We found no differences between WT and spartin KO cells in cellular TG synthesis assays (Fig. 4h) or in short-term (30 min) LD formation assays (Extended Data Fig. 5c,d), indicating that TG accumulation in spartin KO cells was not likely due to altered rates of LD biogenesis. In contrast, visualizing LDs by fluorescence microscopy after oleate withdrawal revealed that spartin KO cells had impaired LD degradation, with many more and larger LDs remaining after 6 h and particularly 24 h of fatty acid removal from the cell-culture medium (Fig. 4i,j). Additionally, pulse-chase assays using radio-labelled fatty acids to trace degradation of TG showed that spartin KO cells had ~48% lower rates of TG degradation during 3 h after OA withdrawal (Fig. 4k). A similar impairment of TG degradation was found in cells lacking a core component of the autophagy machinery (ATG7) (Fig. 4k). The degree of impairment in TG degradation for spartin KO cells was comparable (35–40% reduction) in WT cells and cells depleted for the major TG lipase ATGL (Fig. 4l), indicating that the TG degradation defect was independent of lipolysis. These data are consistent with a model in which, under conditions of lipid withdrawal, spartin deficiency impairs LD turnover by lipophagy. Notably, depletion of other medium nutrients was also assessed with the Keima–spartin flux assay. These perturbations also induced spartin-mediated lipophagy, but, of the conditions tested, spartin-mediated lipophagy showed the highest activity with lipid deprivation (Extended Data Fig. 6).

In humans, loss of spartin function due to *SPART* mutations leads to Troyer syndrome, a complex hereditary spastic paraplegia with degeneration of motor neurons[1,38]. Our findings led us to consider whether impaired spartin function in neurons compromises lipophagy in this cell type. Indeed, lipophagy has been reported to occur in neurons[39,40]. Although neurons normally do not contain large amounts of TGs or LDs, deficiency of the neuronal TG lipase DDHD2/Spg54, which is also associated with development of hereditary spastic paraplegia, causes TG accumulation in neurons[8,41], providing evidence that these cells can synthesize and degrade TGs.

Immunohistochemistry staining of endogenous spartin revealed that spartin is ubiquitously expressed in murine brain and widely detectable in neurons (Extended Data Fig. 7). To test whether spartin functions in neuronal LD biology, we generated two distinct induced pluripotent stem (iPS) cell lines with spartin deleted (Fig. 5a) and differentiated these and parental control cells into motor neurons[42] (Extended Data Fig. 8). The resultant induced motor neurons lacking spartin accumulated more BODIPY-stained LDs than controls, a phenotype that was exacerbated with the inclusion of OA in the culture medium (Fig. 5b,c).

To enable testing how impairing spartin function affects LDs in neurons in vivo, we identified a dominant-negative C-terminal fragment of spartin (amino acids 380–666, spartin-DN) that binds to LDs but does not engage the autophagic machinery. The expression of spartin-DN in cultured cells triggered accumulation of LDs (Extended Data Fig. 9a,b) without affecting TG synthesis (Extended Data Fig. 9c). Selective inhibition of lipophagy by spartin-DN was further validated by showing that its expression reduced the levels of Keima–*LiveDrop* (Extended Data Fig. 9d,e) and Keima–spartin (Extended Data Fig. 9f,g) in acidic compartments, and reduced co-localization of spartin-DN-decorated LDs with LAMP1 after fatty acid withdrawal (Extended Data Fig. 9h). We next injected into one hemisphere of the mouse M1 motor cortex an adeno-associated virus (AAV) to express mScarlet-I–spartin-DN under the *synapsin* promoter, leading to interference with spartin function in one hemisphere (Fig. 5d,e). As a control, we injected AAVs to express either mScarlet-I or mScarlet-I–spartin-full-length (FL) into the contralateral hemisphere (Fig. 5d,e). Fluorescence microscopy showed that the AAVs resulted in robust expression of mScarlet-I or mScarlet-I–spartin-DN in neurons (Fig. 5e). Co-staining of these neurons with the LD probe BODIPY493/503 showed a marked increase of LDs specifically in cells where spartin function was inhibited (Fig. 5f,g). Consistent with an increase in LDs, spartin-DN-expressing cortical neurons showed approximately ten-fold more total TGs, with a marked increase across the detected TG species, than mScarlet-I controls (Fig. 5h,i, Extended Data Fig. 10 and Supplementary Tables 3 and 4). The levels of DAG were also approximately two-fold greater, and sphingolipid and phospholipid levels were similar in the two regions (Fig. 5h, Extended Data Fig. 10 and Supplementary Tables 3 and 4). These data indicate that spartin has an important function in mediating TG turnover in neurons of the mouse motor cortex.

Taken together, our data indicate that spartin links LDs to the autophagy machinery as a lipophagy receptor. This model is supported by spartin's ability to bind LDs as well as LC3A/C, core components of the autophagy machinery, and is consistent with the effect of spartin deficiency in cultured neurons and the brain, where it leads to accumulation of LDs and TGs. Our data are most consistent with spartin functioning in macroautophagy of LDs, yet some studies have linked LD clearance to microautophagy at the lysosome[12,43,44], and it remains possible that spartin acts in a process more akin to microautophagy.

Whether additional lipophagy receptors exist for different cell types or different metabolic conditions is unknown and remains an area of active investigation[45]. It is also unclear whether spartin is important in different cell types. The current findings show, however, that the spartin-mediated lipophagy pathway may be particularly important in neurons given the connection of spartin with hereditary spastic paraplegia. Inasmuch as deficiency of either spartin or DDHD2 (ref. 7) results in neuronal TG accumulation by different mechanisms, and both lead to hereditary spastic paraplegia, the available data suggest that impaired TG turnover in neurons may be directly linked to neurodegeneration in humans.

## Online content

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

## Methods

This study complies with all relevant ethical regulations that are approved by the Harvard T. H. Chan School of Public Health Institutional Review Board and Institutional Animal Care and Use Committee (approval number IACUC-18-11-0836).

### Cell culture

SUM159 breast cancer cells were obtained from the laboratory of Tomas Kirchhausen (Harvard Medical School) and were maintained in Dulbecco's modified Eagle medium (DMEM)/F12 GlutaMAX (Life Technologies) supplemented with 5 µg ml$^{-1}$ insulin (Cell Applications), 1 µg ml$^{-1}$ hydrocortisone (Sigma), 5% foetal bovine serum (Life Technologies 10082147; Thermo Fisher), 50 µg ml$^{-1}$ streptomycin and 50 U ml$^{-1}$ penicillin. Where noted, cells were incubated with medium containing 0.5 mM OA complexed with essentially fatty-acid-free bovine serum albumin (BSA). For immunoprecipitation analyses, HEK293T cells (ATCC) were used. HEK293T cells were cultured in DMEM (Gibco) supplemented with 10% foetal bovine serum (Life Technologies), 50 µg ml$^{-1}$ streptomycin and 50 U ml$^{-1}$ penicillin.

### Special reagents and antibodies

Janelia Fluor dyes with HaloTag (JF549 and JF646) (ref. [46]) were kind gifts of Luke Lavis (Janelia Research Campus). BODIPY493/503, HCS Lipid TOX Deep Red Neutral Lipid Stain and puromycin were purchased from Thermo Fisher Scientific. OA was purchased from Millipore-Sigma. OA [1-$^{14}$C] was purchased from American Radiolabeled Chemicals.

Primary antibodies used in this study were: rabbit polyclonal anti-SPG20/spartin (Proteintech), rabbit polyclonal anti-FIP200/RB1CC1 (Proteintech), mouse monoclonal anti-FLAG (Millipore-Sigma), anti-FLAG® M2 affinity gel (Millipore-Sigma), rat monoclonal anti-HA clone 3F10 (Millipore-Sigma), mouse monoclonal anti-α-tubulin (Millipore-Sigma), rabbit polyclonal anti-mCherry (for detection of mScarlet-I, Abcam), rabbit polyclonal anti-LC3A/B (Cell Signaling Technology), mouse monoclonal anti-actin (Cell Signaling Technology), rabbit polyclonal anti-ATGL (Cell Signaling Technology), rabbit monoclonal anti-ATG5 (Cell Signaling Technology), rabbit monoclonal anti-ATG7 (Cell Signaling Technology), chicken polyclonal anti-MAP2 (Synaptic Systems), rat polyclonal anti-GFAP (Thermo Fisher Scientific), rabbit polyclonal anti-GST (Thermo Fisher Scientific), anti-NBR1 (Proteintech), anti-OPTN (Proteintech) and anti-SQSTM1 clone 2C11 (Novus). All primary antibodies were used in 1:1,000 dilution. HRP-conjugated secondary antibodies against mouse and rabbit were from Santa Cruz Biotechnology. Fluorescent secondary antibodies used in this study were purchased from Thermo Fisher Scientific and included DyLight 488-conjugated goat anti-chicken IgY (H + L) cross-adsorbed secondary antibody, Alexa Fluor 488-conjugated donkey anti-rat IgG (H + L) highly cross-adsorbed secondary antibody, and Alexa Fluor 594-conjugated goat anti-rabbit IgG (H + L) highly cross-adsorbed secondary antibody.

A 10 mM OA stock solution was made in 3 mM fatty-acid-free BSA (Millipore-Sigma)–phosphate-buffered saline (PBS). The solution was incubated in 37 °C shaking incubator for 1 h to completely dissolve OA in the 3 mM BSA–PBS. The stock solution was filtered and stored at −20 °C.

### Plasmid construction

The following plasmids were kind gifts: pCMV 3XFLAG–LC3A WT (Addgene plasmid #123089), pCMV 3XFLAG–LC3B WT (Addgene plasmid #123092), pCMV 3XFLAG–LC3C WT (Addgene plasmid #123095), pCMV 3XFLAG–GABARAP WT (Addgene plasmid #123097), pCMV 3XFLAG–GABARAPL1 WT (Addgene plasmid #123100), pCMV 3XFLAG–GABARAPL2 WT (Addgene plasmid #123103) from Robin Ketteler, pSpCas9(BB)-2A-Puro (PX459) V2.0 (Addgene plasmid #62988) from Feng Zhang, and pmScarlet-I_C1 (Addgene plasmids #85044) from Dorus Gadella. pEGFP–N1 and pEGFP–C1 plasmids were purchased

from Clontech Laboratories, pSMART-HC-Amp plasmid was purchased from Lucigen. pET28-12XHis-SUMO plasmid was from Xudong Wu.

For AAV constructs, ENTR clones were first generated by inserting polymerase chain reaction (PCR) fragments encoding either mScarlet-I or mScarlet-I–spartin (380–666 amino acids) using AgeI and KpnI sites. Both PCR fragments were generated from pCMV–mScarlet-I–spartin with primer sets: 5′-GCACAACCGGTGCCACCATGGTGAGCAAGG-3′ and 5′-CGGGGTACCTCACTTGTACAGCT CGTCCATGCC-3′ (for mScarlet-I), 5′-GCACAACCGGTGCCACCATGGTGAGCAAGG-3′ and 5′-CGGGGTACCTCATTTATCTTTCTTCTTTGCCTCCTTTACTTCCT-3′ (for mScarlet-I–spartin (amino acids 380–666)). The AAV constructs were then generated by Gateway LR recombination, using a destination vector of the modified pAAV-MCS[47] containing a synapsin promoter and a woodchuck hepatitis virus post-transcriptional regulatory element.

For other plasmid construction, all PCRs were performed using PfuUltra II Fusion HotStart DNA Polymerase (Agilent Technologies), and restriction enzymes were from New England Biolabs. The synthetic DNAs (gBlock, Integrated DNA Technologies) that were used in this study and cloning strategies of the other plasmids (including primer information) are summarized in Supplementary Tables 1 and 2.

### Generation of KI and KO cells with CRISPR/Cas9-mediated genome editing

A spartin-KO SUM159 cell line was generated by CRISPR/Cas9 gene editing from a WT background[48]. The sequence 5′-CTCTACAGAATG TACGCACC-3′ was used as single-guide RNA (sgRNA) to direct Cas9 into the exon 2 of the *SPG20/spartin* locus. Cells were selected with 1.5 µg ml$^{-1}$ puromycin for 48 h. Genomic DNA of clones showing depletion of spartin protein by immunoblot analysis with the spartin antibody were extracted (DNeasy Blood and Tissue Kit, Qiagen), and the genomic DNA sequence surrounding the target exon of *spartin* was amplified by PCR (sense: 5′-AAATGGAGCAAGAGCCACAAATGGAG-3′, antisense: 5′-GAGGAGCTTCT GCTGGACAACTTTGTG-3′). PCR products were subcloned into a plasmid (Zero Blunt TOPO PCR Cloning Kit, Thermo Fisher Scientific) to validate the edited region of positive KO clones by sequencing.

To generate N-terminally mScarlet-I-tagged spartin and N-terminally mScarlet-I-tagged PLIN3, SUM159 cells were transfected by FuGENE HD transfection reagent (Promega Corporation) with an individual donor plasmid containing arms with ~800-nucleotide-long homology upstream and downstream of the target site and a sgRNA targeting downstream of start codon (for spartin, 5′-CTCTACAGAATGTACGCACC-3′), and downstream of start codon (for PLIN3, 5′-AGAGACCAT GTCTGCCGACG-3′), respectively. The homology arm sequence information of individual donor plasmid (pSMART-mScarlet-I–spartin and pSMART-mScarlet-I–PLIN3) are described in Supplementary Tables 1 and 2. Cells were selected with 1.5 µg ml$^{-1}$ puromycin for 48 h, and single-cell fluorescence-activated cell sorting was performed (Harvard, Division of Immunology). To validate the insertion of tags, target regions were amplified by PCR and sequenced.

The double KI SUM159 cell line (HaloTag–KI–spartin and mScarlet-I–KI–PLIN3) was generated by sequential clonal generation. The mScarlet-I–KI–PLIN3 cell line was initially generated and used for the generation of double KI cell line. The donor plasmid (pSMART-HaloTag–spartin) is described in Supplementary Tables 1 and 2.

### Transfection and RNA interference

Transfection of plasmids into SUM159 cells was performed with FuGENE HD transfection reagent (Promega Corporation) ~24 h before imaging.

Specific knockdown of spartin in SUM159 cells was performed by transfection of small interfering RNA (siRNA) duplexes by Lipofectamine RNAiMAX (Life Technologies). The siRNAs for negative control (#D-001220-01), ATGL/PNPLA2 (#L-009003-01-0005), PLIN3 (#L-015979-00-0005), NBR1 (#L-010522-00-0005), SQSTM1/p62

(#L-010230-00-0005) and OPTN (#L-016269-00-0005) were purchased from Dharmacon.

## Expression and purification of GST-fusion proteins and UBR–HA
GST-fusion proteins and 12xHis-SUMO-UBR–HA were expressed in NiCo21(DE3) competent *Escherichia coli* cells (New England Biolabs). Expression was induced with 0.5 mM IPTG in a 1 litre culture of liquid broth (supplemented with ampicillin) for 20 h at 18 °C. Cells were collected by centrifugation.

GST-fusion protein expressing cells were resuspended in buffer A (50 mM Tris Cl, pH 8.0, 500 mM NaCl, 0.5 mM dithiothreitol (DTT) and cOmplete Protease Inhibitor Cocktail tablet, EDTA-free (Millipore-Sigma)) and lysed by sonication. The cell debris was removed by centrifugation at 5,000*g* for 15 min. The supernatant was incubated with 1 ml of glutathione Sepharose 4B (Cytiva) for 2 h at 4 °C. The resins were collected and washed with 30 ml of buffer A and once with 10 ml of buffer (50 mM Tris Cl, pH 8.0, and 150 mM NaCl). GST-fusion proteins were eluted with 2 ml of elution buffer (50 mM Tris Cl, pH 8.0, 150 mM NaCl and 10 mM reduced glutathione).

Cells expressing 12xHis-SUMO-UBR–HA were resuspended in buffer (50 mM Tris Cl, pH 8.0, 500 mM NaCl, 10 mM imidazole, 0.5 mM DTT and cOmplete Protease Inhibitor Cocktail tablet, EDTA-free (Millipore-Sigma)). Cells were lysed by sonication. The cell debris was removed by centrifugation at 5,000*g* for 15 min. The supernatant was incubated with 1 ml of PureCube 100 INDIGO Ni-Agarose (Cube Biotech) for 2 h at 4 °C. The resins were then collected and washed with 30 ml of buffer (50 mM Tris Cl, pH 8.0, 300 mM NaCl and 25 mM imidazole) and eluted with buffer (50 mM Tris Cl, pH 8.0, 300 mM NaCl and 500 mM imidazole). Once imidazole concentration was reduced to 10 mM by sample concentration and re-dilution in buffer (50 mM Tris Cl, pH 8.0, and 300 mM NaCl), 12xHis-SUMO tag was removed by SUMO protease (a gift from X. Wu) for overnight at 4 °C.

The eluted proteins were concentrated and further purified by size-exclusion chromatography on a Superdex200 column, equilibrated with buffer containing 50 mM Tris Cl, pH 8.0, 150 mM NaCl and 1 mM DTT. Peak fractions were pooled and concentrated for in vitro assay.

## Immunoprecipitation and immunoblotting
For protein-level analyses (Extended Data Figs. 2b–d and 5), cells were lysed in 1% SDS lysis buffer (50 mM Tris Cl, pH 8.0, 150 mM NaCl, 1% SDS and cOmplete Protease Inhibitor Cocktail tablet, EDTA-free (Millipore-Sigma)) with ~100 units of Benzonase Nuclease (Millipore-Sigma). After protein concentrations were determined using Pierce BCA Protein Assay Kit (Thermo Fisher Scientific), cell lysates were mixed with Laemmli sample buffer and heated for 10 min at 75 °C before SDS–PAGE.

For immunoprecipitation in Fig. 2d, cells were lysed in 500 µl of lysis buffer (50 mM Tris Cl, pH 8, 150 mM NaCl, 1 mM EDTA, 1% Triton X-100, PhosSTOP (Millipore-Sigma) and cOmplete Protease Inhibitor Cocktail tablet, EDTA-free (Millipore-Sigma)). After rocking for 20 min at 4 °C, the cell lysates were centrifuged at 18,000*g* for 10 min at 4 °C, and the supernatants were collected. The 400 µl of protein lysates were incubated with 20 µl of anti-FLAG M2 affinity gel (Millipore-Sigma) for 2 h at 4 °C. The bead-bound materials were washed with 500 µl of washing buffer (50 mM Tris–HCl, pH 8, 150 mM NaCl, 1 mM EDTA and 1% Triton X-100) three times and were eluted with Laemmli sample buffer for 10 min at 75 °C before SDS–PAGE.

For immunoblot analyses, all gels were transferred to Immuno-Blot polyvinylidene fluoride membranes (Bio-Rad) with 1× Tris/glycine transfer buffer (Bio-Rad) with 20% methanol for 1.5 h at 100 V in a cold room. The membranes were incubated in TBS-T supplemented with 5% non-fat dry milk (Santa Cruz Biotechnology) at room temperature for 20–60 min and subsequently incubated with primary antibodies for overnight in cold room with gentle shaking. Membranes were washed three times in TBS-T for 5 min each and incubated at room temperature for 60 min with appropriate HRP-conjugated secondary antibodies (Santa Cruz Biotechnology) before analysis by chemiluminescence with the SuperSignal West Pico or Dura reagents (Thermo Fisher Scientific).

For Coomassie staining, SDS–PAGE gels were washed with Milli-Q water for 20 min to get rid of residual SDS, then the gels were incubated with colloidal Coomassie staining buffer (10% ethanol, 0.02% Coomassie brilliant blue G-250, 5% aluminium sulfate-(14-18)-hydrate and 2% ortho-phosphoric acid, 85%) for more than 3 h.

## GST-LC3A pulldown with HEK293T cell extracts
The indicated GST-fusion proteins (3 nM) were equilibrated in 500 µl of assay buffer (50 mM Tris Cl, pH 8, 150 mM NaCl, 1% Triton X-100, PhosSTOP (Millipore-Sigma) and cOmplete Protease Inhibitor Cocktail tablet, EDTA-free (Millipore-Sigma)) and mixed with 10 µl of MagneGST Glutathione Particles (Promega) for 1.5 h at 4 °C. Meanwhile, the HEK293T cell lysates transiently expressing mScarlet-I-tagged spartin truncation mutants were lysed in 500 µl of lysis buffer (50 mM Tris Cl, pH 8, 150 mM NaCl, 1 mM EDTA, 1% Triton X-100, PhosSTOP (Millipore-Sigma) and cOmplete Protease Inhibitor Cocktail tablet, EDTA-free (Millipore-Sigma)). After rocking for 20 min at 4 °C, the cell lysates were centrifuged at 18,000*g* for 10 min at 4 °C, and the supernatants were collected. Subsequently, 500 µg of the cell lysates were mixed with GST-conjugated LC3A for 2 h at 4 °C. The resin was washed with wash buffer (50 mM Tris Cl, pH 8, 150 mM NaCl and 1% Triton X-100) three times and were eluted with Laemmli sample buffer for 10 min at 75 °C before SDS–PAGE.

## In vitro binding assay for GST-LC3 and spartin-UBR
The in vitro binding assay was performed by incubating the indicated protein combinations (1 µM of each protein) in 300 µl of assay buffer (25 mM Tris Cl, pH 8, 150 mM NaCl and 5% glycerol) for 2 h at 4 °C. The resin was washed with wash buffer (25 mM Tris Cl, pH 8, 150 mM NaCl and 0.1% Triton X-100) three times and eluted with Laemmli sample buffer for 10 min at 75 °C before SDS–PAGE.

## Animals
C57BL/6J WT mice were obtained from The Jackson Laboratory (stock no. 000664). Mice were maintained at the Division of Laboratory Animal Resources facility of Wayne State University and treated under the guidelines of the Institutional Animal Care and Use Committee of Wayne State University. All mice were housed on a 12 h light–dark cycle with food and water ad libitum (65 ± 75 °F and 40 ± 60% humidity).

## AAV production
AAVs were generated using AAV-DJ Helper Free system (Cell Biolabs) as reported[47]. An AAV construct was transfected with pAAV-DJ and pHelper (Cell Biolabs) into 293FT cells (ThermoFisher). Transfected cells were lysed, and AAVs were purified with HiTrap Heparin HP columns (Cytiva) as described[49]. The titre of each AAV was evaluated by quantitative PCR.

## Virus injection
C57BL/6 mice (7–8 weeks old) were anaesthetized with 1.5–3.0% isoflurane and placed in a stereotaxic apparatus (Kopf Instruments). Subcutaneous injections of meloxicam (5 mg kg⁻¹ body weight) were administered for three consecutive days, beginning one day before the operation. The skull was exposed over the M1 motor cortex based on stereotaxic coordinates. Then, 2 µl of AAV (approximately $6 \times 10^7$ vg) was injected into each hemisphere of the M1 motor cortex using a glass pipette (tip diameter ~5–8 mm) at a rate of 100 nl min⁻¹ using a syringe pump (Micro4, World Precision Instruments). The injection site was standardized among animals by using stereotaxic coordinates (ML, ±1.50; AP, +1.00; DV, −1.50 and −1.25) from bregma. At the end of the injections, we waited at least 10 min before retracting the pipette.

## Histology

WT mice (9–10 weeks old) were deeply anaesthetized and transcardially perfused with 4% paraformaldehyde (PFA) in PBS (pH 7.4). Brains were fixed overnight in 4% PFA, and 50-μm-thick sagittal sections were cut on a vibratome (Leica Biosystems) at 4 °C. Free-floating sections were washed in PBS and then incubated for 45 min at room temperature with 5% normal goat serum and 0.3% Triton X-100 in PBS. Slices were incubated overnight at room temperature with PBS containing 3% normal goat serum, 0.1% Triton X-100 and primary antibodies: rabbit polyclonal anti-spartin (1:500), chicken polyclonal anti-MAP2 (1:1,000) and rat polyclonal anti-GFAP (1:1,000). Sections were then washed three times in PBS and incubated for 2 h at room temperature with PBS containing 3% normal goat serum and secondary antibodies (for example, Alexa Fluor 594-conjugated goat anti-rabbit IgG, DyLight 488-conjugated goat anti-chicken IgY and Alexa Fluor 488-conjugated donkey anti-rat IgG). For BODIPY493/503 staining, AAV-injected brains were fixed overnight in 4% PFA and cut into 50-μm-thick coronal sections on a vibratome at 4 °C. Free-floating sections were washed three times in PBS and then incubated for 10 min at room temperature with 0.3 μM BODIPY493/503 in PBS. After staining, sections were washed three times in PBS and mounted on microscope slides with Vectashield antifade mounting medium (Vector Laboratories). Confocal fluorescence images were acquired on a laser scanning confocal microscope (LSM 780; Zeiss) equipped with a 10× (numerical aperture (NA) 0.3) objective or a 63× (NA 1.4) objective.

## Lipidomic profiling of mouse brain tissues

At 12 days after AAV injection, mScarlet-I-positive regions of the M1 motor cortex were dissected in ice-cold PBS under an epifluorescent microscope (CKX53, Olympus). Dissected tissues were immediately frozen in liquid nitrogen and stored at −80 °C until lipid extraction. Excised brain tissues were then homogenized in ice-cold nuclease-free water using Bead Mill homogenizer (VWR). Lipids were extracted from 7.5 mg of tissue homogenate containing 8 μl of SPLASH LIPIDOMIX mass spectrometry standard (Avanti Polar Lipids) with the addition of methyl *tert*-butyl ether (Sigma Aldrich) and methanol (Sigma Aldrich) under 7:2:1.5 (v:v) mixing ratio of methyl *tert*-butyl ether, methanol and tissue homogenate, respectively. Samples were mixed and incubated on a thermo shaker for 60 min at 4 °C with 1,000 r.p.m. agitating speed. After extraction, samples were centrifuged at 10,000g for 5 min at 4 °C. Organic upper layer was transferred into a fresh microfuge tube and dried using vacuum concentrator (Eppendorf).

Lipids were separated using the Thermo Acclaim C30 reverse-phase column (inner diameter 2.1 × 250 mm, 3 μM pore size) (Thermo Fisher Scientific) connected to a Dionex UltiMate 3000 UHPLC system and a Q-Exactive orbitrap mass spectrometer (Thermo Fisher Scientific). Extracted lipid samples were dissolved in 150 μl of solvent containing chloroform/methanol (2:1 v:v). Ten microlitres of lipid was injected for analysis positive and negative ionization modes, respectively. UHPLC solvents consist of stationary phase 60:40 water/acetonitrile (v:v), 10 mM ammonium formate and 0.1% formic acid, and mobile phase 90:10 isopropanol/acetonitrile (v:v), 10 mM ammonium formate and 0.1% (v:v) formic acid. Lipids were separated over 90 min gradient at 55 °C with chromatographic flow rate of 0.2 ml min⁻¹. Mass spectrometer data were converted to mzML format using MSConvert[50] before analysis using LipidXplorer version 1.2.8.1 (ref. [51]), and duplicate lipids were excluded. Each lipid class was normalized by total phosphotidylcholine level since total phosphotidylcholine were not significantly changed across the samples. Statistical analyses on lipid abundance were calculated using GraphPad Prism 9.

## Lipid extraction and thin-layer chromatography

Cells in six-well cell-culture plates were pulse-labelled with 500 μM [¹⁴C]-OA (50 μCi μmol⁻¹) for designated time. Cells were washed with ice-cold PBS two times, then collected by trypsinization. Cell pellets were collected by centrifugation at 600g for 5 min and resuspended with 85 μl Milli Q water on ice. For DNA measurement, 5 μl of total resuspension was aliquoted into 95 μl Milli Q water and lysed by freezing–thawing and subsequent 1 h incubation at 37 °C with gentle shaking. DNA samples were prepared with FluoReporter blue fluorometric dsDNA quantification kit (Thermo Fisher Scientific) and measured by TECAN microplate reader (TECAN). For lipid extraction, the remaining 80 μl of total lysates was mixed with 300 μl of CHCl₃:MeOH (1:2, v/v), vortexed, mixed with 100 μL of CHCl₃, vortexed and mixed with 100 μl of Milli Q water. After centrifugation at 2,000g for 5 min, organic bottom layers were transferred into a fresh microfuge tube and dried under nitrogen stream. Samples were normalized by DNA concentration and separated by thin-layer chromatography (TLC) with hexane:diethyl ether:acetic acid (80:20:1) solvent system. TLC plates were exposed to a phosphor imaging cassette overnight and revealed by Typhoon FLA 7000 phosphor imager. Standard lipids on TLC plate were stained with iodine vapours afterwards.

## Fluorescence microscopy

Cells were plated on 35 mm glass-bottom dishes (MatTek Corp). Imaging was carried out at 37 °C approximately 24 h after transfection. For fixed samples, cells were washed with ice-cold PBS twice, followed by incubation with 4% formaldehyde (Polysciences)–PBS for 20 min at room temperature. After fixation, cells were washed three times with PBS for 5 min. Where noted, cells were stained with 0.5 μM BODIPY493/503 (Thermo Fisher Scientific) and 1 μg ml⁻¹ Hoechst 33342 (Thermo Fisher Scientific) approximately 20 min before imaging.

Spinning-disc confocal microscopy was performed using a Nikon Eclipse Ti inverted microscope equipped with Perfect Focus, a CSU-X1 spinning disk confocal head (Yokogawa), Zyla 4.2 Plus scientific complementary metal-oxide semiconductor cameras (Andor), and controlled by NIS-Elements software (Nikon). To maintain 85% humidity, 37 °C and 5% CO₂ levels, a stage top chamber was used (Okolab). Images were acquired through a 60× Plan Apo 1.40 NA objective or 100× Plan Apo 1.40 NA objective (Nikon). Image pixel sizes were 0.107 and 0.065 μm, respectively. Blue, green, red and far-red fluorescence was excited by 405, 488, 560 or 637 nm (solid state; Andor, Andor, Cobolt and Coherent, respectively) lasers. All laser lines shared a quad-pass dichroic beamsplitter (Di01-T405/488/568/647, Semrock). Blue, green, red and far-red emission was selected with FF01-452/45, FF03-525/50, FF01-607/36 or FF02-685/40 filters (Semrock), respectively, mounted in an external filter wheel. Multi-colour images were acquired sequentially. For Keima dual-excitation ratiometric measurement, images were acquired by 488 (green) and 561 nm (magenta) excitations, collected with a 607/36 nm emission filter.

High-throughput imaging was performed on an IN CELL Analyzer 6000 microscope (GE Healthcare Life Sciences) using a 60× 0.95 NA objective lens. Cells were prepared in 24-well glass-bottom plates and fixed in 4% formaldehyde–PBS at room temperature for 20 min, washed three times with PBS and stained with 0.5 μM BODIPY493/503 for LDs and Hoechst 33342 (Thermo Fisher Scientific) for nuclei. Twenty images were acquired per well. LD areas and numbers from high-throughput microscopic images were quantified using CellProfiler software[52].

## Image processing and quantification

All acquired images were processed and prepared for figures using Fiji[53], using 'Subtract Background', 'Merge Channel' and 'Crop' tools.

For LD number quantification (shown in Fig. 5c and Extended Data Fig. 9b), more than 10 (Fig. 5c) or 30 cells (Extended Data Fig. 9b) of each condition were analysed. For Extended Data Fig. 8b, cell outlines were manually drawn on the basis of mScarlet-I channel, and LDs in BODIPY channel were identified by 'Find maxima' tool in Fiji.

For LD area quantification (shown in Fig. 4j and Extended Data Fig. 3g), cell outlines were manually drawn on the basis of BODIPY

channel. LD areas were detected by Otsu image thresholding algorithm. Total LD area per cell area was calculated as the total LD area divided by cell area, then all sample values were normalized by the value of the WT, 0 h sample.

For ratiometric fluorescence Keima measurement (shown in Fig. 4c,e and Extended Data Figs. 4f, 5b, 6 and 9d–g), areas expressing Keima-reporter were detected by Otsu image thresholding algorithm for both 488 nm (488 nm excitation and FF01-607/36 emission filter) and 561 nm (561 nm excitation and FF01-607/36 emission filter) channels, and those selected areas were combined to get total area expressing Keima-reporter. For the Keima–*LiveDrop* measurement, cells with proper Keima–*LiveDrop* localization in the endoplasmic reticulum were selected for quantification to minimize misinterpretation of the results obtained from cleavage of the Keima fluorescent protein from *LiveDrop*. Using the 'Calculator Plus' tool in Fiji, signal intensity in 561 nm channel were divided by signal intensity in 488 nm channel. The values reported in each graph were normalized by one of control samples as described in individual figures.

Image analysis was performed in Python for the following figures: Figure 1b,g: For each of the images analysed (1b: 30 min $n = 9$, 60 min $n = 6$; 1 g: (108–666) $n = 4$, ΔUBR $n = 5$, (380–666) $n = 5$, (1–380) $n = 5$), a single cell entirely contained within the image was manually outlined with napari (https://github.com/napari/napari). Image intensities were background subtracted and 'total spartin' content was calculated as the sum intensity of the spartin channel falling within the manually outlined cell mask. LD centroids and radii were detected using the Laplacian of Gaussian (LoG: skimage.feature.blob_log) method from scikit-image[54]. 'LD-localized spartin content' was calculated as the sum intensity of pixels in the spartin channel falling inside of the radii of one of the detected LDs. 'Spartin enrichment at LDs' is defined as the ratio of LD-localized spartin to total spartin.

Extended Data Fig. 1f: PLIN3 enrichment at LDs was calculated in the same way as described above for spartin, but for the PLIN3 channel.

Figure 1d,j and Extended Data Fig. 1i: In either two- or three-channel images of spartin, PLIN3 and LDs, LDs were again detected using LoG blob detection, and a binary LD mask was created using the detected centroids and radii. To create a mask for the surface of the LD, the radius for each LD mask was expanded by 2–5 pixels (in proportion to the radius of the LD) and a disk in the centre of the LD was removed from the mask, leaving a donut-shaped mask. For each cell, 'Spartin enrichment at LDs' was calculated as the mean spartin intensity inside the LD mask relative to the mean intensity in the entire cell. The 'lipid-droplet-localized' Pearson correlation coefficient between spartin and PLIN3 (Fig. 1d) or LD (Fig. 1j and Extended Data Fig. 1i) was calculated using numpy.corrcoef[55].

Figures 2b,d,h and 3d, h, and Extended Data Figs. 3d and 4d: For each of the images analysed (2b: OA loading $n = 6$, OA withdrawal $n = 6$, 2d: WT $n = 12$, KO $n = 14$; 2 h: WT $n = 8$, ATG5 KO $n = 3$, ATG7 KO $n = 13$, FIP200 KO $n = 11$; 3d: FL $n = 10$, ΔUBR $n = 10$; 3 h: WT $n = 5$, ΔLIR $n = 5$; Extended Data Fig. 3d: WT $n = 7$, ΔMIT $n = 10$, ΔPPAY $n = 8$; Extended Data Fig. 4d: dimethyl sulfoxide $n = 10$, TAK-243 $n = 1$), images were background subtracted, and a single cell per field of view was outlined, and the nucleus was selected as described for Fig. 1b. To generate cell periphery masks, the nuclear mask was dilated by 160 pixels and subtracted from the whole-cell mask. Within this peripheral mask, covariance between LAMP1 and *LiveDrop* (Figs. 2d and 3d, and Extended Data Fig. 1d) or spartin (Fig. 2b,h and Extended Data Fig. 3d) or between LC3A and spartin (Fig. 3h) was calculated by Pearson correlation coefficient as described for Fig. 1d.

Figure 5g: The number of mature LDs per unit area was measured from 2D confocal images. Putative LDs were first detected using LoG as described above. To reduce false positives from non-specific tissue, 'true mature' LDs were filtered as follows: in a small image patch containing a putative LD coordinate (stained by BODIPY493/503), the correlation between the BODIPY and mScarlet-I channels was measured. Because true mature LDs were generally negatively correlated with mScarlet-I (that is, the core of matured LD stained with BODIPY was not labelled with mScarlet-I, as mScarlet-I-tagged spartin-DN decorates the surface of the LD monolayer), only LDs with a negative correlation coefficient between the LD and mScarlet-I channel were included. Finally, for each image analysed, we report the number of true LDs detected normalized to the total image area (2,048 × 2,084 pixels at 0.066 μm pixel size 18,270 μm$^2$).

## iPS cell generation by genome editing and motor neuron differentiation

Human episomal iPS cells (Gibco, cat. no. A18944) were cultured in E8 medium[56] on Matrigel-coated tissue plates with daily medium changes. Cells were passaged every 4–5 days with 0.5 mM EDTA in 1× DPBS (Thermo Fisher Scientific). The HiFi Cas9 (R691A) (ref. 57) expression plasmid pET-(R691A) SpCas9 HiFi-NLS-6xHis was generated by introducing R691A (CGC to GCC) mutation into the Cas9 sequence in pET-Cas9-NLS-6xHis (Addgene plasmid #62933). The primers for PCR mutagenesis were: 5′-TTTTGCCAATgcCAATTTTATGCAGC and 5′-CCATCTGATTTCAAAAAATCTAATATTG. The resulting plasmid was transformed into Rosetta(DE3)pLysS Competent Cells (Novagen). HiFi Cas9 (R691A) protein was purified as described[58]. The single guide RNA was generated using GeneArt Precision gRNA Synthesis Kit (Thermo Fisher Scientific), according to the manufacturer's instruction, and purified using RNeasy Mini Kit (Qiagen).

For introduction of TRE3G–NGN2–ISL1–LHX3 (NIL) into the AAVS1 site, a donor plasmid pAAVS1–TRE3G–NIL was generated by inserting the P2A–ISL1–T2A–LHX3 sequence, amplified from plasmid pCSC-ISL1-T2A-LHX3 (Addgene plasmid #90215), into plasmid pAAVS1–TRE3G–NGN2 (ref. 59) at the 3′ end of NGN2 sequence by Gibson assembly. Five micrograms of pAAVS1–TRE3G–NIL, 2.5 μg of hCas9 (Addgene plasmid #41815) and 2.5 μg of gRNA_AAVS1-T2 (Addgene plasmid #41818) were electroporated into $1 × 10^6$ human iPS cells. The cells were treated with 0.25 μg ml$^{-1}$ of puromycin for 7 days, and surviving colonies were expanded and subjected to genotyping. The primers for 5′ junction PCR were: 5′-CTCTAACGCTGCCGTCTCTC and 5′-TGGGCTTGTACTCGGTC ATC. The primers for 3′ junction PCR were 5′-CACACAACATACGAGCCGGA and 5′-AC CCCGAAGAGT GAGTTTGC. The primers for locus PCR were 5′-AACCCCAAAGTACC CCGTCT and 5′-CCAGGATCAGTGAAACGCAC.

To generate Gibco-SGP20$^{-/-}$, 0.6 μg of single guide RNA targeting sequence CTCTACAGAATGTAC GCACC was incubated with 3 μg of SpCas9 protein for 10 min at room temperature and electroporated into $2 × 10^5$ H9 cells. Mutants were identified by Illumina MiSeq and further confirmed by western blot (WB).

The neural differentiation of human iPS cells was performed as published[60] with minor modifications. Briefly, cells were dissociated with Accutase (Thermo Fisher Scientific) and plated at $1–2 × 10^4$ cm$^{-2}$ on Matrigel-coated tissue plates in DMEM/F12:Neurobasal (2:1, Thermo Fisher Scientific) supplemented with N2, B27 (0.8% each, Thermo Fisher Scientific), FGF2 (10 ng ml$^{-1}$), Forskolin (10 μM, BioGems), dorsomorphin (1 μM, BioGems), doxycycline (2 μg ml$^{-1}$, ALFAR AESAR) and Y27632 (10 μM, BioGems) on day 0. On day 1, Y27632 was withdrawn. From day 3 to day 9, half of the medium was changed every other day. On day 10, medium was replaced with DMEM/F12:Neurobasal (2:1, Thermo Fisher Scientific) supplemented with N2, B27 (0.8% each, Thermo Fisher Scientific), Forskolin (5 μM, BioGems), BDNF, GDNF and NT3 (10 ng ml$^{-1}$ each, Peprotech). Half of the medium was changed twice a week thereafter. For imaging, cells were fixed with 4% PFA and 4% sucrose for 20 min at room temperature, rinsed twice with PBS, permeabilized with PBS containing 0.05% saponin and 0.1% BSA for 5 min, incubated in PBS containing 0.05% saponin and 5% BSA for 1 h. Then, cells were stained with MAP2 antibody for 1 h at room temperature. After three

Letter

washes with PBS containing 0.05% saponin and 0.2% BSA for 5 min each, cells were stained with chicken IgY-AF488 for 1 h at room temperature. Before imaging, LDs were stained with BODIPY493/503 dye.

## Statistics and reproducibility

Unless otherwise stated, results are presented as mean ± standard deviation. Statistical analyses of results were performed using GraphPad Prism 9 (for statistical details of each experiment, see figure legends). Information about sample size and type of significance test is provided in the legends. Statistically significant differences are denoted as follows: $*P < 0.05$, $**P < 0.01$, $***P < 0.001$, $****P < 0.0001$.

The number of independent experiments repeated for each representative result shown in main and extended data figures is provided here unless otherwise noted in the figure legends: for Fig. 2e, $n = 3$ independent experiments; Fig. 2f, $n = 3$ independent experiments; Fig. 3a, $n = 4$ independent experiments; Fig. 3b, $n = 2$ independent experiments; Fig. 3f, $n = 3$ independent experiments; Fig. 5a, $n = 3$ independent experiments; Extended Data Fig. 1b, $n \geq 10$ biologically independent cells; Extended Data Fig. 1c,d, $n = 3$ independent experiments; Extended Data Fig. 1g, $n = 3$ independent experiments; Extended Data Fig. 1j, $n \geq 8$ biologically independent cells; Extended Data Fig. 1k, $n = 4$ independent experiments; Extended Data Fig. 2b, $n = 5$ independent experiments; Extended Data Fig. 2c, $n = 3$ independent experiments; Extended Data Fig. 2d, $n = 3$ independent experiments; Extended Data Fig. 2c, $n = 2$ independent experiments; Extended Data Fig. 2e, $n = 3$ independent experiments; Extended Data Fig. 4a, $n = 4$ biological replicates in two independent experiments; Extended Data Fig. 4b, $n = 2$ independent experiments; Extended Data Fig. 5a, $n = 3$ independent experiments; Extended Data Fig. 7, $n = 2$ independent experiments; Extended Data Fig. 8b, we checked sgDNA insertion using PCR once before confirming it by sequencing; Extended Data Fig. 8d, $n = 5$ independent experiments; Extended Data Fig. 9h, $n = 3$ independent experiments.

The sample size of each measurement was determined by the practical limitations of the protocol utilized. The sample size of Fig. 5f–h was chosen on the basis of the previous studies[8]. No statistical methods were used to pre-determine the sample size. No sample size calculation was performed for fluorescence microscopy experiments.

No data were excluded for analysis post-image acquisition. Of note, however, for Keima–*LiveDrop* imaging, cells showing cytosolic soluble Keima accumulation (reflecting a cleavage of Keima fluorescent protein tag from the *LiveDrop* protein, thereby detaching it from the *LiveDrop* marker) were not included for image acquisition.

All AAV injection experiments (of Fig. 5f–h) were conducted with randomization of individual animals. For all other cell experiments, randomization was not relevant/not performed, and the control and test conditions were performed on the same day using the same reagents except for the treatment tested (such as RNA interference or transfection).

All lipidomics and analyses were performed in a blind manner to the AAV genotypes. For all other cell experiments, randomization was not relevant/not performed.

## Reporting summary

Further information on research design is available in the Nature Portfolio Reporting Summary linked to this article.

## Data availability

Lipidomics data generated in this study are included in Supplementary Tables 3 and 4. Source data are provided with this paper. All other data supporting the findings of this study are available from the corresponding authors on reasonable request. Requests will be handled according to the Harvard T. H. Chan School of Public Health policies regarding MTA and related matters.

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

## Acknowledgements

We thank members of Farese & Walther laboratory and S. Kim for helpful discussions, W. Harper for helpful discussion and for sharing of reagents, I. Dikic and T. Kirchhausen for sharing reagents, X. Wu for advice on in vitro GST-pulldown analysis and for kindly sharing p12xHis-SUMO-plasmid, and G. Howard for editorial assistance. Other gifts of reagents are acknowledged in Methods. This work was supported by NIH R01GM124348 (to R.V.F.), NIH R01GM097194 (to T.C.W.) and NIH R21AG068423 (to J.P.). J.C. is a fellow of the Damon Runyon Cancer Research Foundation. T.C.W. is a Howard Hughes Medical Institute Investigator.

## Author contributions

J.C., R.V.F. and T.C.W. conceived the project; J.C., J.P., T.C.W. and R.V.F. designed the experiments; and J.C. performed and analysed most of the experiments. J.P. performed virus production, injections and mouse brain histology work. Z.W.L. performed lipidomics analyses. J.Z. generated spartin KO iPS cell lines. T.J.L. wrote scripts for imaging analyses, and T.J.L. and J.C. analysed imaging data together. R.C.R. helped J.C. with molecular cloning experiments and iPS cell experiments. J.C., T.C.W. and R.V.F. wrote the manuscript. All authors discussed the results and contributed to the manuscript.

## Competing interests

The authors declare no competing interests.

## Additional information

**Extended data** is available for this paper at

**Supplementary information** The online version contains supplementary
material available at https://doi.org/10.1038/s41556-023-01178-w.

**Correspondence and requests for materials** should be addressed to
Tobias C. Walther or Robert V. Farese.

**Peer review information** *Nature Cell Biology* thanks Evan Reid,
Peng Li and the other, anonymous, reviewer(s) for their contribution to
the peer review of this work. Peer reviewer reports are available.

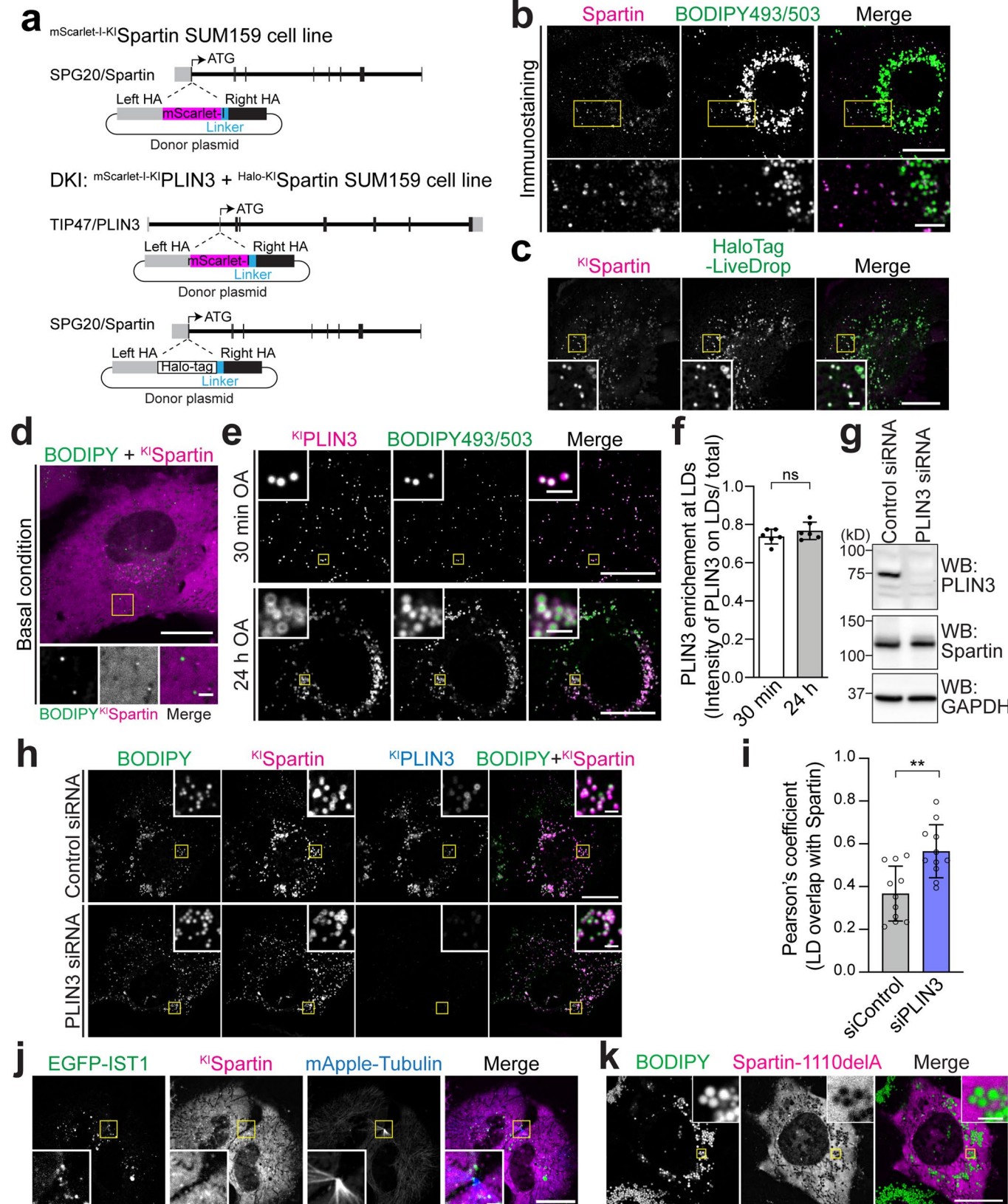

**Extended Data Fig. 1 | See next page for caption.**

**Extended Data Fig. 1 | Endogenous spartin targets to LDs at the cell periphery. a**, Strategy for the endogenous tagging of *SPG20/spartin* and *TIP47/PLIN3*. Individual rectangular bar represents exon (gray color, untranslated region; black color, translated region). A donor plasmid was co-transfected with appropriate gRNA/Cas9 plasmid to induce homologous recombination of mScarlet-I or HaloTag into upstream of *SPG20/spartin* or *TIP47*/PLIN3. **b**, Confocal imaging of fixed SUM159 cells stained with endogenous spartin antibody and BODIPY493/503. Cells were treated with 0.5 mM OA for 24 h. Scale bars: full-size, 20 μm; insets, 2 μm. **c,d**, Confocal imaging of live SUM159 cells expressing endogenously fluorescent-tagged spartin (with mScarlet-I) at their gene locus. HaloTag–*LiveDrop* was transiently expressed and stained with 100 nM JF646 (c) and BODIPY493/503 (d). Cells were treated with 0.5 mM OA for 24 h (c). Scale bars: full-size, 20 μm; insets, 2 μm. **e**, Confocal imaging of live SUM159 cells expressing endogenously fluorescent-tagged spartin (HaloTag) and PLIN3 (mScarlet-I). Cells were pre-incubated with 0.5 mM OA for 30 min or 24 h prior to confocal live-cell microscopy. Scale bars: full-size, 20 μm; insets, 2 μm. **f**, Quantification of PLIN3 enrichment at LDs in PLIN3 and spartin double-KI SUM159 cells shown in (e). Mean ± SD, n = 6 fields of view from three independent experiments, n.s. not significant, two-tailed unpaired t-test. **g**, Immunoblot analyses of SUM159 PLIN3 (mScarlet-I) and spartin (HaloTag) double KI cells. Cells were treated with 20 nM negative control siRNA or PLIN3 siRNA for 72 h prior to lysate preparation. **h**, Confocal imaging of live SUM159 PLIN3 (mScarlet-I) and spartin (HaloTag) double KI cells. Cells were treated with 20 nM negative control siRNA or PLIN3 siRNA for 72 h prior to imaging. Cells were pre-incubated with 0.5 mM OA for 24 h and chased for 3 h after OA withdrawal, stained with JF646 (for HaloTag KI–spartin) and BODIPY493/503. Scale bars: full-size, 20 μm; insets, 2 μm. **i**, Quantification of spartin enrichment at LDs in PLIN3 and spartin double-KI SUM159 cells shown in (h). Mean ± SD, n = 11 cells from three independent experiments, **p = 0.0015, two-tailed unpaired t-test. **j**, Confocal imaging of live SUM159 cells expressing endogenously fluorescent-tagged spartin (with mScarlet-I) at their gene locus. EGFP–IST1 and mApple–Tubulin were transiently expressed. Scale bars: full-size, 20 μm; insets, 2 μm. **k**, Confocal imaging of live SUM159 spartin KO cells transiently expressing mScarlet-I–spartin-1110delA. Cells were treated with 0.5 mM OA for 24 h and stained with BODIPY493/503. Scale bars: full-size, 20 μm; insets, 2 μm. Source numerical data and unprocessed blots are available in source data.

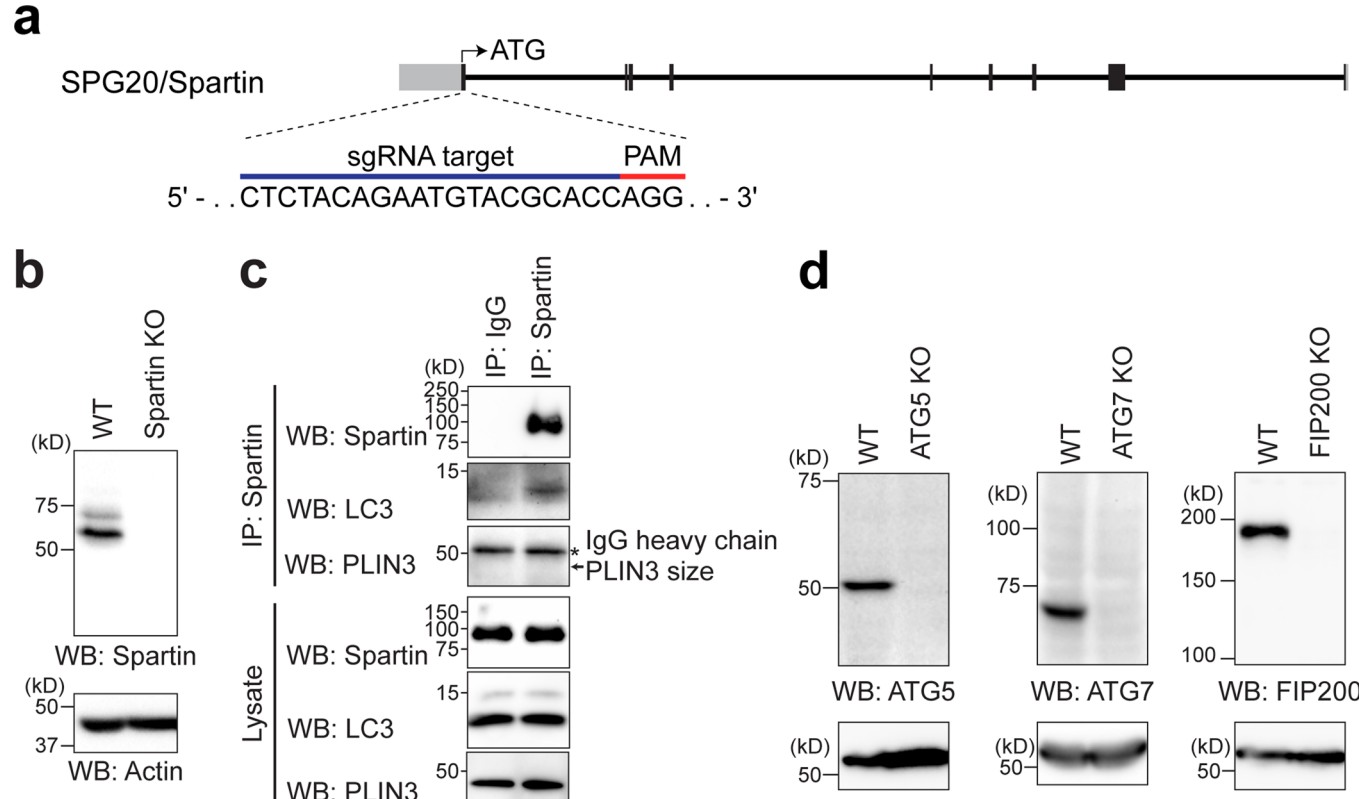

**Extended Data Fig. 2 | Spartin is associated with lysosomes and autophagy machineries. a**, Strategy for the knockout of *SPG20/spartin*. Individual rectangular bar represents exon (gray color, untranslated region; black color, translated region). An sgRNA target sequence is shown with PAM sequence. **b**, Immunoblot analyses of SUM159 cells, wild-type and cells lacking spartin. **c**, Endogenous spartin interacts with endogenous LC3. Endogenous spartin was immunoprecipitated with anti-spartin antibody in SUM159 wild-type cells and analyzed by immunoblot with anti-spartin, anti-LC3, and anti-PLIN3 antibodies. * denotes IgG heavy change. **d**, Immunoblot analyses of SUM159 cells wild-type and cells lacking ATG5, ATG7, or FIP200. Unprocessed blots are available in source data.

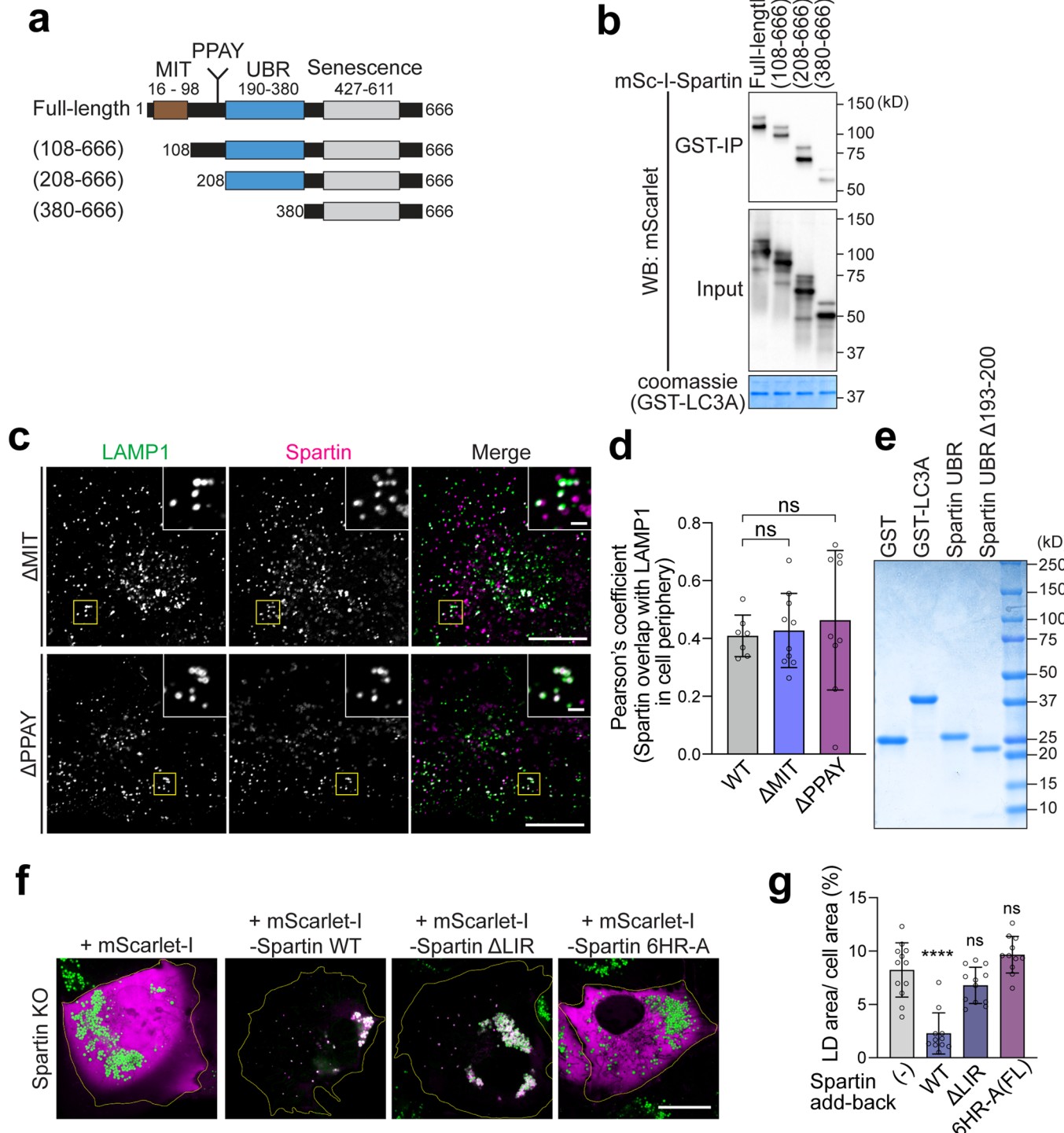

**Extended Data Fig. 3 | Domain mapping of interaction between spartin and ATG8 proteins. a**, Schematic representation of spartin truncation mutants used in GST pull-down analysis. **b**, SDS-PAGE analysis of recombinant GST and GST−LC3A purified from bacteria (after size-exclusion column). Proteins were visualized by Coomassie blue staining. **c**, Confocal imaging of live SUM159 cells transiently expressing LAMP1−mNG and mScarlet-I−spartin-ΔMIT (top) or mScarlet-I−spartin-ΔPPAY (bottom). Cells were treated with 0.5 mM OA for 24 h and chased for 3 h after OA withdrawal. Scale bars: full-size, 20 μm; insets, 2 μm. **d**, Overlap between spartin and LAMP1 in cell periphery shown in (c) was quantified by Pearson's coefficient analysis. Mean ± SD, n = 7 cells (FL), 10 cells (dMIT), and 9 cells (dPPAY) from two independent experiments, n.s. not significant, one-way ANOVA, Tukey's multiple comparisons test. **e**, SDS-PAGE analysis of recombinant proteins purified from bacteria. 12XHis-SUMO tag

of spartin constructs were removed by SUMO protease and all proteins were further purified by size-exclusion column. Protein was visualized by Coomassie blue staining. **f**, Confocal imaging of live SUM159 spartin KO cells, transiently transfected with either mScarlet-I or mScarlet-I−spartin constructs as annotated in the image (shown in magenta color). Cells were stained with BODIPY493/503 (shown in green color) prior to image acquisition. Mean ± SD, n = 17 cells from two independent experiments. Scale bars: full-size, 20 μm. **g**, Area of LDs stained by BODIPY493/503 was quantified per cell in SUM159 spartin KO, transiently transfected with mScarlet-I−spartin constructs as annotated below the graph. Mean ± SD, n = 12 cells (no transfection), 11 cells (+ Spartin WT), 12 cells (+ Spartin ΔLIR), and 11 cells (+Spartin 6HR-A) from two independent experiments. one-way ANOVA, Dunnett's multiple comparisons test, ****p < 0.0001, n.s. not significant. Source numerical data and unprocessed blots are available in source data.

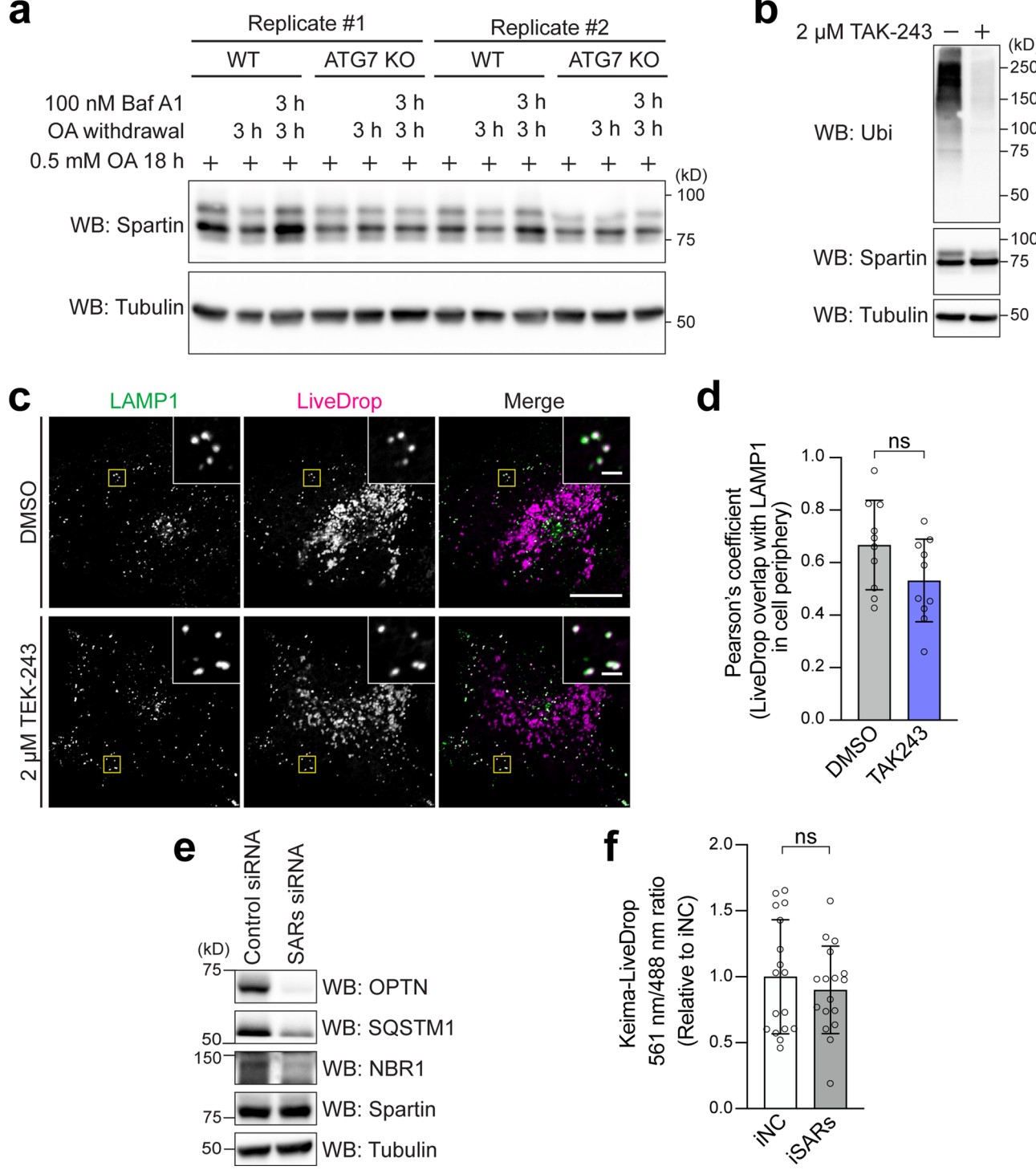

**Extended Data Fig. 4 | Spartin-mediated lipophagy is ubiquitination-independent. a**. Immunoblot analysis of SUM159 cells wild-type and cells lacking ATG7 with the indicated antibodies. **b**, Immunoblot analysis of SUM159 wild-type cells with the indicated antibodies. Cells were treated with DMSO (vehicle) or 2 μM TAK-243 for 2 h prior to cell lysis. **c**, Confocal imaging of live SUM159 cells transiently expressing LAMP1–mNG and mCherry–*LiveDrop*. Cells were treated with 0.5 mM OA for 24 h and chased for 3 h after OA withdrawal, with DMSO (vehicle) or 2 μM TAK-243 for 2 h prior to imaging. Scale bars: full-size, 20 μm; insets, 2 μm. **d**, Overlap between *LiveDrop* and LAMP1 in cell periphery shown in (c) was quantified by Pearson's coefficient analysis. Mean ± SD, n = 10 cells from

two independent experiments, n.s. not significant, two-tailed unpaired t-test. **e**, Immunoblot analyses of SUM159 wild-type cells with indicated antibodies. Cells were treated with 20 nM negative control siRNA or selective autophagy receptors (SARs; OPTN, NBR1, SQSTM1/p62) siRNAs for 72 h prior to lysate preparation. **f**, Ratiometric fluorescence measurement of Keima–*LiveDrop* in SUM159 wild-type cells treated with control siRNA (iNC) or SARs siRNAs for 72 h, n.s. not significant, two-tailed unpaired t-test. Mean ± SD, n = 17 cells from three independent experiments. Source numerical data and unprocessed blots are available in source data.

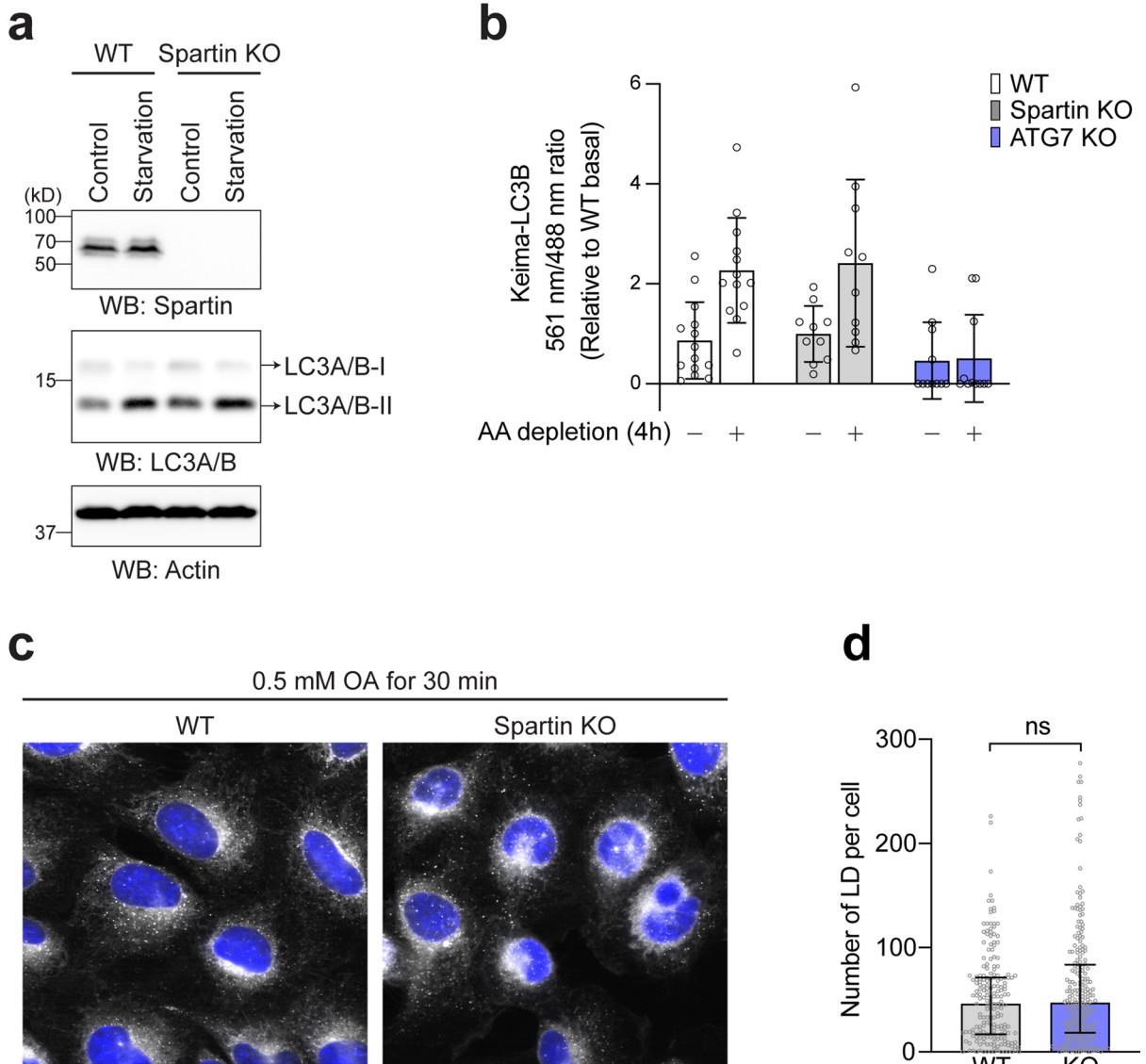

**Extended Data Fig. 5 | Spartin deficiency does not affect general autophagy machinery and LD biogenesis. a**, Immunoblot analyses of SUM159 wild-type and spartin KO cells. Starvation samples were incubated in serum-free and low-glucose medium (1 g/L) for 3 h prior to lysate preparation. **b**, Ratiometric fluorescence measurement of Keima-LC3B in SUM159 wild-type, ATG7 KO, and spartin KO in basal or amino acid depletion for 4 h. Mean ± SD, n = 14 cells (WT-Basal), 13 cells (WT-AA depletion), 10 cells (Spartin KO-Basal), 10 cells (Spartin KO-AA depletion), 11 cells (ATG7 KO-Basal), and 11 cells (ATG7 KO-AA depletion) from two independent experiments. **c**, Confocal imaging of fixed SUM159 wild-type and spartin KO cells stained with BODIPY493/503 and Hoechst 33342. Cells were treated with 0.5 mM OA for 30 min prior to fixation. Scale bar: 20 μm. **d**, Quantification of number of LDs per cell in wild-type and spartin KO SUM159 cells shown in (c) The images were taken on high-throughput confocal microscope. n = 202 cells (WT) and 240 cells (Spartin KO) from three independent experiments, median with interquartile range, n.s. not significant, two-tailed unpaired Welch's t-test. Source numerical data and unprocessed blots are available in source data.

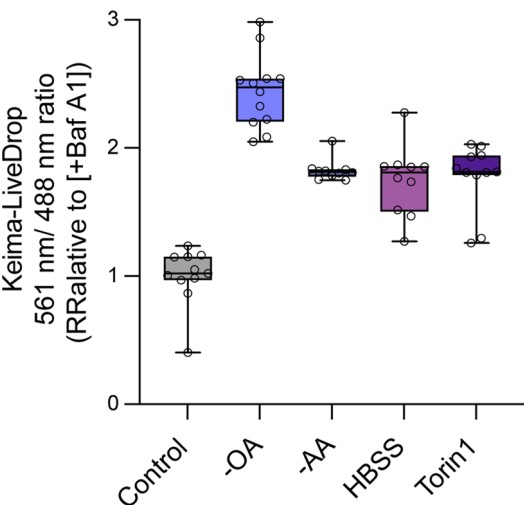

**Extended Data Fig. 6 | Activation of spartin-mediated lipophagy in different metabolic perturbations.** Ratiometric fluorescence measurement of Keima–spartin in SUM159 wild-type cell in various conditions annotated below the graph. Cells were treated with 0.5 mM OA for 24 h and chased for 4 h after OA withdrawal together with metabolic perturbations (100 nM Baf A1 treatment, amino acid depletion, HBSS, or 250 nM Torin 1 addition). Median ± the 25th to 75th percentiles, the whiskers extended to the minima and the maxima, n = 11 cells (Baf A1), 12 cells (OA withdrawal), 10 cells (AA depletion), 10 cells (HHBS), and 11 cells (Torin1) from two independent experiments. Source numerical data are available in source data.

**Extended Data Fig. 7 | Spartin is expressed in neurons of the mouse motor cortex.** Confocal micrographs of the M1 motor cortex slices from 9-week-old wild-type mice. PFA-fixed sagittal sections were co-stained with anti-spartin (magenta) and anti-MAP2 (a neuronal marker; green in the top panels) or anti-GFAP (an astrocyte marker; green in the lower panels). Spartin signals localized on neuronal cell bodies and projections (top). Slices stained with secondary antibodies alone (Alexa Fluor 488 and 594; bottom) were used as background signal controls. Scale bar: 10 μm.

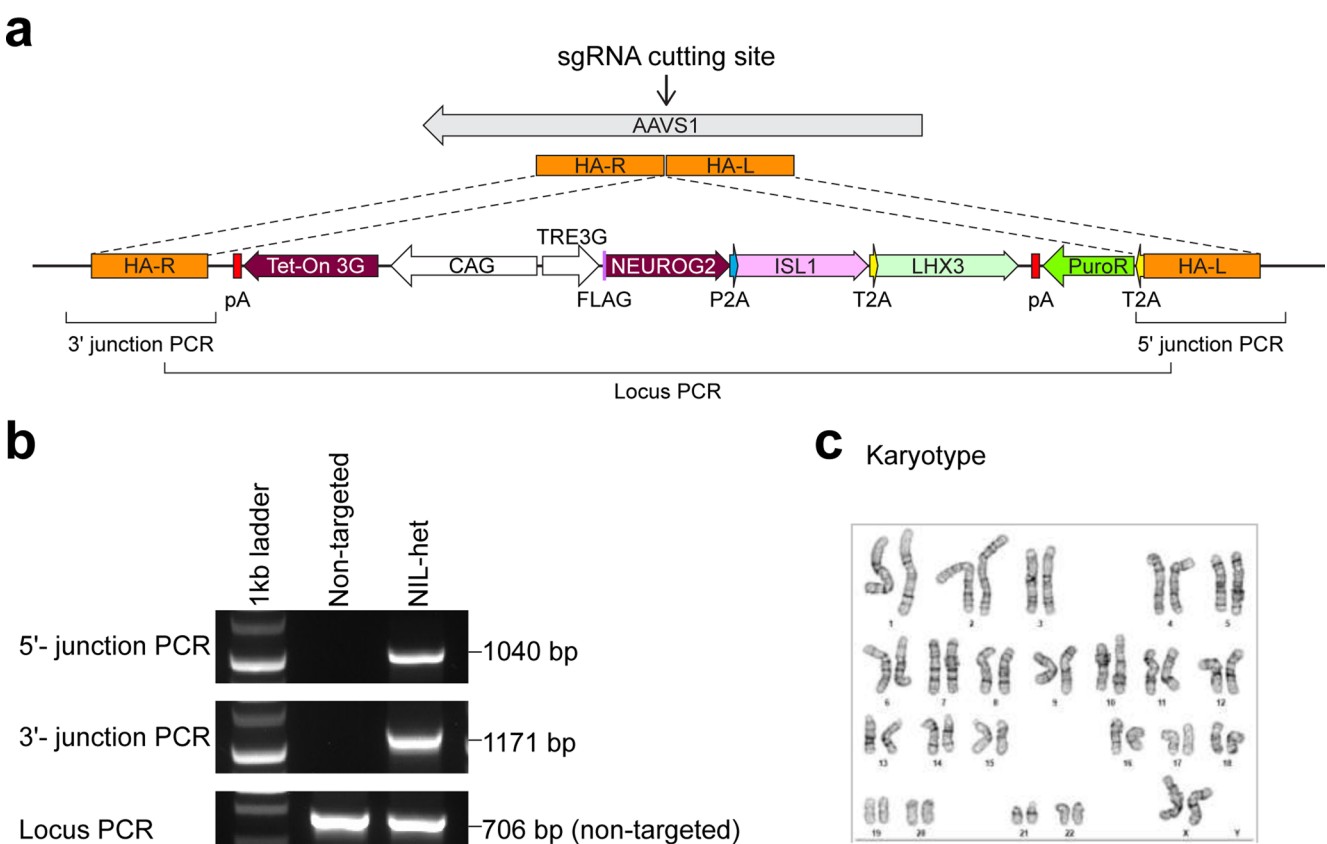

**a**

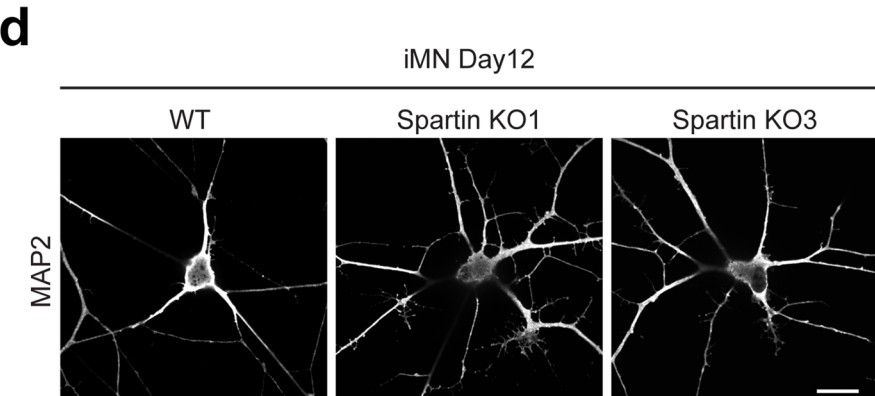

**Extended Data Fig. 8 | Spartin deficiency does not affect motor neuron differentiation. a**, Scheme of the strategy used to target TRE3G-NIL to the AAVS1 locus. **b**, Genotyping PCR results for iPSC-AAVS1-TRE3G-NIL cell line. **c**, Karyotype of iPSC-AAVS1-TRE3G-NIL cell line. **d**, Parental and spartin KO iPSC clones were differentiated to motor neurons in the differentiation medium for 12 days. The cells were fixed and stained with MAP2 antibody. Scare bar: 10 μm.

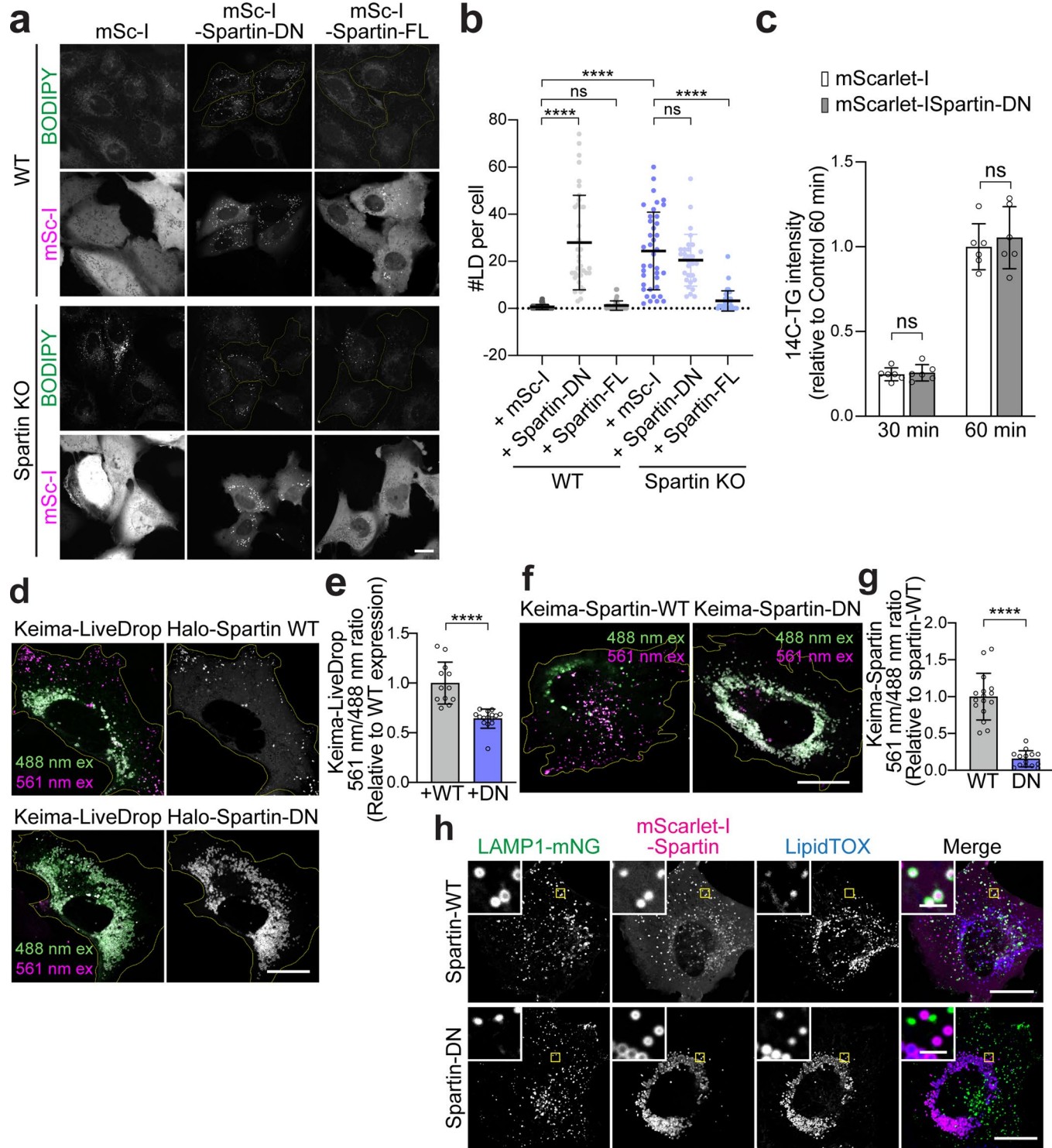

**Extended Data Fig. 9 | See next page for caption.**

**Extended Data Fig. 9 | Spartin-DN impairs LD turnover in SUM159 cells.**
**a**, Confocal imaging of live SUM159 wild-type (top panel) and spartin KO (bottom panel) cells transiently expressing mScarlet-I-tagged spartin-DN or spartin full-length (Spartin-FL) proteins. LDs were stained with BODIPY493/503. Transfected cells were marked with yellow outline. Scale bars: 20 μm. **b**, Quantification of LD numbers per cell shown in (a). Mean ± SD, n = 58 cells (WT+mSc-I), 39 cells (WT+Spartin-DN), 37 cells (WT+Spartin-FL), 39 cells (Spartin KO+Spartin-DN), 33 cells (Spartin KO+Spartin-DN), and 30 cells (Spartin KO+Spartin-FL) from three independent experiments, n.s., not significant, ****$p < 0.0001$, one-way ANOVA, Dunnett's multiple comparisons test. **c**, SUM159 wild-type cells transiently expressing mScarlet-I or mScarlet-I-spartin DN were pulse-labeled with [$^{14}$C]-OA, and incorporation into TG was measured after treatment with 0.5 mM OA for 30 and 60 min. Values were calculated relative to cells expressing mScarlet-I highest value at 60 min. Mean ± SD, n = 6 independent experiments, n.s. not significant, two-way ANOVA, multiple comparisons test. **d**, Overlay live-cell confocal images of Keima–*LiveDrop* acquired by 488 nm (green) and

561 nm (magenta) excitations collected with a 607/36 nm emission filter, expressed in SUM159 wild-type, transiently transfected with Halo–spartin WT and DN constructs as described. Scale bars: 20 μm. **e**, Ratiometric fluorescence measurement of Keima–*LiveDrop* per cell shown in (d). Mean ± SD, n = 12 cells from three independent experiments, ****$p < 0.0001$, two-tailed unpaired t-test. **f**, Overlay live-cell confocal images of Keima–spartin-WT and Keima–spartin-DN acquired by 488 nm (green) and 561 nm (magenta) excitations collected with a 607/36 nm emission filter, expressed in SUM159 wild-type cells. **g**, Ratiometric fluorescence measurement of Keima signal per cell shown in (f). Mean ± SD, n = 14 cells from three independent experiments, ****$p < 0.0001$, two-tailed unpaired t-test. **h**, Confocal imaging of live SUM159 cells transiently expressing LAMP1-mNG and mScarlet-I–spartin-WT (top) or mScarlet-I–spartin-DN (bottom). Cells were treated with 0.5 mM OA for 24 h and chased for 3 h after OA withdrawal. Scale bars: full-size, 20 μm; insets, 2 μm. Source numerical data are available in source data.

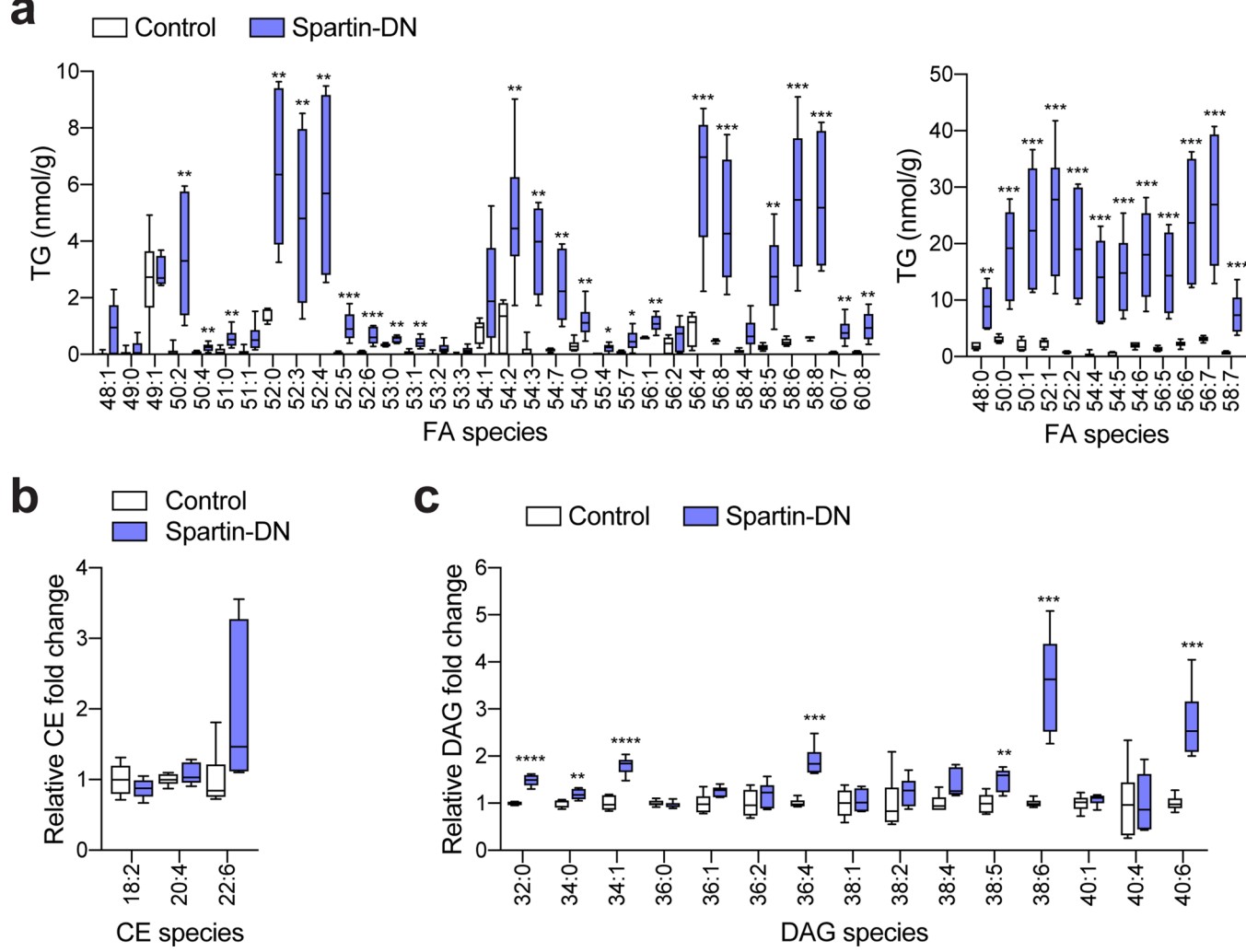

**Extended Data Fig. 10 | Major lipid class changes in the spartin-DN-infected M1 cortices. a–c,** Lipidomic profiles of the AAV-infected M1 cortices. Lipids were extracted from tissues using an MTBE method and analyzed by LC-MS/MS positive ion mode. Absolute amounts of detected TG species, normalized by tissue weight (a). Relative fold changes of CE (b) and DAG (c) lipid classes. Box-and-whisker plot, Median ± the 25th to 75th percentiles, the whiskers extended to the minima and the maxima, n = 6 mice, **p < 0.01, ***p < 0.001, ****p < 0.0001 (a, 58:7 = 0.000754, 56:7 = 0.000393, 56:6 = 0.000444, 56:5 = 0.000738, 54:6 = 0.000463, 54:5 = 0.000426, 54:4 = 0.000849, 52:2 = 0.000516, 52:1 = 0.000452, 50:1 = 0.000688, 50:0 = 0.000782, 48:0 = 0.001107,

60:8 = 0.001432, 60:7 = 0.001566, 58:8 = 0.000339, 58:6 = 0.00057, 58:5 = 0.001099, 56:8 = 0.000765, 56:4 = 0.00035, 56:1 = 0.001576, 55:7 = 0.012073, 55:4 = 0.011659, 54:0 = 0.00632, 54:7 = 0.00109, 54:3 = 0.000166, 54:2 = 0.004482, 53:1 = 0.00169, 53:0 = 0.002413, 52:6 = 0.00086, 52:5 = 0.000911, 52:4 = 0.000621, 52:3 = 0.002428, 52:0 = 0.000684, 51:0 = 0.008827, 50:4 = 0.008494, 50:2 = 0.002913; c, 40:6 = 0.000276, 38:6 = 0.000118, 38:5 = 0.003104, 36:4 = 0.000045, 34:1 = 0.00001, 34:0 = 0.006981, 32:0 = 0.000003), Two-tailed unpaired t-test on each row, multiple comparisons test using the two-stage step-up method of Benjamini, Krieger and Yekutieli. Source numerical data are available in source data.

# Reporting Summary

## Statistics

For all statistical analyses, confirm that the following items are present in the figure legend, table legend, main text, or Methods section.

| n/a | Confirmed | |
|---|---|---|
| ☐ | ☒ | The exact sample size (*n*) for each experimental group/condition, given as a discrete number and unit of measurement |
| ☐ | ☒ | A statement on whether measurements were taken from distinct samples or whether the same sample was measured repeatedly |
| ☐ | ☒ | The statistical test(s) used AND whether they are one- or two-sided *Only common tests should be described solely by name; describe more complex techniques in the Methods section.* |
| ☒ | ☐ | A description of all covariates tested |
| ☐ | ☒ | A description of any assumptions or corrections, such as tests of normality and adjustment for multiple comparisons |
| ☐ | ☒ | A full description of the statistical parameters including central tendency (e.g. means) or other basic estimates (e.g. regression coefficient) AND variation (e.g. standard deviation) or associated estimates of uncertainty (e.g. confidence intervals) |
| ☐ | ☒ | For null hypothesis testing, the test statistic (e.g. *F*, *t*, *r*) with confidence intervals, effect sizes, degrees of freedom and *P* value noted *Give P values as exact values whenever suitable.* |
| ☒ | ☐ | For Bayesian analysis, information on the choice of priors and Markov chain Monte Carlo settings |
| ☒ | ☐ | For hierarchical and complex designs, identification of the appropriate level for tests and full reporting of outcomes |
| ☐ | ☒ | Estimates of effect sizes (e.g. Cohen's *d*, Pearson's *r*), indicating how they were calculated |

*Our web collection on statistics for biologists contains articles on many of the points above.*

## Software and code

Policy information about availability of computer code

| Data collection | NIS-Elements (4.51.01), IN Cell Analyzer 6000 Acquisition Software v1.0 |
|---|---|
| Data analysis | Lipidomics analyses were done using MaxQuant (1.5.2.8) and Perseus (1.5.1.6). Image analyses were done by open-source software, including Fiji (ImageJ 2.3.0/1.53f51), napari (0.3.4), python (3.7.6), scikit-image (0.19.1), and Cell Profiler (v1.0.9717). The detailed information about imaging analyses for each figure is described in the Method section. Statistic analyses were done by GraphPad Prism 9. |

For manuscripts utilizing custom algorithms or software that are central to the research but not yet described in published literature, software must be made available to editors and reviewers. We strongly encourage code deposition in a community repository (e.g. GitHub). See the Nature Portfolio guidelines for submitting code & software for further information.

## Data

Policy information about availability of data

All manuscripts must include a data availability statement. This statement should provide the following information, where applicable:
- Accession codes, unique identifiers, or web links for publicly available datasets
- A description of any restrictions on data availability
- For clinical datasets or third party data, please ensure that the statement adheres to our policy

Lipidomics data generated in this study are included in Supplementary Table 3 and 4. Source data have been provided in Source Data. All other data supporting the findings of this study are available from the corresponding authors on reasonable request. Requests will be handled according to the Harvard T. H. Chan School of Public Health policies regarding MTA and related matters.

# Field-specific reporting

Please select the one below that is the best fit for your research. If you are not sure, read the appropriate sections before making your selection.

☒ Life sciences ☐ Behavioural & social sciences ☐ Ecological, evolutionary & environmental sciences

For a reference copy of the document with all sections, see nature.com/documents/nr-reporting-summary-flat.pdf

# Life sciences study design

All studies must disclose on these points even when the disclosure is negative.

| | |
|---|---|
| Sample size | Sample size of each measurement was determined by the practical limitations of the protocol utilized. The sample size of Fig. 5f-h was chosen based on the previous studies (Reference 8; PNAS 111, 14924–14929 (2014), 92, 75-83). No statistical methods were used to predetermine sample size. No sample size calculation was performed for fluorescence microscopy experiments. All fluorescence microscopy experiments were repeated 2-3 independent times with more than 5 observations (cells) each. |
| Data exclusions | No data were excluded for analysis post-image acquisition. Of note, however, for Keima-LiveDrop imaging, cells showing cytosolic soluble Keima accumulation (reflecting a cleavage of Keima fluorescent protein tag from the LiveDrop protein, thereby detaching it from the LiveDrop marker) were not included for image acquisition. |
| Replication | Number of independent experiments are indicated in the respective figure legends. Replication were successful in all cases. |
| Randomization | All AAV injection experiments (of Fig. 5f-h) were conducted with randomization of individual animals. For all other cell experiments, randomization was not relevant/not performed, and the control and test conditions were performed on the same day using the same reagents except for the treatment tested (such as RNAi or transfection). |
| Blinding | All lipidomics and analyses were performed in a blind manner to the AAV genotypes. For all other cell experiments, randomization was not relevant/not performed. |

# Reporting for specific materials, systems and methods

We require information from authors about some types of materials, experimental systems and methods used in many studies. Here, indicate whether each material, system or method listed is relevant to your study. If you are not sure if a list item applies to your research, read the appropriate section before selecting a response.

## Materials & experimental systems

| n/a | Involved in the study |
|---|---|
| ☐ | ☒ Antibodies |
| ☐ | ☒ Eukaryotic cell lines |
| ☒ | ☐ Palaeontology and archaeology |
| ☐ | ☒ Animals and other organisms |
| ☒ | ☐ Human research participants |
| ☒ | ☐ Clinical data |
| ☒ | ☐ Dual use research of concern |

## Methods

| n/a | Involved in the study |
|---|---|
| ☒ | ☐ ChIP-seq |
| ☒ | ☐ Flow cytometry |
| ☒ | ☐ MRI-based neuroimaging |

## Antibodies

| | |
|---|---|
| Antibodies used | Primary antibodies used in this study were (all primary antibodies are used in 1:1000 dilution): rabbit polyclonal anti-SPG20/spartin (Proteintech, #13791-1-AP), rabbit polyclonal anti-FIP200/RB1CC1 (Proteintech, #17250-1-AP), mouse monoclonal anti-FLAG (Millipore-Sigma, #F3165), anti-FLAG® M2 affinity gel (Millipore-Sigma, #A2220), rat monoclonal anti-HA clone 3F10 (Millipore-Sigma, #11867423001), mouse monoclonal anti-α-tubulin (Millipore-Sigma, #T5168), rabbit polyclonal anti-mCherry (for detection of mScarlet-I, Abcam #ab167453), rabbit polyclonal anti-LC3A/B (Cell Signaling Technology, #4108S), mouse monoclonal anti-actin (Cell Signaling Technology, #3700S), rabbit polyclonal anti-ATGL (Cell Signaling Technology, #2138S), rabbit monoclonal anti-ATG5 (Cell Signaling Technology, #12994T), rabbit monoclonal anti-ATG7 (Cell Signaling Technology, #8558S), chicken polyclonal anti-MAP2 (Synaptic Systems), rat polyclonal anti-GFAP (Thermo Fisher Scientific, #13-030-0), rabbit polyclonal anti-GST (Thermo Fisher Scientific, #A5800). ), rabbit polyclonal anti-NBR1 (Proteintech, #16004-1-AP), rabbit polyclonal anti-OPTN (Proteintech, #10837-1-AP), mouse monoclonal anti-SQSTM1 clone 2C11 (Novus, #H00008878-M01). Mouse anti-rabbit IgG-HRP (Santa Cruz Biotechnology, # sc-2357), Mouse anti-IgG kappa binding protein-HRP (Santa Cruz Biotechnology, # sc-516102), Goat anti-rat IgG H&L-HRP (Abcam, # ab97057), DyLight 488-conjugated goat anti-chicken IgY (Thermo Scientific, #SA5-10070), Alexa Fluor 488 donkey anti-rat IgG (Thermo Scientific, #A-21208), and Alexa Fluor 594 goat anti-rabbit IgG (Thermo Scientific, #A-11037). |
| Validation | Rabbit polyclonal anti-SPG20/spartin (Proteintech, #13791-1-AP; validated with Western blot), rabbit polyclonal anti-FIP200/RB1CC1 (Proteintech, #17250-1-AP; validated with Western blot), mouse monoclonal anti-FLAG (Millipore-Sigma, #F3165; validated with Western blot), anti-FLAG® M2 affinity gel (Millipore-Sigma, #A2220; validated with Western blot), rat monoclonal anti-HA clone 3F10 |

(Millipore-Sigma, #11867423001; validated with Western blot and immunofluorescence), mouse monoclonal anti-α-tubulin (Millipore-Sigma, #T5168; validated with Western blot), rabbit polyclonal anti-mCherry (for detection of mScarlet-I, Abcam #ab167453; validated with Western blot and immunofluorescence), rabbit polyclonal anti-LC3A/B (Cell Signaling Technology, #4108S; validated with Western blot), mouse monoclonal anti-actin (Cell Signaling Technology, #3700S; validated with Western blot), rabbit polyclonal anti-ATGL (Cell Signaling Technology, #2138S; validated with Western blot), rabbit monoclonal anti-ATG5 (Cell Signaling Technology, #12994T; validated with Western blot), rabbit monoclonal anti-ATG7 (Cell Signaling Technology, #8558S; validated with Western blot), chicken polyclonal anti-MAP2 (Synaptic Systems; validated with immunohistochemistry), rat polyclonal anti-GFAP (Thermo Fisher Scientific, #13-030-0; validated with immunohistochemistry), rabbit polyclonal anti-GST (Thermo Fisher Scientific, #A5800; validated with Western blot), rabbit polyclonal anti-NBR1 (Proteintech, #16004-1-AP; validated with Western blot), rabbit polyclonal anti-OPTN (Proteintech, #10837-1-AP; validated with Western blot), mouse monoclonal anti-SQSTM1 clone 2C11 (Novus, #H00008878-M01; validated with Western blot). Mouse anti-rabbit IgG-HRP (Santa Cruz Biotechnology, # sc-2357; validated with Western blot), Mouse anti-IgG kappa binding protein-HRP (Santa Cruz Biotechnology, # sc-516102; validated with Western blot), Goat anti-rat IgG H&L-HRP (Abcam, # ab97057; validated with Western blot), DyLight 488-conjugated goat anti-chicken IgY (Thermo Scientific, #SA5-10070; validated with immunofluorescence), Alexa Fluor 488 donkey anti-rat IgG (Thermo Scientific, #A-21208; validated with immunofluorescence), and Alexa Fluor 594 goat anti-rabbit IgG (Thermo Scientific, #A-11037; validated with immunofluorescence).

# Eukaryotic cell lines

Policy information about cell lines

| | |
|---|---|
| Cell line source(s) | Human episomal iPSCs (Gibco, Cat# A18944), HEK293T (ATCC, #CRL-3216). SUM159 breast cancer cells were obtained from the laboratory of Dr. Tomas Kirchhausen (Harvard Medical School; original source, RRID:cvcl_5423). |
| Authentication | We routinely perform cell authentication via STR profiling and the last authentication of the cell lines used in the study was performed on 04/27/22. The generated cell lines were verified with western blot and mass spectrometer-based proteomics. |
| Mycoplasma contamination | All cell lines tested negative for mycoplasma. |
| Commonly misidentified lines (See ICLAC register) | No commonly misidentified cell lines were used. |

# Animals and other organisms

Policy information about studies involving animals; ARRIVE guidelines recommended for reporting animal research

| | |
|---|---|
| Laboratory animals | C57BL/6J wild-type mice obtained from the Jackson Laboratory (Stock# 000664); male and female randomized; 7- to 10-week-old littermates. All mice were housed on a 12-h light-dark cycle with food and water ad libitum (65±75F and 40±60% humidity). |
| Wild animals | No wild animals were used in the study. |
| Field-collected samples | No field-collected samples were used in the study. |
| Ethics oversight | Studied under the guidelines of the Institutional Animal Care and Use Committee of Wayne State University. |

Note that full information on the approval of the study protocol must also be provided in the manuscript.

