## [Peer Review File · Nature Cell Biology]

Peer Review Information

Journal: Nature Cell Biology

Manuscript Title: The Troyer syndrome protein spartin mediates selective autophagy of lipid droplets

Corresponding author name(s): Dr Tobias Walther

Editorial Notes:

Reviewer Comments & Decisions:

Decision Letter, initial version:
--

Dear Tobi,

Thank you for submitting your manuscript, "The Troyer syndrome protein spartin mediates selective autophagy of lipid droplets", to Nature Cell Biology. It has now been evaluated by 3 referees, who are experts in neurological diseases, organelles & membrane traffic (Referee #1); autophagy (Referee #2); and lipid droplets (Referee #3). As you will see from their comments (attached below), they found this work of potential interest but have raised substantial concerns, which in our view would need to be addressed with considerable revisions before we can consider publication in Nature Cell Biology.

As per our standard editorial process, we have discussed the reviews in detail within the editorial team, including our Chief Editor, to define the scope of revisions that is needed to strengthen the analyses, as opposed to questions that are more peripheral and beyond the scope of the current study. To guide the scope of the revisions, I have listed these points below. We are committed to providing a fair and constructive peer-review process, so please feel free to contact me if you would like to discuss any of the referee comments further. As you know, our typical revision period is six months; please do let us know if you expect any issues addressing the reviews or would like to discuss further.

The reviewers were not yet convinced that spartin acts as a lipophagy receptor and their concerns were numerous and significant. We feel that the referees' concerns point to a premature dataset and concerns regarding the role of spartin in autophagy globally would need to be addressed thoroughly with experiments and data. Reconsideration of the study for this journal and re-engagement of the

referees would depend on the strength of these revisions at resubmission. We appreciate that the reviewers suggested a substantial amount of work and that these experiments are not trivial. In our view, they would nonetheless be important to convince experts in this field that spartin does functionally act as a lipophagy receptor, with mechanistic data to support this role.

A. Validating that spartin is a true lipophagy receptor should be a priority in revision. As per the reviewers, this conclusion requires more mechanistic support. The reviewers feel (and we agree) it's important to clarify the contribution of ubiquitin (Rev#2 #7; Rev#3 #2; see also Rev#1 point #6), to provide more time-lapse/live imaging of the lipophagy process and the role of spartin in it (Rev#2 #5, Rev#3 #4), rule out roles for NBR1/OPTN/p62 (Rev#3 #2 - this is a very important question in our view), further explore and define the interactions between spartin and LC3 proteins (Rev#1 #2, Rev#3 #3 - mapping the interaction would be sufficient, structural analyses would not be required if the reviewer's question can be addressed with the mapping studies).

Rev#3 asked for a mutant that disrupts spartin's localization at LDs and its interaction with ATG8; would any of the figure 1 mutants (and/or in combination with Fig 2 mutants) fit for this line of analysis?

B. Please also check that spartin functions in lipophagy (Rev#3 #5) under other types of stress, and not in lipolysis (Rev#2 #6), with validation that it does specifically function to regulate lipophagy in neurons (Rev#2 #4), as opposed to LD biogenesis. We strongly encourage you to study the role of spartin in adipocytes (Rev#2 #1) and agree with Rev#2 that a function for spartin in bulk autophagy should be ruled out thoroughly (Rev#2 #8).

C. Given your access to the mice expressing spartin-DN, please answer Rev#2's question about their phenotypes (Rev#2 #2) to enrich the phenotypic analyses.

D. All other referee concerns pertaining to strengthening existing data, providing controls, methodological details, clarifications and textual changes should also be addressed.

E. Finally, please pay close attention to our guidelines on statistical and methodological reporting (listed below) as failure to do so may delay the reconsideration of the revised manuscript. In particular please provide:

We would be happy to consider a revised manuscript that would satisfactorily address these points, unless a similar paper is published elsewhere, or is accepted for publication in Nature Cell Biology in the meantime.

In contrast, although we agree with Rev#2 that exploring the role of spartin in other forms of selective autophagy (Rev#2 #9) would provide valuable insights, we do not think it is strictly needed to support the current conclusions. Similarly, testing human disease-linked Spartin mutations would be a significant addition to the paper (#3 Rev#2) but in our view this is not strictly required to support the role of spartin as a lipophagy receptor. Thus, addressing these comments experimentally will not be strictly necessary for reconsideration of the manuscript at this journal.

- ensure that it conforms to our format instructions and publication policies (see below and <https://www.nature.com/nature/for-authors>).
- provide a point-by-point rebuttal to the full referee reports verbatim, as provided at the end of this letter.
- provide the completed Reporting Summary (found here <https://www.nature.com/documents/nr-reporting-summary.pdf>). This is essential for reconsideration of the manuscript will be available to editors and referees in the event of peer review. For more information see <http://www.nature.com/authors/policies/availability.html> or contact me.

When submitting the revised version of your manuscript, please pay close attention to our [href="https://www.nature.com/nature-research/editorial-policies/image-integrity">Digital Image Integrity Guidelines](https://www.nature.com/nature-research/editorial-policies/image-integrity). and to the following points below:

Nature Cell Biology is committed to improving transparency in authorship. As part of our efforts in this direction, we are now requesting that all authors identified as 'corresponding author' on published papers create and link their Open Researcher and Contributor Identifier (ORCID) with their account on the Manuscript Tracking System (MTS), prior to acceptance. ORCID helps the scientific community achieve unambiguous attribution of all scholarly contributions. You can create and link your ORCID from the home page of the MTS by clicking on 'Modify my Springer Nature account'. For more information please visit please visit www.springernature.com/orcid.

This journal strongly supports public availability of data. Please place the data used in your paper into a public data repository, or alternatively, present the data as Supplementary Information. If data can

only be shared on request, please explain why in your Data Availability Statement, and also in the correspondence with your editor. Please note that for some data types, deposition in a public repository is mandatory - more information on our data deposition policies and available repositories appears below.

[Redacted]

We hope that you will find our referees' comments and editorial guidance helpful. Please do not hesitate to contact me if there is anything you would like to discuss. Thank you again for considering NCB for your work.

Best wishes,

Melina

Melina Casadio, PhD
Senior Editor, Nature Cell Biology
ORCID ID: <https://orcid.org/0000-0003-2389-2243>

Reviewers' Comments:

Reviewer #1:

Remarks to the Author:

Summary of the key results

This paper focuses on the function of spartin, a protein encoded by a gene mutated in hereditary spastic paraplegia. It confirms previous observations that spartin can localise to lipid droplets and endolysosomal organelles, but greatly extends these observations by i) defining a subset of lipid droplets that interact with spartin, ii) showing how spartin localises to lipid droplets (via amphipathic helices in the senescence domain), iii) showing that spartin is necessary for lipophagy of lipid droplets, iv) fleshing out the mechanism of this process (including employing a novel assay with Keima fusion proteins) to show that spartin links lipid droplets to the autophagy machinery via UBR-domain-mediated interactions with Lc3 components, v) showing that this biology may be relevant to HSP by demonstrating that human neurons lacking spartin show lipid droplet accumulation, and that expression of dominant negative spartin in mouse cortex does the same.

Originality and significance: if not novel, please include reference

This work is important and highly original. The existence of lipophagy has been known for some time,

but the lipophagy receptors are not known - this work now provides strong evidence that spartin serves this purpose. In addition, the work may be clinically useful by suggesting mechanisms by which spartin disrupts neurons to cause spastic paraplegia.

Data & methodology: validity of approach, quality of data, quality of presentation

The quality of the experimental work is a hallmark of this paper and the authors are to be congratulated on the robustness of their work. A particularly positive feature is that the authors have striven to show the endogenous relevance of their work at every opportunity, rather than relying on over-expression systems. The work is presented very nicely.

Appropriate use of statistics and treatment of uncertainties

For the most part that statistics are appropriate, with one general exception. The authors use paired t-test in many instances where more than two conditions are being compared on a histogram. This is not correct - ANOVA with pairwise comparisons (and post-hoc testing for multiple testing) should be used.

Conclusions: robustness, validity, reliability

Overall I think the conclusions of this paper are robust and reliable.

Suggested improvements: experiments, data for possible revision

The paper is already very convincing, but I have a few suggestions for possible improvements. I regard all of these suggestions as addressing relatively minor points:

1. A surprising facet of this paper is that the lysosomal receptor consisted of autophagy proteins, rather than ESCRT proteins, which would be the natural candidate in view of their endolysosomal localisation and known interaction with spartin. Are the authors sure that the MIT domain plays no role in this process? It would be helpful to nail this point by including MIT-domain deleted (or better F24 mutated) forms of spartin in the rescue experiments for the lysosome-lipid droplet co-localisation assays used in figure 2 (or perhaps using the Keima-livedrop system with rescues of the spartin KO phenotype).
2. The experiments showing interaction between spartin and LC3 components are convincing, but would be made even more convincing if shown with the endogenous proteins.
3. Enhanced figure 3c suggests that some modest interaction between spartin and LC3 is maintained with the construct that expresses only the senescence domain. Can the authors comment/explain, as this doesn't quite fit with the model of only the UBR being involved.
4. In the Keimsa experiments in Figure 3b, I think it would be helpful to carry out a control experiment adding bafilomycin to the WT OA withdrawal condition - this would validate that the colour change is dependent on vacuolar ATPase activity (i.e. acidification of lysosomes).
5. While not strictly necessary (the human neuronal results are pretty convincing), the neuronal LD accumulation would be even more convincing if the authors could show LD accumulation in the brain of the published spartin knock out mouse. If this mouse is not easily available I would not hold up

publication of the important results of this paper.

6. A previous study (Eastman et al, JCB, 2009) suggested that spartin binds to the lipid droplet protein TIP47. Could the authors clarify whether TIP47 binding is involved in the mechanism of recruitment of spartin to LDs that they have described via amphipathic helices in the senescence domain, for example by depleting TIP47 and testing if spartin's recruitment to LDs is reduced? The Eastman study also highlighted the role of ubiquitin ligases, binding to spartin's PPXY motif, in regulating spartin's association with lipid droplets. Could the authors test whether the PPXY motif plays a role in regulating the lipophagy mechanism that they have described?

References: appropriate credit to previous work?

Yes, although the authors might make it clearer that endogenous spartin had already been localised to LDs in reference 14 (Edwards et al)

Clarity and context: lucidity of abstract/summary, appropriateness of abstract, introduction and conclusions

The paper is very nicely written and very clear. The abstract is appropriate.

Reviewer #2:

Remarks to the Author:

Lipophagy is one of selective autophagy and has an essential for lipid mobilization. However, the molecular mechanism regulating lipophagy is not well understood compared to other selective autophagy. Furthermore, the physiological relevance of lipophagy also remains largely elusive. In this study, Chung et al. found that the Troyer syndrome protein spartin as a novel regulator of lipophagy. They found that Spartín localizes on lipid droplets and interacts with autophagosomal proteins LC3A/C to serve as an autophagy receptor. Depletion of Spartín impaired the turnover of lipid droplet as revealed by Keima reporter assay and biochemical analysis of TG level. Moreover, motor neurons derived from spartin KO iPS cells and mouse brains injected with dominant negative spartin showed the accumulation of lipid droplets, suggesting that defect of lipophagy could contribute to Troyer syndrome development. Although the identification of a novel regulator of lipophagy contributes to understanding of lipophagy, the current study did not go beyond the functional analysis of spartin during the turnover of lipid droplets. Considering that spartin has been shown to be localized on lipid droplets (JCB, 2009), I personally doubt that it brings sufficient novelty and conceptual advance which are required for this journal and recommend a more specialized journal to submit. The lack of significance of findings in in vivo lipid metabolism also dampens the enthusiasm of this reviewer to support its publications. The followings are specific comments for authors.

Major

1. Liver and brown adipose tissue play a central role in lipid metabolism. Is spartin essential for lipophagy in liver or brown adipose tissue? Previous studies showed that lipophagy occurs in liver or brown adipose tissue; however, the authors performed most of the experiments using SUM159 breast cancer cells. It is to be determined whether spartin is required for hepatic or adipose lipophagy.

2. Did mice expressing the DN mutant spartin in neurons show any defect in lipid metabolism and

neuronal functions?

3. It is critical to examine actual human SPART mutations impair lipophagy. human SPART mutations affects the localization of spartin on lipid droplets? Author should also check if SPART mutation constructs can rescue the LD turnover in iPS derived spartin KO human motor neuron.

4. The in vivo experiment using the DN mutant is confusing. Although mouse brain is Not incubated with fatty acids, the DN mutant spartin induced LD formation in neurons. Both LD formation and lipophagy constitutively occur in neurons? It is also conceivable that the mutant spartin promotes LD formation with unknown mechanisms. The authors need to clarify if the DN mutant inhibits lipophagy per se in neurons.

5. Because lipophagy is one of cellular "degradation" systems, it is to be determined if spartin is required for LD degradation. The authors performed the radioisotope labeling assays to see TG degradation. To further examine this, the authors should show the detailed kinetics of LD degradation in spartin WT and KO cells using live imaging.

6. Does deletion of spartin impair lipolysis? The authors need to dissect the role of spartin in lipolysis and lipophagy.

7. During lipophagy, ubiquitin or other ATG proteins are recruited onto LDs? The authors claim that the spartin-LC3 interaction mediates lipophagy. But, in other selective autophagy, the ubiquitinated substrates recruit autophagy receptors and ATG proteins. For example, an autophagy receptor NDP52 recruits the ULK1 complex during mitophagy.

8. Observation of conversion from LC-I to LC3-II is not sufficient to exclude the possible involvement of spartin in starvation induced bulk autophagy (Extended data. Fig.2). Authors also need to examine the actual autophagy flux with/without BafilomycinA1.

9. Is spartin only involved in lipophagy? It is worth checking the role of spartin in other selective autophagy such as mitophagy and aggrephagy.

10. The quantification data in Fig.1f, Fig.2e, Fig. 3b is missing.

Minor

1. Some data lack controls. The following controls are to be included: 30min OA in Fig. 1c; Before the OA withdrawal in Fig. 2a; WT cells in Fig. 2h; Before the OA withdrawal in Fig. 3c.

2. In Fig.4f, the label of bottom low is missing (probably Spartin-FL?).

Reviewer #3:

Remarks to the Author:

In this manuscript, Chung et al describe a role for the Troyer syndrome protein spartin as a lipophagy receptor that mediates the turnover of lipid droplets. Spartin is demonstrated by previous studies to target to lipid droplets, the authors show the C-terminal amphipathic helices were required for its

recruitment to lipid droplets. They further demonstrated that show spartin interacts with ATG8 family protein LC3A, potentially through its ubiquitin-binding region (UBR). Knock out of spartin causes deficient lipid droplet turnover in cultured cells. They further show impairing spartin function by overexpression of a dominant negative spartin fragment leads to compromised lipophagy and TG accumulation in cultured human neurons and murine brain neurons. Overall, this study would be of interest for the readership as it would highlight a potential new lipophagy receptor. However, the conclusion of spartin as a lipophagy receptor is not fully supported by the data, more key evidences are needed to clarify the role of spartin in lipophagy.

Major points:

1. The authors investigated the function of spartin in lipophagy mainly by spartin knockout or by overexpressing a dominant negative fragment that with a large truncation of its N-terminal MIT domain and the UBR. It is hard to conclude that spartin acts as a cargo receptor based on the effects caused by either fully knockout or such a big truncation. Specific mutants without disrupting the global domain conformation that block its localization on lipid droplets and its interaction with ATG8 family proteins respectively, are need to determine the role of spartin in the turnover of lipid droplets.
2. Spartin was shown to interact with the Nedd4-family E3 ligase AIP4 and promote the ubiquitination of adipophilin on lipid droplets (Eastman SW, 2009; Hooper C, 2010). Given the authors analyzed the function of spartin in lipophagy mostly using spartin knock out cells, one possibility is that spartin is mainly involved in the ubiquitination of proteins on the lipid-droplet membrane, and the ubiquitinated proteins recruit the SAR family receptors such as p62, NBR1 that further mediate the engulfment of lipid droplet by autophagy. And actually, it was shown in macrophage lipid droplets are decorated with ubiquitin, and SARs including p62, OPTN and NBR could target to ubiquitinated lipid droplets (Karlsson AB, 2014; Robichaud S, 2021). The authors should test whether lipid droplets are ubiquitinated in the neuron system, and test the function of SAR receptors in spartin-mediated lipophagy. The authors need to clarify spartin functions directly as a receptor itself or indirectly by recruiting the SAR receptors.
3. One major evidence shown by the authors that supporting spartin acts as a lipophagy receptor is the interaction between spartin and ATG8 family members LC3A and LC3C. They claimed that the binding between spartin and LC3A is mediated through spartin UBR domain, but not a conventional LC3 interaction motif (LIR). The authors should show the data that the LIR is not required for its binding to LC3A in the manuscript. Have the authors checked whether the LIR is required for its binding with LC3C? Do the authors have particular reasons to pick LC3A? Would it be possible spartin binds to LC3C via the LIR and thus LC3C is the functional partner of spartin? If the consensus LIR motif is not required for its binding for neither LC3A nor LC3C, to prove a specific binding between spartin UBR and LC3A/C, a comprehensive mapping of the interaction or structural analysis that characterizing the interface between spartin UBR and LC3A/C is needed. A specific mutant of spartin disrupts its binding for LC3A/C is needed to be characterized and used to check the role of spartin in lipophagy.
4. To further verify spartin serves as a lipophagy receptor, it is essential to determine the lysosomal turnover of spartin protein, as the cargo receptors are engulfed by the autophagosome and delivered to lysosomes. Although the authors designed a Keima-Spartin reporter, it is important to show the protein degradation by western blot, and show that the spartin degradation is mediated by the autophagy-lysosome pathway. Besides, time course images of Keima-Spartin before OA loading, OA loading and OA withdraw need to be shown to prove the turnover of spartin.

5. The author developed an assay for lipophagy by treating cells with oleic acid first and then remove the oleic acid, it is important to determine whether spartin is generally required for lipophagy induced by other stresses, such as starvation.

Minor points:

1. Does spartin has an effect on lipid droplet biogenesis at basal condition? Although the authors describe that spartin knock out did not affect biogenesis by showing that there was no obvious difference of lipid droplet numbers between WT and KO cells after 30 min OA treatment in Extended Data Fig. 4, but lipid droplet numbers of WT and KO cells before OA treatment were not shown here. Indeed, the lipid droplet number was significantly increased in spartin KO cells compared to the WT cells in Extended Data Fig. 8. These data look counteract, given if spartin does not affect newly lipid droplet formation, after OA treatment, there should be more lipid droplets in KO cells because of the accumulated lipid droplets at basal condition. Otherwise, these data suggest that spartin knock out suppresses lipid droplet biogenesis. The authors need to double check and comment on this.
2. In Fig. 1a and b, the location of spartin under normal condition need to be shown as controls.
3. In Fig. 2a, the quantification data about ratio of spartin that colocalized with lysosomes under OA loading and OA withdraw conditions are needed. Does the colocalization increase upon lipophagy induction?
4. In Fig. 2d, f and g, it looks the binding affinity of GST-LC3A and spartin from cell lysate is much lower than that of GST-LC3A and purified spartin UBR, because 3 nM and 1 uM GST-LC3A protein were used for the pull-down experiments, respectively. Is this a typo? Or does it suggest that modifications of spartin in cells improve its binding with LC3A?
5. In Fig. 3b, images of WT and KO cells with or without OA loading need to be shown.

Methods should be written concisely, but should contain all elements necessary to allow interpretation and replication of the results. As a guideline, Methods sections typically do not exceed 3,000 words. The Methods should be divided into subsections listing reagents and techniques. When citing previous methods, accurate references should be provided and any alterations should be noted. Information must be provided about: antibody dilutions, company names, catalogue numbers and clone numbers for monoclonal antibodies; sequences of RNAi and cDNA probes/primers or company names and catalogue numbers if reagents are commercial; cell line names, sources and information on cell line identity and authentication. Animal studies and experiments involving human subjects must be reported in detail, identifying the committees approving the protocols. For studies involving human subjects/samples, a statement must be included confirming that informed consent was obtained. Statistical analyses and information on the reproducibility of experimental results should be provided in a section titled "Statistics and Reproducibility".

All Nature Cell Biology manuscripts submitted on or after March 21 2016 must include a Data availability statement as a separate section after Methods but before references, under the heading "Data Availability". For Springer Nature policies on data availability see <http://www.nature.com/authors/policies/availability.html>; for more information on this particular policy see <http://www.nature.com/authors/policies/data/data-availability-statements-data-citations.pdf>. The Data availability statement should include:

- Accession codes for primary datasets (generated during the study under consideration and designated as "primary accessions") and secondary datasets (published datasets reanalysed during the study under consideration, designated as "referenced accessions"). For primary accessions data should be made public to coincide with publication of the manuscript. A list of data types for which submission to community-endorsed public repositories is mandated (including sequence, structure, microarray, deep sequencing data) can be found here <http://www.nature.com/authors/policies/availability.html#data>.
- Unique identifiers (accession codes, DOIs or other unique persistent identifier) and hyperlinks for datasets deposited in an approved repository, but for which data deposition is not mandated (see here for details <http://www.nature.com/sdata/data-policies/repositories>).
- At a minimum, please include a statement confirming that all relevant data are available from the authors, and/or are included with the manuscript (e.g. as source data or supplementary information), listing which data are included (e.g. by figure panels and data types) and mentioning any restrictions on availability.
- If a dataset has a Digital Object Identifier (DOI) as its unique identifier, we strongly encourage including this in the Reference list and citing the dataset in the Methods.

We recommend that you upload the step-by-step protocols used in this manuscript to the Protocol Exchange. More details can found at www.nature.com/protocolexchange/about.

All imaging data should be accompanied by scale bars, which should be defined in the legend. Cropped images of gels/blots are acceptable, but need to be accompanied by size markers, and to retain visible background signal within the linear range (i.e. should not be saturated). The boundaries of panels with low background have to be demarked with black lines. Splicing of panels should only be considered if unavoidable, and must be clearly marked on the figure, and noted in the legend with a statement on whether the samples were obtained and processed simultaneously. Quantitative comparisons between samples on different gels/blots are discouraged; if this is unavoidable, it should only be performed for samples derived from the same experiment with gels/blots were processed in parallel, which needs to be stated in the legend.

Regardless of format, all figures must be vector graphic compatible files, not supplied in a flattened

raster/bitmap graphics format, but should be fully editable, allowing us to highlight/copy/paste all text and move individual parts of the figures (i.e. arrows, lines, x and y axes, graphs, tick marks, scale bars etc.). The only parts of the figure that should be in pixel raster/bitmap format are photographic images or 3D rendered graphics/complex technical illustrations.

The total number of Supplementary Figures (not including the “unprocessed scans” Supplementary Figure) should not exceed the number of main display items (figures and/or tables (see our Guide to Authors and March 2012 editorial <http://www.nature.com/ncb/authors/submit/index.html#suppinfo>; <http://www.nature.com/ncb/journal/v14/n3/index.html#ed>). No restrictions apply to Supplementary Tables or Videos, but we advise authors to be selective in including supplemental data.

Each Supplementary Figure should be provided as a single page and as an individual file in one of our

accepted figure formats and should be presented according to our figure guidelines (see above). Supplementary Tables should be provided as individual Excel files. Supplementary Videos should be provided as .avi or .mov files up to 50 MB in size. Supplementary Figures, Tables and Videos must be accompanied by a separate Word document including titles and legends.

GUIDELINES FOR EXPERIMENTAL AND STATISTICAL REPORTING

REPORTING REQUIREMENTS – We are trying to improve the quality of methods and statistics reporting in our papers. To that end, we are now asking authors to complete a reporting summary that collects information on experimental design and reagents. The Reporting Summary can be found here <https://www.nature.com/documents/nr-reporting-summary.pdf> If you would like to reference the guidance text as you complete the template, please access these flattened versions at <http://www.nature.com/authors/policies/availability.html>.

Author Rebuttal to Initial comments

Point-by-point response to reviewers (NCB-LE48087):

Reviewer #1:

- Summary of the key results: This paper focuses on the function of spartin, a protein encoded by a gene mutated in hereditary spastic paraplegia. It confirms previous observations that spartin can localise to lipid droplets and endolysosomal organelles, but greatly extends these observations by i) defining a subset of lipid droplets that interact with spartin, ii) showing how spartin localises to lipid droplets (via amphipathic helices in the senescence domain), iii) showing that spartin is necessary for lipophagy of lipid droplets, iv) fleshing out the mechanism of this process (including employing a novel assay with Keima fusion proteins) to show that spartin links lipid droplets to the autophagy machinery via UBR-domain-mediated interactions with Lc3 components, v) showing that this biology may be relevant to HSP by demonstrating that human neurons lacking spartin show lipid droplet accumulation, and that expression of dominant negative spartin in mouse cortex does the same.

- Originality and significance: This work is important and highly original. The existence of lipophagy has been known for some time, but the lipophagy receptors are not known - this work now provides strong evidence that spartin serves this purpose. In addition, the work may be clinically useful by suggesting mechanisms by which spartin disrupts neurons to cause spastic paraplegia.

- Data & methodology: The quality of the experimental work is a hallmark of this paper and the authors are to be congratulated on the robustness of their work. A particularly positive feature is that the authors have striven to show the endogenous relevance of their work at every opportunity, rather than relying on over-expression systems. The work is presented very nicely.

- Conclusions: Overall I think the conclusions of this paper are robust and reliable.

We thank this reviewer for his/her critical and helpful evaluation of our manuscript. In response to the reviewer's critique, our manuscript has undergone a major revision.

- Appropriate use of statistics and treatment of uncertainties: For the most part that statistics are appropriate, with one general exception. The authors use paired t-test in many instances where more than two conditions are being compared on a histogram. This is not correct - ANOVA with pairwise comparisons (and post-hoc testing for multiple testing) should be used.

We thank the reviewer for this comment and agree. We now provide ANOVA analyses and description for each histogram that has more than two conditions.

Suggested improvements: The paper is already very convincing, but I have a few suggestions for possible improvements. I regard all of these suggestions as addressing relatively minor points:

1. A surprising facet of this paper is that the lysosomal receptor consisted of autophagy proteins, rather than ESCRT proteins, which would be the natural candidate in view of their endolysosomal localisation and known interaction with spartin. Are the authors sure that the MIT domain plays no role in this process? It would be helpful to nail this point by including MIT-domain deleted (or better F24 mutated) forms of spartin in the rescue experiments for the lysosome-lipid droplet co-localisation assays used in figure 2 (or perhaps using the Keima-livedrop system with rescues of the spartin KO phenotype).

We thank the reviewer for bringing up this important point. Following the suggestion from the reviewer, we tested the requirement of spartin's MIT domain lipophagy by analyzing co-localization between LAMP1 and a spartin mutant lacking the MIT domain. Our new results shown in Extended data Figs. 3c and 3d (new addition) reveal that the MIT domain is dispensable for spartin-mediated lipophagy.

2. The experiments showing interaction between spartin and LC3 components are convincing, but would be made even more convincing if shown with the endogenous proteins.

We agree with the reviewer and, although the interactions between LC3 and autophagy receptors are often somewhat weak, we, performed immunoprecipitation analysis between endogenous LC3 and

endogenous spartin. In these experiments, now added in Extended data Fig. 2d (new addition), we observe co-precipitation specifically of spartin with LC3.

3. Extended figure 3c suggests that some modest interaction between spartin and LC3 is maintained with the construct that expresses only the senescence domain. Can the authors comment/explain, as this doesn't quite fit with the model of only the UBR being involved. (biochemistry?)

This is an interesting question. The reviewer is correct that we detect some LC3 protein in this case, albeit much less than with the UBR domain present. We currently do not fully understand this result. It is possible that since the senescence domain still binds to LDs, a pulldown might immunoprecipitate some LDs that interact with LC3 via endogenous spartin.

To further test whether the senescence domain of spartin may function in recruiting LDs to lysosomes, we co-localized spartin and lysosomes in cells. As the reviewer can appreciate from the figure below, the senescence domain does not co-localize with lysosomes, consistent with the interpretation that it cannot function in recruiting LC3.

4. In the Keima experiments in Figure 3b, I think it would be helpful to carry out a control experiment adding bafilomycin to the WT OA withdrawal condition - this would validate that the color change is dependent on vacuolar ATPase activity (i.e. acidification of lysosomes).

Thank you for this suggestion for a control experiment. We agree with the reviewer and performed more validation experiments (including bafilomycin A1 treatment to block lysosomal acidification) to strengthen our findings reported in Figure 3b (Figure 4b in the revised manuscript). This control is now shown in Figure 4c (new addition).

5. While not strictly necessary (the human neuronal results are pretty convincing), the neuronal LD accumulation would be even more convincing if the authors could show LD accumulation in the brain of the published spartin knock out mouse. If this mouse is not easily available, I would not hold up publication of the important results of this paper.

We appreciate the reviewer's comment and we also wish to pursue this. With the generous support from Dr. Craig Blackstone who originally established the Spartin KO mouse line (Renvoisé *et al*, 2012), we are in the early stages of the mouse line recovery. Due to the time constraints and the requirement to age these mice for evaluation (LDs were found at age 3-6 months for Cravatt's lab DDHD2 knockout mice (Inloes *et al*, 2014), we are unable to include the mouse analyses in the current manuscript.

6. A previous study (Eastman et al, JCB, 2009) suggested that spartin binds to the lipid droplet protein TIP47. Could the authors clarify whether TIP47 binding is involved in the mechanism of recruitment of spartin to LDs that they have described via amphipathic helices in the senescence domain, for example by depleting TIP47 and testing if spartin's recruitment to LDs is reduced? The Eastman study also highlighted the role of ubiquitin ligases, binding to spartin's PPXY motif, in regulating spartin's association with lipid droplets. Could the authors test whether the PPXY motif plays a role in regulating the lipophagy mechanism that they have described?

Thank you for the interesting questions. In response, we performed additional experiments to clarify 1) the involvement of PLIN3/TIP47 in LD recruitment of Spartin and 2) the importance of the PPXY motif in lipophagy.

First, to test whether PLIN3/TIP47 is required for spartin localization, we knocked down PLIN3 in PLIN3/spartin double-knock-in cell lines. Spartin enrichment in LDs was not diminished by PLIN3 depletion, rather, there was a slight increase of spartin recruitment to LDs, suggesting that spartin and PLIN3 might compete to target to LD surface using their 3-11 amphipathic helices (Extended Data Fig. 1g, i, h; new addition).

Second, concerning whether the PPAY motif is required for spartin-mediated lipophagy, we examined co-localization between LAMP1 and a spartin mutant lacking the PPAY motif. Our results, shown in Extended Data Fig. 3c and 3d (new addition), indicate that the PPAY motif is not required for spartin-mediated lipophagy as assessed by lysosomal colocalization. Supporting this, we also found that spartin-mediated lipophagy is ubiquitination-independent (Extended Data Fig. 4b, c, d; new addition).

7. References: Yes, although the authors might make it clearer that endogenous spartin had already been localised to LDs in reference 14 (Edwards et al)

Thanks. We have made this clearer.

Reviewer #2:

Remarks to the Author: Lipophagy is one of selective autophagy and has an essential for lipid mobilization. However, the molecular mechanism regulating lipophagy is not well understood compared to other selective autophagy. Furthermore, the physiological relevance of lipophagy also remains largely elusive. In this study, Chung et al. found that the Troyer syndrome protein spartin as a novel regulator of lipophagy. They found that Spartin localizes on lipid droplets and interacts with autophagosomal proteins LC3A/C to serve as an autophagy receptor. Depletion of Spartin impaired the turnover of lipid droplet as revealed by Keima reporter assay and biochemical analysis of TG level. Moreover, motor neurons derived from spartin KO iPS cells and mouse brains injected with dominant negative spartin showed the accumulation of lipid droplets, suggesting that defect of lipophagy could contribute to Troyer syndrome development. Although the identification of a novel regulator of lipophagy contributes to understanding of lipophagy, the current study did not go beyond the functional analysis of spartin during the turnover of lipid droplets. Considering that spartin has been shown to be localized on lipid droplets (JCB, 2009), I personally doubt that it brings sufficient novelty and conceptual advance which are required for this journal and recommend a more specialized journal to submit. The lack of significance of findings in in vivo lipid metabolism also dampens the enthusiasm of this reviewer to support its publications. The followings are specific comments for authors.

We thank this reviewer for her/his critical and helpful evaluation of our manuscript. In response to the reviewer's critique, our manuscript has undergone a major revision.

We respectfully disagree that our paper does not bring sufficient novelty and conceptual advance for this journal. Previous studies (Eastman *et al*, 2009; Hooper *et al*, 2010) showed spartin can co-localize with lipid droplets but they did not demonstrate a functional requirement in lipophagy or show mechanistically how this occurs. Here we identify Spartin as a selective autophagy receptor for lipid droplets (i.e., the first clearly identified lipophagy receptor), we show how spartin mechanistically works in this process, and we demonstrate its functional requirement for TG turnover in cultured cells and neuronal lipid metabolism. Given the very limited mechanistic insights into LD autophagy currently available, the unclear function of spartin, and the emerging role of LDs in the brain (with pertinent links to neurodegenerative hereditary spastic paraplegias), we believe that this study constitutes an important contribution to lipid droplet cell biology and its application to medical questions.

Major points:

1. Liver and brown adipose tissue play a central role in lipid metabolism. Is spartin essential for lipophagy in liver or brown adipose tissue? Previous studies showed that lipophagy occurs in liver or brown adipose tissue; however, the authors performed most of the experiments using SUM159 breast cancer cells. It is to be determined whether spartin is required for hepatic or adipose lipophagy.

We thank the reviewer for these interesting questions. We use SUM159 as a cell culture model enabling rapid progress on mechanistic questions, but of course agree that it will be important to determine the physiological relevance of spartin-mediated lipophagy. Unfortunately, little is yet known about the mechanisms of LD autophagy in any tissue, including liver or brown adipose tissue. Since Spartin is prominently expressed in neurons and in since mutations in this protein lead to a neurological disease, spastic paraplegia, we elected to analyze its function there first. It will be interesting of course to determine whether spartin's role in lipophagy is neuron-specific or also present in other cell types and tissues. Yet, because this would require a series of long experiments in mice, finding the answer to this fascinating question is beyond the scope of this first report of spartin function in lipophagy.

2. Did mice expressing the DN mutant spartin in neurons show any defect in lipid metabolism and neuronal functions?

Yes, we detected neuronal LD accumulations and accumulation of triglyceride lipids in brain samples from mice expressing the DN mutant spartin. This is shown in Figures 5f, 5g, 5h, 5i.

We agree it would be quite interesting to study a variety of neurological functions with spartin deficiency. However, due to the requirements of larger numbers of animals and the complexity of

stereotaxic injections in the mouse motor cortex and uneven DN mutant overexpression, we are unfortunately not able to test neuronal functions adequately in this model. We plan to do such experiments in future experiments with knock-out animals when they become available in the next year.

3. It is critical to examine actual human SPART mutations impair lipophagy. human SPART mutations affects the localization of spartin on lipid droplets? Author should also check if SPART mutation constructs can rescue the LD turnover in iPS derived spartin KO human motor neuron.

We agree that this is an important and interesting question. Almost all pathogenic mutations in spartin are nonsense mutations that are predicted to result spartin lacking the senescence (lipid droplet-binding) domain. Thus, if pathogenic mutants are expressed, they would show similar behavior of Spartin (1-380) that we included in Figure 1f. To further clarify this point, we now also performed the localization analysis of the patient mutant (Spartin-1110delA). As expected, Spartin-1110delA does not localize to LDs (Extended Data Fig. 1k; new addition). This was also previously reported in Eastman et al., JCB 2009(Eastman *et al*, 2009).

4. The in vivo experiment using the DN mutant is confusing. Although mouse brain is not incubated with fatty acids, the DN mutant spartin induced LD formation in neurons. Both LD formation and lipophagy constitutively occur in neurons? It is also conceivable that the mutant spartin promotes LD formation with unknown mechanisms. The authors need to clarify if the DN mutant inhibits lipophagy per se in neurons.

We appreciate the reviewer's concern and share the view that currently, LD biology in neurons is not well understood. Our data, including new experiments measuring the cellular TG biosynthesis rate in cells expressing the DN mutant spartin (Extended Data Fig. 9c) show that impairment of spartin function interferes with TG clearance, not TG synthesis for LD formation. Thus, we believe the most parsimonious explanation of our results for dominant-negative spartin is that there is some level of lipophagy in wildtype neurons, and when this is interfered with, TG-rich LDs accumulate. We note that although neurons do not normally exhibit many LDs at baseline, this appears to be due to a high turnover rate and resultant low pool size. There are now two studies, one by the Cravatt lab and ours that show that if TG clearance is interfered with, TGs and LDs accumulate. This suggests high flux through a small pool of TG.

5. Because lipophagy is one of cellular "degradation" systems, it is to be determined if spartin is required for LD degradation. The authors performed the radioisotope labeling assays to see TG degradation. To further examine this, the authors should show the detailed kinetics of LD degradation in spartin WT and KO cells using live imaging.

We appreciate the reviewer's suggestion. In response, we now included more detailed analyses of LD degradation rates in WT and spartin KO cells, showing delayed LD turnover in spartin deficiency (Figure 4g and 4h; new addition).

6. Does deletion of spartin impair lipolysis? The authors need to dissect the role of spartin in lipolysis and lipophagy.

We thank the reviewer for this important comment. To test a spartin function in lipolysis, we examined the LD degradation rate in the presence and absence of a lipase ATGL. The result previously shown in Figure 4j (Figure 3g in original submission) demonstrated that spartin does not affect lipolysis-mediated clearance of TG.

7. During lipophagy, ubiquitin or other ATG proteins are recruited onto LDs? The authors claim that the spartin-LC3 interaction mediates lipophagy. But, in other selective autophagy, the ubiquitinated substrates recruit autophagy receptors and ATG proteins. For example, an autophagy receptor NDP52 recruits the ULK1 complex during mitophagy.

We thank the reviewer for the comment and have performed additional experiments to investigate involvement of ubiquitination or other ATG proteins in spartin-mediated lipophagy. To test whether ubiquitination of spartin or other ubiquitinated LD proteins mediate lipophagy, we blocked the ubiquitination pathway using a small molecular inhibitor of the ubiquitin activating enzymes TAK-243(Hyer *et al*, 2018). We verified successful inhibition of ubiquitination reactions (Extended Data Fig. 4b; new addition), and found that LD delivery (labeled by LiveDrop) to lysosomes was not largely affected by E1 inhibition (Extended Data Fig. 4c, d; new addition). We therefore think that ubiquitination does not play a major role in spartin-mediated autophagy. We have added this to the discussion.

Furthermore, to address an involvement of known selective autophagy receptors (SARs) in spartin-mediated lipophagy, we knocked down three key SARs (NBR1, OPTN, and SQSTM1) and examined lipophagy flux using our Keima-LiveDrop reporter system. As now shown in Extended Data Fig. 4e, f (new addition), the lipophagy pathway was not affected by knockdown of these SARs.

Taken together, along with our data showing direct protein interactions with LDs and the autophagy machinery, these results suggest that spartin directly mediates lipophagy reaction as a lipophagy receptor protein.

8. Observation of conversion from LC-I to LC3-II is not sufficient to exclude the possible involvement of spartin in starvation induced bulk autophagy (Extended data. Fig.2). Authors also need to examine the actual autophagy flux with/without BafilomycinA1.

We thank the reviewer for this comment, agree and have performed additional experiments using a Keima-LC3B autophagy reporter system. New results show that spartin appears not to be involved in bulk autophagy (Extended Data Fig. 5b; new addition).

9. Is spartin only involved in lipophagy? It is worth checking the role of spartin in other selective autophagy such as mitophagy and aggrephagy.

This is an intriguing idea, and we thank the reviewer for suggesting this interesting possibility. Spartin appears not to be involved in bulk autophagy, and we were not yet able to systematically test a function for spartin in other selective autophagy pathways but will examine this in future studies.

10. The quantification data in Fig.1f, Fig.2e, Fig. 3b is missing.

We thank the reviewer for the comment, apologize for the oversight and have added the quantification data of Fig. 1f (added on the bottom of panel), Fig. 2e (Figure 3h), and Fig. 3b (Figure 4c).

Minor

1. Some data lack controls. The following controls are to be included: 30min OA in Fig. 1c; Before the OA withdrawal in Fig. 2a; WT cells in Fig. 2h; Before the OA withdrawal in Fig. 3c.

We appreciate this comment. All requested controls were added in the new figures: 30 min OA in Fig. 1c (original Extended Data Fig. 1d); Before the OA withdrawal in Fig. 2a (Extended Data Fig. 2c and Fig. 2b); WT cells in Fig. 2h (Figure 2f top panel); Before the OA withdrawal in Fig. 3c (Figure 4d).

2. In Fig.4f, the label of bottom low is missing (probably Spartin-FL?).

Thank you for catching this! We corrected the label in new Figure 5f (original Figure 4f).

Reviewer #3:

Remarks to the Author: In this manuscript, Chung et al describe a role for the Troyer syndrome protein spartin as a lipophagy receptor that mediates the turnover of lipid droplets. Spartin is demonstrated by previous studies to target to lipid droplets, the authors show the C-terminal amphipathic helices were required for its recruitment to lipid droplets. They further demonstrated that show spartin interacts with ATG8 family protein LC3A, potentially through its ubiquitin-binding region (UBR). Knock out of spartin causes deficient lipid droplet turnover in cultured cells. They further show impairing spartin function by overexpression of a dominant negative spartin fragment leads to compromised lipophagy and TG accumulation in cultured human neurons and murine brain neurons. Overall, this study would be of interest for the readership as it would highlight a potential new lipophagy receptor. However, the conclusion of spartin as a lipophagy receptor is not fully supported by the data, more key evidences are needed to clarify the role of spartin in lipophagy.

We thank this reviewer for his/her critical and helpful evaluation of our manuscript. In response to the reviewer's critique, our manuscript has undergone a major revision.

Major points:

1. The authors investigated the function of spartin in lipophagy mainly by spartin knockout or by overexpressing a dominant-negative fragment that with a large truncation of its N-terminal MIT domain and the UBR. It is hard to conclude that spartin acts as a cargo receptor based on the effects caused by either fully knockout or such a big truncation. Specific mutants without disrupting the global domain conformation that block its localization on lipid droplets and its interaction with ATG8 family proteins respectively, are needed to determine the role of spartin in the turnover of lipid droplets.

We thank the reviewer for this comment and agree. We added a substantial amount of new data that have strengthened our conclusions. Specifically, we added data to show that a mutation that disrupts Spartin binding to LC3 impairs lipophagy in our model systems.

For this, we determined the motif in spartin responsible for LD targeting. Initially, we narrowed down the spartin region required for LD binding (original Figure 1f). Once we identified the repeats of 3-11 amphipathic helices in the senescence domain, we identified point mutations of hydrophobic residues changed to alanine in 3-11 helices in the context of full-length spartin (current Figure 1h and 1i). The results indicate the importance of the amphipathic helices in mediating LD binding.

We now also map a LIR motif within UBR domain. Deletion of the LIR motif (residue 193-200) in full-length spartin abolished co-localization of spartin and LC3a (Figure 3g and 3h). In addition, a recombinant UBR- Δ LIR impaired the interaction of LC3A and spartin (Figure 3F).

2. Spartin was shown to interact with the Nedd4-family E3 ligase AIP4 and promote the ubiquitination of adipophilin on lipid droplets (Eastman SW, 2009; Hooper C, 2010). Given the authors analyzed the function of spartin in lipophagy mostly using spartin knock out cells, one possibility is that spartin is mainly involved in the ubiquitination of proteins on the lipid-droplet membrane, and the ubiquitinated proteins recruit the SAR family receptors such as p62, NBR1 that further mediate the engulfment of lipid droplet by autophagy. And actually, it was shown in macrophage lipid droplets are decorated with ubiquitin, and SARs including p62, OPTN and NBR could target to ubiquitinated lipid droplets (Karlsson AB, 2014; Robichaud S, 2021). The authors should test whether lipid droplets are ubiquitinated in the neuron system, and test the function of SAR receptors in spartin-mediated lipophagy. The authors need to clarify spartin functions directly as a receptor itself or indirectly by recruiting the SAR receptors.

We appreciate the suggestions and have performed additional experiments to investigate involvements of ubiquitination or other ATG proteins in spartin-mediated lipophagy.

To test whether ubiquitination of spartin or other ubiquitinated LD proteins mediate lipophagy, we blocked the ubiquitination pathway using a small molecular inhibitor of the ubiquitin activating enzymes TAK-243 (Hyer *et al*, 2018). Inhibition of the ubiquitination reaction (Extended Data Fig. 4b; new addition), did not impair LD delivery (labeled by LiveDrop) to lysosomes (Extended Data Fig. 4c, d; new addition).

To address an involvement of selective autophagy receptors (SARs) in spartin-mediated lipophagy, we knocked down three key SARs (NBR1, OPTN, and SQSTM1) and examined lipophagy flux using Keima-LiveDrop reporter system. As shown in Extended Data Fig. 4e, f (new addition), lipophagy was not affected by knockdown of these SARs.

We appreciate the suggestion to examine ubiquitination of LDs in the nervous system. However, we no longer have the fractions to analyze this. Additionally, since our new data suggest ubiquitination is not necessary for spartin-mediated lipophagy, we feel the results will not change the major conclusion and don't warrant the additional months of experiments.

3. One major evidence shown by the authors that supporting spartin acts as a lipophagy receptor is the interaction between spartin and ATG8 family members LC3A and LC3C. They claimed that the binding between spartin and LC3A is mediated through spartin UBR domain, but not a conventional LC3 interaction motif (LIR). The authors should show the data that the LIR is not required for its binding to LC3A in the manuscript. Have the authors checked whether the LIR is required for its binding with LC3C? Do the authors have particular reasons to pick LC3A? Would it be possible spartin binds to LC3C via the LIR and thus LC3C is the functional partner of spartin?

If the consensus LIR motif is not required for its binding for neither LC3A nor LC3C, to prove a specific binding between spartin UBR and LC3A/C, a comprehensive mapping of the interaction or structural analysis that characterizing the interface between spartin UBR and LC3A/C is needed. A specific mutant of spartin disrupts its binding for LC3A/C is needed to be characterized and used to check the role of spartin in lipophagy.

We thank the reviewer for the comment and have performed additional experiments to map the interaction between spartin and LC3. Our previous attempts to find a LIR motif using various prediction tools were unsuccessful. We now utilized Alphafold2 (ColabFold) to take account a tertiary structure of spartin and improve a prediction of interaction interface between Spartin-UBR and LC3 (new Figure 3e; new addition). All five top ranked prediction suggested a similar interaction interface, pointing an importance of residues 193-200 in the UBR domain. We validated that indeed 193-200 acts as the LIR motif using imaging and in vitro pull-down analyses (new Figures 3f, g, h; new addition).

We appreciate the intriguing idea for LC3C. There was no particular reason to pick LC3A for further analyses, and thus, we performed additional pull-down experiment to clarify direct interaction between Spartin-UBR and LC3A, LC3B, and LC3C. Consistent with our cell lysate-immunoprecipitation analysis, both recombinant LC3A and LC3C interact with recombinant Spartin-UBR (new Figure 3b). These data have been added to the manuscript.

4. To further verify spartin serves as a lipophagy receptor, it is essential to determine the lysosomal turnover of spartin protein, as the cargo receptors are engulfed by the autophagosome and delivered to lysosomes. Although the authors designed a Keima-Spartin reporter, it is important to show the protein degradation by western blot, and show that the spartin degradation is mediated by the autophagy-lysosome pathway. Besides, time course images of Keima-Spartin before OA loading, OA loading and OA withdraw need to be shown to prove the turnover of spartin.

We thank the reviewer for bringing up this important point, which we addressed experimentally. First, we tested degradation of spartin by autophagy using western blot analyses (Data Extended Fig. 4a). We used Baf A1 treatment and ATG7 KO as controls to validate selective degradation of spartin and find that spartin is indeed degraded by a lysosome-dependent process. Second, we monitored Keima-

Spartin during a time course and representative images of ‘before OA loading’, ‘OA loading’, and ‘OA withdrawal’ are now included in new Figure 4d.

5. The author developed an assay for lipophagy by treating cells with oleic acid first and then remove the oleic acid, it is important to determine whether spartin is generally required for lipophagy induced by other stresses, such as starvation.

We agree with the reviewer and, in response, performed Keima-Spartin flux measurement to address which cellular stresses requires spartin-mediated lipophagy activity. As shown in Extended Data Fig. 6 (new addition), spartin mediates lipophagy in various metabolic stresses, but lipid deficiency activates spartin-mediated lipophagy most. This suggests a possibility of distinct cellular mechanism activating lipophagy apart from bulk autophagy.

Minor points:

1. Does spartin has an effect on lipid droplet biogenesis at basal condition? Although the authors describe that spartin knock out did not affect biogenesis by showing that there was no obvious difference of lipid droplet numbers between WT and KO cells after 30 min OA treatment in Extended Data Fig. 4, but lipid droplet numbers of WT and KO cells before OA treatment were not shown here. Indeed, the lipid droplet number was significantly increased in spartin KO cells compared to the WT cells in Extended Data Fig. 8. These data look counteract, given if spartin does not affect newly lipid droplet formation, after OA treatment, there should be more lipid droplets in KO cells because of the accumulated lipid droplets at basal condition. Otherwise, these data suggest that spartin knock out suppresses lipid droplet biogenesis. The authors need to double check and comment on this.

We thank the reviewer for this comment and we have clarified the apparent discrepancy. The two datasets in Extended Data Fig. 5c and Extended Data Fig. 9b cannot be directly compared since we transiently overexpressed mScarlet-I only in the Extended Data Fig. 9b. Transient protein expression tends to elevate basal LD levels (as a part of cellular stresses), compared with the normal situation of Extended Data Fig. 5c. Thus, spartin KO would set higher baseline of LDs in mScarlet-I-overexpressed condition since lipophagy is defective. We have clarified this in the revised version of our manuscript. Of course, we also cannot rule out a possibility of newly synthesized TGs incorporated into pre-existing LDs in Extended Data Fig. 5b since we are only counting the number of LDs.

2. In Fig. 1a and b, the location of spartin under normal condition need to be shown as controls.

Thank you for the suggestion. We added the location of spartin under normal condition in Extended Data Fig. 1d (new addition).

3. In Fig. 2a, the quantification data about ratio of spartin that colocalized with lysosomes under OA loading and OA withdraw conditions are needed. Does the colocalization increase upon lipophagy induction?

We added the quantification data for the ratio of spartin that colocalized with lysosomes under OA loading and OA withdrawal in Extended Data Fig. 2c and Fig. 2b (new addition). We observed that association between spartin and lysosomes increased upon lipophagy induction.

4. In Fig. 2d, f and g, it looks the binding affinity of GST-LC3A and spartin from cell lysate is much lower than that of GST-LC3A and purified spartin UBR, because 3 nM and 1 uM GST-LC3A protein were used for the pull-down experiments, respectively. Is this a typo? Or does it suggest that modifications of spartin in cells improve its binding with LC3A?

Thank you for the comment, but we think that binding affinity in two conditions are difficult to compare since we cannot estimate the concentration of spartin in our ‘cell lysate’ experiments. It is also possible that modifications of spartin in cells affects binding, as suggested. To address this will require more, future investigation.

5. In Fig. 3b, images of WT and KO cells with or without OA loading need to be shown.

We appreciate the reviewer's comment. We performed the additional requested experiments, with new quantification data added in new Figure 4c.

References

1. Eastman SW, Yassaee M & Bieniasz PD (2009) A role for ubiquitin ligases and Spartin/SPG20 in lipid droplet turnover. *The Journal of Cell Biology* 184: 881–894
2. Hooper C, Puttamadappa SS, Loring Z, Shekhtman A & Bakowska JC (2010) Spartin activates atrophin-1-interacting protein 4 (AIP4) E3 ubiquitin ligase and promotes ubiquitination of adipophilin on lipid droplets. *Bmc Biol* 8: 72
3. Hyer ML, Milhollen MA, Ciavarrri J, Fleming P, Traore T, Sappal D, Huck J, Shi J, Gavin J, Brownell J, *et al* (2018) A small-molecule inhibitor of the ubiquitin activating enzyme for cancer treatment. *Nat Med* 24: 186–193
4. Inloes JM, Hsu K-L, Dix MM, Viader A, Masuda K, Takei T, Wood MR & Cravatt BF (2014) The hereditary spastic paraplegia-related enzyme DDHD2 is a principal brain triglyceride lipase. *Proc National Acad Sci* 111: 14924–14929
5. Renvoisé B, Stadler J, Singh R, Bakowska JC & Blackstone C (2012) Spg20^{-/-} mice reveal multimodal functions for Troyer syndrome protein spartin in lipid droplet maintenance, cytokinesis and BMP signaling. *Hum Mol Genet* 21: 3604–3618

Decision Letter, first revision:

Dear Tobi,

Thank you for submitting your revised manuscript "The Troyer syndrome protein spartin mediates selective autophagy of lipid droplets" to the journal. It has now been seen by the original reviewers, whose comments are pasted below. In light of their advice, we regret that we cannot offer to publish the study in Nature Cell Biology.

As you will see, the reviewers appreciated the significant revisions to bolster the core conclusions linking spartin to lipophagy. While the reviewers continued to find this work interesting, and although Reviewer #1 remains especially positive (and raises points that could likely be addressed in a straightforward manner), Reviewers #2 and #3 have overlapping, persisting concerns that the key claim of showing that spartin acts as a lipophagy receptor has not been sufficiently demonstrated. We have discussed the points from these reviewers in depth editorially and find them significant. Importantly, we are not sure that these points would be straightforward to address in a final minor revision. As we make every effort to limit all papers to one round of major experimental revision, we must regrettably return the manuscript to you.

We are very sorry that we could not be more positive on this occasion, but we thank you for the opportunity to consider this work.

With kind regards,
Melina

Melina Casadio, PhD
Senior Editor, Nature Cell Biology
ORCID ID: <https://orcid.org/0000-0003-2389-2243>

Reviewers' comments:

Reviewer #1 (Remarks to the Author):

Thank you to the authors for addressing my comments so diligently. All of my previous remarks have been dealt with in a satisfactory way. In my initial review I said:

"This work is important and highly original. The existence of lipophagy has been known for some time, but the lipophagy receptors are not known - this work now provides strong evidence that spartin serves this purpose. In addition, the work may be clinically useful by suggesting mechanisms by which spartin disrupts neurons to cause spastic paraplegia"

....and my enthusiasm for this work has been enhanced by the revisions that have been made. I find the story convincing, novel and a significant advance for the field.

I have two very minor remaining comments and one more significant one:

1. Between lines 90 and 95 the text on the Troyer syndrome-related frameshift mutation are somewhat misleading. Such mutations are almost certainly subject to nonsense mediated decay (Craig Blackstone published a paper validating this) and so it is likely that no protein is produced. I think this should be made clear to avoid misunderstanding.
2. In extended data figure 2d, tubulin is spelled incorrectly throughout.
3. The more significant comment relates to the unit of repeat used for "n" values in the presentation and stats for the microscopy experiments (and many apologies for not picking this up on my first review and so introducing this at a late stage). The "n" used is "number of fields". While this is an improvement on the "n=number of cells" that is sometimes (and almost always incorrectly) used, it does raise the question of whether these can be considered true biological repeats. While I find the results convincing as they are demonstrated through multiple orthogonal approaches, it would be comforting to know for the key experiments that the data arose from more than one (preferably at least 3) experiments, with the "n" used being number of biological repeats. This is perhaps a philosophical question for the journal editor to decide on their preferred approach.

Reviewer #2 (Remarks to the Author):

The authors responded to part of my concerns I raised for the original manuscript in a satisfactory manner. However, there remains some critical points that should be resolved for accepting publication of the manuscript in the NCB.

Regarding comment #1, in most main figures (from Fig. 1 to Fig. 4), the authors used SUM159 as a cell culture model to show the mechanism by which Spartin acts as a lipophagy receptor. While many previous studies demonstrated that lipophagy occurs in liver or brown adipocytes (R Singh et al., Nature 2009; N Martinez-Lopez et al., Cell Metab 2016; MB Schott et al., JCB 2019), there is no report showing that lipophagy can occur in neurons. Therefore, to show the physiological relevance of these findings, the authors should determine if Spartin is required for lipophagy in liver or brown adipocytes. Or, the authors must show that lipophagy occurs also in neurons. As noted in comment #4, there is no evidence showing that Spartin-DN inhibits lipophagy, even in SUM159 cells. Because of the lack of this evidence, it is still undetermined if the LD accumulation by the DN mutant is due to loss of lipophagy, i.e., we still don't know if lipophagy occurs in neurons or not.

Reviewer #3 (Remarks to the Author):

The authors have addressed or responded to most of my comments raised in the previous round of review.

However, the authors did not pick up from my previous review was that 'A specific mutant of spartin disrupts its binding for LC3 is needed to be used to check the role of spartin in lipophagy'. It was nice that the authors characterized a LIR mutant disrupted spartin binding to LC3, but they did not provide evidences that this mutant would impair lipophagy. It would be essential to check whether LIR mutant could rescue LD turnover in spartin KO cells. Such evidence is needed to support spartin acts as a lipophagy receptor.

Minor point: Extended data Fig.2c was not showing the quantification data about ratio of spartin that colocalized with lysosomes as indicated by the authors.

**Although we cannot publish your paper, it may be appropriate for another journal in the Nature Portfolio. If you wish to explore the journals and transfer your manuscript please use our manuscript transfer portal. You will not have to re-supply manuscript metadata and files, but please note that this link can only be used once and remains active until used. For more information, please see our manuscript transfer FAQ page.

Note that any decision to opt in to In Review at the original journal is not sent to the receiving journal on transfer. You can opt in to In Review at receiving journals that support this service by choosing to modify your manuscript on transfer. In Review is available for primary research manuscript types only.

**For Nature Portfolio general information and news for authors, see <http://npg.nature.com/authors>.

Author Rebuttal, first revision:

Point-by-point response to reviewers (NCB-LE48087B-Z):

Reviewer #1:

Thank you to the authors for addressing my comments so diligently. All of my previous remarks have been dealt with in a satisfactory way. In my initial review I said:

"This work is important and highly original. The existence of lipophagy has been known for some time, but the lipophagy receptors are not known - this work now provides strong evidence that spartin serves this purpose. In addition, the work may be clinically useful by suggesting mechanisms by which spartin disrupts neurons to cause spastic paraplegia"

....and my enthusiasm for this work has been enhanced by the revisions that have been made. I find the story convincing, novel and a significant advance for the field.

We thank this reviewer for his/her critical and positive evaluation of our manuscript.

I have two very minor remaining comments and one more significant one:

1. Between lines 90 and 95 the text on the Troyer syndrome-related frameshift mutation are somewhat misleading. Such mutations are almost certainly subject to nonsense mediated decay (Craig Blackstone published a paper validating this) and so it is likely that no protein is produced. I think this should be made clear to avoid misunderstanding.

We thank the reviewer for this comment and agree. We now added further explanation about 1110delA mutant and a new reference to the manuscript.

2. In extended data figure 2d, tubulin is spelled incorrectly throughout.

Thank you for catching up the misspelling. We have corrected the error.

3. The more significant comment relates to the unit of repeat used for "n" values in the presentation and stats for the microscopy experiments (and many apologies for not picking this up on my first review and so introducing this at a late stage). The "n" used is "number of fields". While this is an improvement on the "n=number of cells" that is sometimes (and almost always incorrectly) used, it does raise the question of whether these can be considered true biological repeats. While I find the results convincing as they are demonstrated through multiple orthogonal approaches, it would be comforting to know for the key experiments that the data arose from more than one (preferably at least 3) experiments, with the "n" used being number of biological repeats. This is perhaps a philosophical question for the journal editor to decide on their preferred approach.

We thank the reviewer for bringing up this important point. The key microscopy experiments showed in the manuscript were repeated at least three times to ensure reproducibility. To clarify, we now state the number of biological replicates used for the quantification of each experiment in the figure legend.

Reviewer #2:

The authors responded to part of my concerns I raised for the original manuscript in a satisfactory manner. However, there remains some critical points that should be resolved for accepting publication of the manuscript in the NCB.

Regarding comment #1, in most main figures (from Fig. 1 to Fig. 4), the authors used SUM159 as a cell culture model to show the mechanism by which Spartine acts as a lipophagy receptor. While many previous studies demonstrated that lipophagy occurs in liver or brown adipocytes (R Singh et al., Nature 2009; N Martinez-

Lopez et al., Cell Metab 2016; MB Schott et al., JCB 2019), there is no report showing that lipophagy can occur in neurons. Therefore, to show the physiological relevance of these findings, the authors should determine if Spartin is required for lipophagy in liver or brown adipocytes. Or, the authors must show that lipophagy occurs also in neurons. As noted in comment #4, there is no evidence showing that Spartin-DN inhibits lipophagy, even in SUM159 cells. Because of the lack of this evidence, it is still undetermined if the LD accumulation by the DN mutant is due to loss of lipophagy, i.e., we still don't know if lipophagy occurs in neurons or not.

We thank this reviewer for her/his helpful evaluation of our manuscript.

However, we respectfully disagree with this reviewer on the request for liver or BAT evaluation of spartin function in this paper on several grounds. First, we discovered spartin as a lipophagy receptor and showed its relevance in the one cell type where it causes disease – neurons. This is a discovery, and discoveries don't often have much precedent. We believe to ask for data on spartin function in BAT or liver, while possibly interesting, is less relevant than the cell type affected by a human disease. Second, in contrast to what the reviewer states, there are several published papers that show lipophagy occurs in neurons (e.g., PMID 21803288; PMID 33713908). We now cited these articles in the revised manuscript.

Reviewer #3:

The authors have addressed or responded to most of my comments raised in the previous round of review.

However, the authors did not pick up from my previous review was that 'A specific mutant of spartin disrupts its binding for LC3 is needed to be used to check the role of spartin in lipophagy'. It was nice that the authors characterized a LIR mutant disrupted spartin binding to LC3, but they did not provide evidences that this mutant would impair lipophagy. It would be essential to check whether LIR mutant could rescue LD turnover in spartin KO cells. Such evidence is needed to support spartin acts as a lipophagy receptor.

We thank this reviewer for her/his helpful evaluation of our manuscript.

We appreciate the reviewer's clarification and apologize our misunderstanding of his/her comment. We now add new experimental data showing that the spartin mutants disrupting its LD localization or its binding for LC3 (LIR motif deletion) do not rescue the lipophagy defects (measured by Keima-LiveDrop reporter) in spartin KO cells (Fig. 4d, e). Further, we show that these mutants were not able to rescue the LD accumulation found in spartin KO cells, compared with spartin WT cells (Extended Data Fig. 3f, g).

Minor point: Extended data Fig.2c was not showing the quantification data about ratio of spartin that colocalized with lysosomes as indicated by the authors.

During our last assessment of revision, we moved the quantification data to main figure (Fig. 2b). We apologize for this error.

Decision Letter, second revision:

Dear Tobi,

Thank you for submitting your appeal and revised manuscript, "The Troyer syndrome protein spartin mediates selective autophagy of lipid droplets", and thank you very much for your patience with the re-review process. I sincerely apologize for the very long delay.

We have editorially assessed your responses to Rev#1's points and found them very helpful to resolve this referee's final points.

Unfortunately, Rev#3 was not available to re-review the study. As we were interested in expert input to evaluate the LIR-mutant data, and given that this falls within the expertise of our general selective autophagy expert Rev#2, we asked Rev#2 (who agreed to re-review) to evaluate those data.

As you will see from their comments (attached below), they found the new data provided in response to Reviewer #3 fully satisfactory and convincing. Rev#2 also shared their view on a final experiment needed for publication, which we believe should be addressed before we can consider publication in Nature Cell Biology.

We discussed the reviewer's request in great detail editorially. Reviewer #2 feels that it's important to establish that the spartin-DN mutant blocks lipophagy using the lipophagy reporter in SUM159 cells - which would elegantly tie in the in vivo data using the mutant and the in vitro studies establishing that spartin controls lipophagy in this cell model. This comment relates to Rev#2's longstanding question as to whether there is enough evidence supporting the concept that lipophagy occurs in neurons, and whether the effects of spartin in vivo in Figure 5 are therefore dependent on lipophagy. This is not an issue that is new to this round of review.

As mentioned, we have been discussing the need for this experiment editorially, and we strongly recommend carrying out the experiment suggested by Rev#2 in SUM159 cells. In the event that you are unable to provide these data, please do let me know and we'll be happy to discuss, as always.

As typically, in revision, please also pay close attention to our guidelines on statistical and methodological reporting (listed below), as failure to do so may delay the reconsideration of the revised manuscript. In particular, please provide:

- a Supplementary Figure including unprocessed images of all gels/blots in the form of a multi-page pdf file. Please ensure that blots/gels are labeled and the sections presented in the figures are clearly indicated.
- a Supplementary Table including all numerical source data in Excel format, with data for different figures provided as different sheets within a single Excel file. The file should include source data giving rise to graphical representations and statistical descriptions in the paper and for all instances where the figures present representative experiments of multiple independent repeats, the source data of all repeats should be provided.

We therefore invite you to take these points into account when revising the manuscript. In addition, when preparing the revision please:

- ensure that it conforms to our format instructions and publication policies (see below and www.nature.com/nature/authors/).
- provide a point-by-point rebuttal to the full referee reports verbatim, as provided at the end of this letter.
- provide the completed Editorial Policy Checklist (found here <https://www.nature.com/authors/policies/Policy.pdf>), and Reporting Summary (found here <https://www.nature.com/authors/policies/ReportingSummary.pdf>). This is essential for reconsideration of the manuscript and these documents will be available to editors and referees in the event of peer review. For more information see <http://www.nature.com/authors/policies/availability.html> or contact me.

Nature Cell Biology is committed to improving transparency in authorship. As part of our efforts in this direction, we are now requesting that all authors identified as 'corresponding author' on published papers create and link their Open Researcher and Contributor Identifier (ORCID) with their account on the Manuscript Tracking System (MTS), prior to acceptance. ORCID helps the scientific community achieve unambiguous attribution of all scholarly contributions. You can create and link your ORCID from the home page of the MTS by clicking on 'Modify my Springer Nature account'. For more information please visit www.springernature.com/orcid.

[Redacted]

We would like to receive the revision within four weeks. If submitted within this time period, reconsideration of the revised manuscript will not be affected by related studies published elsewhere, or accepted for publication in Nature Cell Biology in the meantime. We would be happy to consider a revision even after this timeframe, but in that case we will consider the published literature at the time of resubmission when assessing the file.

I apologize once again for the long delay in communicating our decision to you. We hope that you will find our referees' comments and editorial guidance helpful. Please do not hesitate to contact me if there is anything you would like to discuss.

Best wishes,

Melina

Melina Casadio, PhD
Senior Editor, Nature Cell Biology
ORCID ID: <https://orcid.org/0000-0003-2389-2243>

Reviewers' Comments:

Reviewer #2:

Remarks to the Author:

This reviewer is convinced with most of the data shown in the revised manuscript. But, there still remains one point to needs be evaluated.

The authors should show lipophagy occurs in neurons. Because a lipophagy receptor is previously undetermined, the two publications (PMCID 21803288; PMCID 33713908) lack the actual evidence showing that lipophagy (LD-specific autophagy) can occur in neurons.

Therefore, the authors should determine if Spartin-DN causes LD accumulation via inhibiting lipophagy in neurons. To do this, the authors can express Spartin-DN in SUM159 cells to easily see whether the mutant inhibits lipophagy in a Keima-LiveDrop system. In the manuscript, the authors just showed Spartin-DN caused LD accumulation, which does not directly indicate that Spartin-DN inhibits lipophagy. Once the paper is published, researchers must use Spartin-DN as a lipophagy inhibitory factor. That's why the authors need to examine if the mutant truly inhibits lipophagy.

COMMENTS ABOUT THE RESPONSES TO REV#3'S POINTS:

I have read the response to reviewer #3 and agree that the authors have adequately addressed the concerns raised by reviewer #3 in this round. The authors showed that lipophagic activity was lost when spartin lacks its LD localization or its binding to LC3 family proteins. I think reviewer #3 would be convinced. I am convinced that spartin acts as a lipophagy receptor. The only thing the authors need to do is to test whether the dominant negative mutant suppresses lipophagy in the Keima LiveDrop system.

GUIDELINES FOR SUBMISSION OF NATURE CELL BIOLOGY ARTICLES

READABILITY OF MANUSCRIPTS – Nature Cell Biology is read by cell biologists from diverse backgrounds, many of whom are not native English speakers. Authors should aim to communicate their findings clearly, explaining technical jargon that might be unfamiliar to non-specialists, and avoiding non-standard abbreviations. Titles and abstracts should concisely communicate the main findings of the study, and the background, rationale, results and conclusions should be clearly explained in the manuscript in a manner accessible to a broad cell biology audience. Nature Cell

Biology uses British spelling.

ARTICLE FORMAT

ABSTRACT – should not exceed 150 words and should be unreferenced. This paragraph is the most visible part of the paper and should briefly outline the background and rationale for the work, and accurately summarize the main results and conclusions. Key genes, proteins and organisms should be specified to ensure discoverability of the paper in online searches.

TEXT – the main text consists of the Introduction, Results, and Discussion sections and must not exceed 3500 words including the abstract. The Introduction should expand on the background relating to the work. The Results should be divided in subsections with subheadings, and should provide a concise and accurate description of the experimental findings. The Discussion should expand on the findings and their implications. All relevant primary literature should be cited, in particular when discussing the background and specific findings.

REFERENCES – are limited to a total of 70 in the main text and Methods combined,. They must be numbered sequentially as they appear in the main text, tables and figure legends and Methods and must follow the precise style of Nature Cell Biology references. References only cited in the Methods should be numbered consecutively following the last reference cited in the main text. References only associated with Supplementary Information (e.g. in supplementary legends) do not count toward the total reference limit and do not need to be cited in numerical continuity with references in the main text. Only published papers can be cited, and each publication cited should be included in the numbered reference list, which should include the manuscript titles. Footnotes are not permitted.

Methods should be written concisely, but should contain all elements necessary to allow interpretation and replication of the results. As a guideline, Methods sections typically do not exceed 3,000 words. The Methods should be divided into subsections listing reagents and techniques. When citing previous methods, accurate references should be provided and any alterations should be noted. Information must be provided about: antibody dilutions, company names, catalogue numbers and clone numbers for monoclonal antibodies; sequences of RNAi and cDNA probes/primers or company names and catalogue numbers if reagents are commercial; cell line names, sources and information on cell line identity and authentication. Animal studies and experiments involving human subjects must be reported in detail, identifying the committees approving the protocols. For studies involving human subjects/samples, a statement must be included confirming that informed consent was obtained. Statistical analyses and information on the reproducibility of experimental results should be provided in a section titled "Statistics and Reproducibility".

All Nature Cell Biology manuscripts submitted on or after March 21 2016, must include a Data availability statement as a separate section after Methods but before references, under the heading "Data Availability". For Springer Nature policies on data availability see <http://www.nature.com/authors/policies/availability.html>; for more information on this particular policy see <http://www.nature.com/authors/policies/data/data-availability-statements-data-citations.pdf>. The Data availability statement should include:

- Accession codes for primary datasets (generated during the study under consideration and designated as "primary accessions") and secondary datasets (published datasets reanalysed during the study under consideration, designated as "referenced accessions"). For primary accessions data should be made public to coincide with publication of the manuscript. A list of data types for which submission to community-endorsed public repositories is mandated (including sequence, structure, microarray, deep sequencing data) can be found here <http://www.nature.com/authors/policies/availability.html#data>.
- Unique identifiers (accession codes, DOIs or other unique persistent identifier) and hyperlinks for datasets deposited in an approved repository, but for which data deposition is not mandated (see here for details <http://www.nature.com/sdata/data-policies/repositories>).
- At a minimum, please include a statement confirming that all relevant data are available from the authors, and/or are included with the manuscript (e.g. as source data or supplementary information), listing which data are included (e.g. by figure panels and data types) and mentioning any restrictions on availability.
- If a dataset has a Digital Object Identifier (DOI) as its unique identifier, we strongly encourage including this in the Reference list and citing the dataset in the Methods.

We recommend that you upload the step-by-step protocols used in this manuscript to the Protocol Exchange. More details can be found at www.nature.com/protocolexchange/about.

DISPLAY ITEMS – main display items are limited to 6-8 main figures and/or main tables. For Supplementary Information see below.

FIGURES – Colour figure publication costs \$395 per colour figure. All panels of a multi-panel figure must be logically connected and arranged as they would appear in the final version. Unnecessary figures and figure panels should be avoided (e.g. data presented in small tables could be stated briefly in the text instead).

All imaging data should be accompanied by scale bars, which should be defined in the legend. Cropped images of gels/blots are acceptable, but need to be accompanied by size markers, and to retain visible background signal within the linear range (i.e. should not be saturated). The boundaries of panels with low background have to be demarked with black lines. Splicing of panels should only be considered if unavoidable, and must be clearly marked on the figure, and noted in the legend with a statement on whether the samples were obtained and processed simultaneously. Quantitative comparisons between samples on different gels/blots are discouraged; if this is unavoidable, it has to be performed for samples derived from the same experiment with gels/blots were processed in parallel, which needs to be stated in the legend.

- Some programs can generate Postscript by 'printing to file' (found in the Print dialogue). If using an application not listed above, save the file in PostScript format or email our Art Editor, Allen Beattie for

advice (a.beattie@nature.com).

Regardless of format, all figures must be vector graphic compatible files, not supplied in a flattened raster/bitmap graphics format, but should be fully editable, allowing us to highlight/copy/paste all text and move individual parts of the figures (i.e. arrows, lines, x and y axes, graphs, tick marks, scale bars etc). The only parts of the figure that should be in pixel raster/bitmap format are photographic images or 3D rendered graphics/complex technical illustrations.

Unprocessed scans of all key data generated through electrophoretic separation techniques need to be presented in a supplementary figure that should be labeled and numbered as the final supplementary figure, and should be mentioned in every relevant figure legend. This figure does not count towards the total number of figures and is the only figure that can be displayed over multiple pages, but should be provided as a single file, in PDF or TIFF format. Data in this figure can be displayed in a relatively informal style, but size markers and the figures panels corresponding to the presented data must be indicated.

The total number of Supplementary Figures (not including the “unprocessed scans” Supplementary Figure) should not exceed the number of main display items (figures and/or tables (see our Guide to Authors and March 2012 editorial <http://www.nature.com/ncb/authors/submit/index.html#suppinfo>; <http://www.nature.com/ncb/journal/v14/n3/index.html#ed>). No restrictions apply to Supplementary

Tables or Videos, but we advise authors to be selective in including supplemental data.

GUIDELINES FOR EXPERIMENTAL AND STATISTICAL REPORTING

REPORTING REQUIREMENTS – To improve the quality of methods and statistics reporting in our papers we have recently revised the reporting checklist we introduced in 2013. We are now asking all life sciences authors to complete two items: an Editorial Policy Checklist (found here <https://www.nature.com/authors/policies/Policy.pdf>) that verifies compliance with all required editorial policies and a Reporting Summary (found here <https://www.nature.com/authors/policies/ReportingSummary.pdf>) that collects information on experimental design and reagents. These documents are available to referees to aid the evaluation of the manuscript. Please note that these forms are dynamic 'smart pdfs' and must therefore be downloaded and completed in Adobe Reader. We will then flatten them for ease of use by the reviewers. If you would like to reference the guidance text as you complete the template, please access these flattened versions at <http://www.nature.com/authors/policies/availability.html>.

Author Rebuttal, Second Revision:

Point-by-point response to reviewers (NCB-LE48087B-Z):

Reviewer #2:

There still remains one point to needs be evaluated. The authors should show lipophagy occurs in neurons. Because a lipophagy receptor is previously undetermined, the two publications (PMCID 21803288; PMCID 33713908) lack the actual evidence showing that lipophagy (LD-specific autophagy) can occur in neurons. Therefore, the authors should determine if Spartin-DN causes LD accumulation via inhibiting lipophagy in neurons. To do this, the authors can express Spartin-DN in SUM159 cells to easily see whether the mutant inhibits lipophagy in a Keima-LiveDrop system. In the manuscript, the authors just showed Spartin-DN caused LD accumulation, which does not directly indicate that Spartin-DN inhibits lipophagy. Once the paper is published, researchers must use Spartin-DN as a lipophagy inhibitory factor. That's why the authors need to examine if the mutant truly inhibits lipophagy.

We thank this reviewer for his/her critical and helpful evaluation of our manuscript. In response to the reviewer's critique, we conducted additional experiments to validate the selective inhibition of lipophagy by Spartin-dominant negative (DN) expression in SUM159 cells. We now show the results from three additional experiments, supporting our conclusion that expression of dominant-negative spartin disrupts lipophagy (see Extended Data Fig. 9, d to h). Specifically, we show new data that i) dominant negative spartin interferes with delivery of Keima-LiveDrop to an acidic compartment (Extended Data Fig. 9, d and e); ii) dominant negative spartin fused to Keima is less efficiently turned over (Extended Data Fig. 9, f and g); and iii) dominant negative spartin interferes with the association of LAMP1-labeled lysosomes and lipidTOX-stained LDs (Extended Data Fig. 9h).

Decision Letter, Third Revision:

Our ref: NCB-LE48087C

18th April 2023

Dear Tobi,

Thank you very much for submitting your revised manuscript "The Troyer syndrome protein spartin mediates selective autophagy of lipid droplets" (NCB-LE48087C). We have editorially evaluated the final revision and greatly appreciated your efforts to provide evidence that the DN spartin mutant, in cells, is associated with reduced levels of Keima-LiveDrop and accumulation of LDs. Analyses of the mutant's association with acidic compartments (using Keima-Spartin constructs) and its interference with association of LAMP1-labeled endolysosomes and lipidTOX-stained LDs are also important new pieces of data. Therefore, we'll be happy in principle to publish the study in Nature Cell Biology, pending minor revisions to comply with our editorial and formatting guidelines.

Please note that the current version of your manuscript is in a PDF format. Could you please email us a copy of the file in an editable format (Microsoft Word or LaTeX), as we can not proceed with PDFs at this stage? Many thanks for your attention to this point.

With the Word file in-hand, we will be performing detailed checks on your paper and will send you a checklist detailing our editorial and formatting requirements in about 1-2 weeks. Please do not upload the final materials and make any revisions until you receive this additional information from us.

Thank you again for your interest in Nature Cell Biology and for all your efforts through the rounds of revision to add support to the conclusion that spartin acts as lipophagy receptor. Please do not hesitate to contact me if you have any questions.

Sincerely,

Melina

Melina Casadio, PhD
Senior Editor, Nature Cell Biology
ORCID ID: <https://orcid.org/0000-0003-2389-2243>

Decision Letter, Final Checks:

Our ref: NCB-LE48087C

26th April 2023

Dear Dr. Walther,

Thank you for your patience as we've prepared the guidelines for final submission of your Nature Cell Biology manuscript, "The Troyer syndrome protein spartin mediates selective autophagy of lipid droplets" (NCB-LE48087C). Please carefully follow the step-by-step instructions provided in the attached file, and add a response in each row of the table to indicate the changes that you have made. Please also check and comment on any additional marked-up edits we have proposed within the text. Ensuring that each point is addressed will help to ensure that your revised manuscript can be swiftly handed over to our production team.

In recognition of the time and expertise our reviewers provide to Nature Cell Biology's editorial process, we would like to formally acknowledge their contribution to the external peer review of your manuscript entitled "The Troyer syndrome protein spartin mediates selective autophagy of lipid droplets". For those reviewers who give their assent, we will be publishing their names alongside the published article.

Nature Cell Biology offers a Transparent Peer Review option for new original research manuscripts submitted after December 1st, 2019. As part of this initiative, we encourage our authors to support increased transparency into the peer review process by agreeing to have the reviewer comments, author rebuttal letters, and editorial decision letters published as a Supplementary item. When you submit your final files please clearly state in your cover letter whether or not you would like to participate in this initiative. Please note that failure to state your preference will result in delays in accepting your manuscript for publication.

Cover suggestions

As you prepare your final files we encourage you to consider whether you have any images or illustrations that may be appropriate for use on the cover of Nature Cell Biology.

Nature Cell Biology has now transitioned to a unified Rights Collection system which will allow our Author Services team to quickly and easily collect the rights and permissions required to publish your work. Approximately 10 days after your paper is formally accepted, you will receive an email in providing you with a link to complete the grant of rights. If your paper is eligible for Open Access, our Author Services team will also be in touch regarding any additional information that may be required to arrange payment for your article.

Please note that *Nature Cell Biology* is a Transformative Journal (TJ). Authors may publish their research with us through the traditional subscription access route or make their paper immediately open access through payment of an article-processing charge (APC). Authors will not be required to make a final decision about access to their article until it has been accepted. Find out more about Transformative Journals

Please use the following link for uploading these materials:
[Redacted]

Best regards,

Kendra Donahue
Staff
Nature Cell Biology

On behalf of

Melina Casadio, PhD
Senior Editor, Nature Cell Biology
ORCID ID: <https://orcid.org/0000-0003-2389-2243>

Final Decision Letter:

Dear Dr Walther,

I am pleased to inform you that your manuscript, "The Troyer syndrome protein spartin mediates selective autophagy of lipid droplets", has now been accepted for publication in Nature Cell Biology. Congratulations on this beautiful study!

Please note that *Nature Cell Biology* is a Transformative Journal (TJ). Authors may publish their research with us through the traditional subscription access route or make their paper immediately

open access through payment of an article-processing charge (APC). Authors will not be required to make a final decision about access to their article until it has been accepted. Find out more about Transformative Journals

If you have not already done so, we strongly recommend that you upload the step-by-step protocols used in this manuscript to the Protocol Exchange (www.nature.com/protocolexchange), an open online resource established by Nature Protocols that allows researchers to share their detailed experimental know-how. All uploaded protocols are made freely available, assigned DOIs for ease of citation and are fully searchable through nature.com. Protocols and Nature Portfolio journal papers in which they are used can be linked to one another, and this link is clearly and prominently visible in the online versions of both papers. Authors who performed the specific experiments can act as primary authors for the Protocol as they will be best placed to share the methodology details, but the Corresponding Author of the present research paper should be included as one of the authors. By uploading your Protocols to Protocol Exchange, you are enabling researchers to more readily reproduce or adapt the methodology you use, as well as increasing the visibility of your protocols and papers. You can also establish a dedicated page to collect your lab Protocols. Further information can be found at www.nature.com/protocolexchange/about

With kind regards,

Melina

Melina Casadio, PhD

Senior Editor, Nature Cell Biology
ORCID ID: <https://orcid.org/0000-0003-2389-2243>

** Visit the Springer Nature Editorial and Publishing website at www.springernature.com/editorial-and-publishing-jobs for more information about our career opportunities. If you have any questions please click here.**